# A CRITICAL ANALYSIS OF OUT-OF-DISTRIBUTION DETECTION FOR DOCUMENT UNDERSTANDING

## ABSTRACT

Large-scale pretraining is widely used in recent document understanding models. During deployment, one may expect that large-scale pretrained models should trigger a conservative fallback policy when encountering out-of-distribution (OOD) samples, which suggests the importance of OOD detection. However, most existing OOD detection methods focus on single-modal inputs such as images or texts. While documents are multi-modal in nature, it is underexplored if and how multi-modal information in documents can be exploited for OOD detection. In this work, we first provide a systematic and in-depth analysis on OOD detection for document understanding models. We study the effects of model modality, pretraining, and finetuning across various types of OOD inputs. In particular, we find that spatial information is critical for document OOD detection. To better exploit spatial information, we propose a simple yet effective spatial-aware adapter, which serves as an add-on module to adapt transformer-based language models to document domain. Extensive experiments show that our method consistently improves ID accuracy and OOD detection performance compared to baselines. We hope our findings can help inspire future works on understanding OOD robustness for documents.

## 1 INTRODUCTION

The recent success of large-scale pretrained models has led to the widespread deployment of deep models in various applications. In the document domain, model predictions are increasingly used to help humans make decisions in important applications ranging from tax form processing, machine learning assistant medical reports analysis, deep analyses from financial forms, *etc*. However, in most cases, models are pretrained on collected data but are then deployed in an environment with a different distribution over the observed data (Cui et al., 2021). For example, with the outbreak of COVID-19 (Velavan & Meyer, 2020), machine-assisted medical document analysis systems have to face continually changing data distributions. This motivates the need for reliable methods in the document domain to detect out-of-distribution (OOD) inputs.

The goal of OOD detection is to categorize in-distribution (ID) test samples into one of the known categories and detect instances that do not belong to any known classes (Huang & Li, 2021; Bendale & Boult, 2016). Generally, a model is optimized on a particular task (*e.g*., image classification (Deng et al., 2009)), and a companion OOD detector is built as a safeguard for the classifier. Recently, large-scale pretrained models have demonstrated promising results in multiple domains (Dosovitskiy et al., 2021; Hendrycks et al., 2020) as pretraining enables models to learn powerful and transferable feature representations (Radford et al., 2021). In particular, the models obtained by finetuning large-scale pretrained models are significantly better at OOD detection even with a simple distance metric (Lee et al., 2018; Radford et al., 2021).

It is underexplored whether existing OOD detection methods that demonstrate success for images or text can be naturally extended to documents. The main challenges posed in document OOD detection stem from the fact that document understanding is inherently multi-modal, thus, it is suboptimal to rely on a single

modality. The majority of recent OOD detection approaches focus on single-modal learning (Hsu et al., 2020; Zhou et al., 2021; Xu et al., 2021a; Jin et al., 2022), and they are not compatible with document understanding tasks which require multi-modal learning. The spatial relationship of text blocks in documents further differentiated the document multimodal learning from the multimodal learning in vision-language domain (Lu et al., 2019; Li et al., 2020). In addition, recent document pretraining methods have demonstrated remarkable performance on various downstream document understanding tasks (Xu et al., 2020; 2021b; Huang et al., 2022; Li et al., 2021; Cui et al., 2021; Hong et al., 2022; Gu et al., 2022; Wang et al., 2022). However, existing pretraining datasets for documents are limited and lack diversity, in sharp contrast to common pre-trainining datasets for natural images. Therefore, it is not obvious which OOD detection methods are reliable in the document domain and how pretraining impacts OOD robustness.

This paper investigates the OOD robustness in the document domain through the following questions: *(1)* Are pretrained models robust to OOD examples? Is further pretraining beneficial? How do the pretraining data and tasks affect the performance? *(2)* How does multimodality (textual, visual, and spatial) affect OOD robustness? *(3)* Are existing OOD detection methods developed for natural images and texts transferrable to documents? We present a large-scale evaluation of recent approaches. We focus on models pretrained on different data types and evaluate them on a diverse range of document understanding benchmarks across visual, textual, and spatial modalities. Our key contributions are summarized as follows:

- We show that pretraining datasets and tasks significantly impact OOD detection performance. Through extensive pretraining and finetuning experiments, we find that higher finetuning performance on ID data does not usually translate to better performance on OOD data. This observation emphasizes the importance of considering metrics beyond ID performance for measuring model reliability.

- We propose a spatial-aware adapter, which can serve as an add-on module to transformer-based models and learn the spatial-aware representation. Our method can easily transfer the pretrained language models to the document domain. Extensive experiments show that our method can consistently improve ID accuracy and OOD detection performance across a broad spectrum of datasets.

- We show that recent conclusions drawn from OOD detection methods are valid for images and texts but do not always transfer to documents. For a wide range of document models, we observe that OOD samples are easier to identify in the feature space than in the logit space.

The rest of the paper is organized as follows. Sec. 2 provides the preliminaries and related works. In Sec. 3, we provide a comprehensive analysis of OOD robustness for document models and conclude in Sec. 4.

## 2 PRELIMINARIES AND RELATED WORKS

### 2.1 DOCUMENT MODELS AND PRETRAINING

Large-scale pretrained models have attracted a lot of attention in the document domain. In vision or natural language processing (NLP) tasks, pretraining has shown great success in producing generic representations that learn from large-scale unlabeled corpora (Devlin et al., 2018; Lu et al., 2019; Su et al., 2019; He et al., 2020). Document pretraining also seeks to find universal representations suitable for any downstream task. However, the unique characteristics of document images distinguish document pretraining works from previous ones in vision or language domains. For documents, the contents are spatially distributed, and visual and textual information co-occurs within the semantic regions. In contrast, inputs in the language domain are pure texts, and inputs in the vision-language domain are image-text pairs.

Recent document pretraining models differ in architecture and objectives, as depicted in Fig. 2. LayoutLM (Xu et al., 2020) extends BERT to learn contextualized word representations for document images through multi-task learning. It takes a sequence of Optical Character Recognition (OCR) (Smith, 2007)

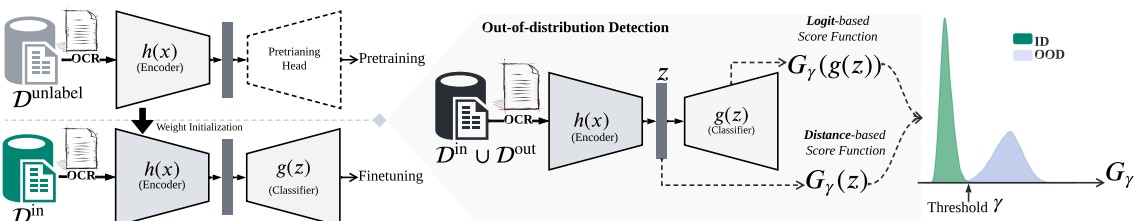

Figure 1: Schematic description of OOD detection for document classification. The left part shows the pretraining and finetuning pipelines. During inference time, for a given input document image, we calculate the OOD detection score $G_\gamma$ according to different methods (logit-base or distance-base). The OOD detector will identify the input document as OOD if the OOD score is smaller than the threshold value $\gamma$.

words and word bounding boxes as inputs during pretraining and finetuning. LayoutLMv2 (Xu et al., 2021b) improves on the prior work by including an image encoder in pretraining and training them jointly. Like LayoutLMv2, DocFormer (Appalaraju et al., 2021) also adopts a CNN model to extract image grid features. It fuses the spatial information as an inductive bias for the self-attention module. The latest version, LayoutLMv3 (Huang et al., 2022), shares similar ideas as LayoutLMv2 and further enhances the visual and spatial characteristics by introducing two other tasks: masked image modeling and word-patch alignment. Another line of works for document pretraining focuses on different granularities of document images and takes region-level text blocks as the basic input elements, such as SelfDoc (Li et al., 2021) and UDoc (Gu et al., 2021). The pretraining tasks of SelfDoc and UDoc are based on feature space. They adopt a cross-modal encoder to model the relationship between visual and textual features. Instead of using the spatial information at the input layer, SelfDoc and UDoc encode the 2D spatial information with a linear mapping and fuse the position embeddings at the output layer of the image encoder and sentence encoder. Despite the promising performance of those pretrained models on downstream applications, it remains largely underexplored whether recent document pretraining models are robust to various types of OOD data, the role of pretraining and finetuning, and the key factors for document OOD detection.

## 2.2 OUT-OF-DISTRIBUTION DETECTION

Many OOD detection methods have been proposed for deep models, including generative model-based methods (Ge et al., 2017; Oza & Patel, 2019; Nalisnick et al., 2019; Ren et al., 2019; Xiao et al., 2020; Morteza & Li, 2022), and discriminative-model based methods. For the latter category, an OOD scoring function can be derived based on the softmax output or logit space (Liu et al., 2020; Hsu et al., 2020; Huang & Li, 2021; Liang et al., 2018; Sun et al., 2021), gradient information (Huang et al., 2021), or the feature space (Sastry & Oore, 2020; Sehwag et al., 2021; Winkens et al., 2020; Sun et al., 2022) of a classifier. Despite their impressive performance, most of the scores are developed for natural images and text inputs. A recent work (Larson et al., 2022) studies OOD detection performance for documents, but only explores a limited number of models and OOD detection methods. Furthermore, the relationship between pretraining, finetuning, and spatial information is underexplored. In this work, we provide a finer-grained and comprehensive analysis and hope to shed light on the key factors of OOD robustness for documents.

**Notations** We denote the input and label space $\mathcal{X}^{\text{in}}$ and $\mathcal{Y}^{\text{in}} = \{1, \ldots, K\}$, respectively. Let $\mathcal{D}^{\text{in}} = \{(\boldsymbol{x}_i^{\text{in}}, y_i^{\text{in}})\}_{i=1}^N$ denote an ID dataset, where $\boldsymbol{x} \in \mathcal{X}^{\text{in}}$ is the input feature vector, and $y^{\text{in}} \in \mathcal{Y}^{\text{in}}$ denotes the semantic label for $K$-way classification. Let $\mathcal{D}^{\text{out}} = \{(\boldsymbol{x}_i^{\text{out}}, y_i^{\text{out}})\}_{i=1}^M$ denote an OOD test set where $y^{\text{out}} \in \mathcal{Y}^{\text{out}}$, and $\mathcal{Y}^{\text{out}} \cap \mathcal{Y}^{\text{in}} = \emptyset$. OOD detection can be formulated as a binary classification problem, which aims to distinguish between ID and OOD data. We express the neural network model $f := g \circ h$ as a composition of a feature extractor $h : \mathcal{X}^{\text{in}} \to \mathbb{R}^d$ and a classifier $g : \mathbb{R}^d \to \mathbb{R}^K$, which maps the feature

embedding of an input to $K$ real-valued numbers known as logits. During inference time, OOD detection can be performed by exercising a thresholding mechanism $G_\gamma(\boldsymbol{x}) = \mathbb{1}\{S(\boldsymbol{x}) \geq \gamma\}$ where by convention samples with higher scores $S(\boldsymbol{x})$ are classified as ID and vice versa. The threshold $\gamma$ is typically chosen so that a high fraction of ID data (*e.g.*, 95%) is correctly classified.

We group OOD detection methods into two major categories: logit-based scores are derived from the logit layer of the model, while distance-based methods are directly based on the feature embedding layer, as shown in Fig. 1. We describe a few popular OOD detection methods for each category as follows.

- *Logit-based:* Maximum Softmax Probability (MSP) score (Hendrycks & Gimpel, 2017) $S_{\mathrm{MSP}} = \max_{i \in [K]} e^{f_i(\boldsymbol{x})} / \sum_{j=1}^{K} e^{f_j(\boldsymbol{x})}$ naturally arises as a classic baseline since logits can be converted to a categorical distribution $p(y|\boldsymbol{x})$; Energy score (Liu et al., 2020): $S_{\mathrm{Energy}} = \log \sum_{i \in [K]} e^{f_i(\boldsymbol{x})}$ utilizes the Helmholtz free energy of the data and theoretically aligns with the logarithm of the ID density; MaxLogit score (Hendrycks et al., 2022): $S_{\mathrm{Maxlogit}} = \max_{i \in [K]} f_i(\boldsymbol{x})$ removes the softmax function in MSP and demonstrates promising performance on large-scale natural image datasets recently.

- *Distance-based*: Distance-based methods directly leverage feature embeddings $h$ based on the idea that OOD inputs are relatively far away from ID centroids or prototypes. Depending on the distributional assumption of feature embeddings, methods can be characterized as 1) parametric methods such as Mahalanobis score (Lee et al., 2018; Sehwag et al., 2021) which assumes ID embeddings follow class-conditional Gaussian distributions and use Mahalanobis distance from the ID centroid as the distance metric; 2) non-parametric methods such as KNN+ (Sun et al., 2022) which uses cosine similarity as the distance metric.

**Evaluation Metrics**  To evaluate OOD detection performance, we adopt two commonly used metrics (Hendrycks & Gimpel, 2017): Area Under the Receiver Operating Characteristic (AUROC) and False Positive Rate at 95% Recall (FPR95). For ID test sets, we report Accuracy (Acc), F1 score, and Mean Average Precision (mAP).

## 3  ANALYZING OOD ROBUSTNESS FOR DOCUMENT MODELS

In this section, we consider the task of document classification, where models are expected to classify documents into categories such as *scientific papers*, *resumes*, *etc.* However, it is underexplored whether models are robust to OOD samples at test time. Most document classification datasets exist in the form of images (Harley et al., 2015). Usually, the first step is to pass the image through an OCR system to obtain a set of text blocks along with their coordinates in the image. Given the input image, extracted words, and coordinates, models can utilize single-modal or multi-modal information to classify the document.

**Models**  Fig. 2 (a) shows common structures for document image pretraining and classification models[1]. According to the input modalities, we categorize them into the following groups:

*(1) Vision-based*: Since current document datasets exist as images, we can treat document classification as the standard image classification problem. In our experiments, we consider ResNet-50 (He et al., 2016) and ViT (Fort et al., 2021) as exemplar document image classification models. As for pretrained weights, we consider two settings: pretrained on ImageNet (Deng et al., 2009) and further pretrained on IIT-CDIP (Lewis et al., 2006). We adopt masked image modeling (MIM) for image pretraining with a mask ratio of 0.6. Note that the document classification dataset we used in this paper, RVL-CDIP, is a subset of IIT-CDIP. Hence, unless otherwise specified, the IIT-CDIP pretraining data used in this paper excludes RVL-CDIP.

---

[1]See Appendix A.1.2 for further details about the models and hyperparameters.

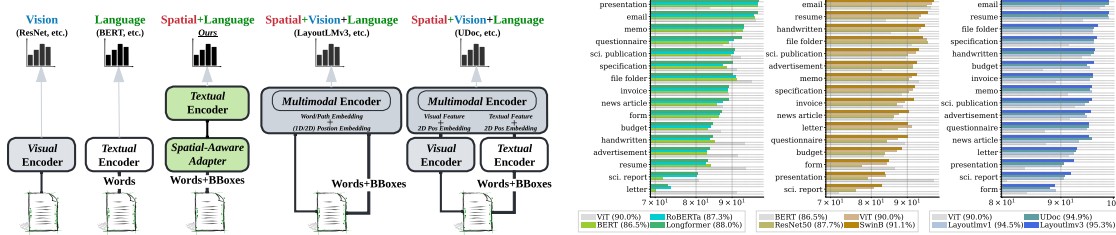

(a) Illustration of common structures for document pre-training/classification.

(b) A detailed comparison of per-category accuracy on the RVL-CDIP test set.

Figure 2: (**Left**) Illustration of models for document pretraining and classification. Our proposed spatial-aware pretraining and finetuning models are the network architectures in green blocks. We also show the modality information on top of each architecture. (**Right**): Evaluating finetuning performance for document classification of pretrained models. We group models into three groups (from left to right): language-only, vision-only, and multimodal. For each group, we also present the performance of a model in another group (shown in grey) for better reference. The average accuracy for each model is shown in the parenthesis.

*(2) Text-based:* Alternatively, we can define the classification as a text classification problem since documents typically contain words. In our experiments, we consider RoBERTa (Liu et al., 2019) as the backbone and append a classifier for finetuning. Since some documents such as scientific papers consist of sentences with more than 512 tokens, we also consider Longformer (Beltagy et al., 2020), which can handle a maximum of 4,096 input tokens. Similar to the vision-based models, we further pretrain the language models with masked language modeling (MLM) on IIT-CDIP extracted text corpus.

*(3) Text+Spatial:* Layout information plays a crucial role in the document domain. As shown in Fig. 3, a document is composed of words or images with some specific layouts. To investigate the effect of layout information, we adopt LayoutLM as a baseline. It is specifically designed for documents and trained on the full IIT-CDIP data. Inspired by the promising OOD detection performance of spatial-aware models (Sec 3.3) and the recent advances in adapter-based transformers (Pfeiffer et al., 2020), we propose a new spatial-aware adaptor, a small learned module that can be inserted within a pretrained language model. Besides the simplicity, our adapter is competitive for both ID classification and OOD detection (Sec 3.4).

*(4) Visual+Textual+Spatial:* Current state-of-the-art methods tailored to documents consider various input granularity and modality and utilize textual, visual, and spatial information for document tasks. Despite the promising performance, such models are large in size and computationally heavy. We select two representative models to evaluate upon: LayoutLMv3 and UDoc. For a fair comparison, both models are pretrained on full IIT-CDIP.

**Constructing ID and OOD Datasets**   We construct ID datasets from RVL-CDIP (Harley et al., 2015). Specifically, we specify 12 out of 16 classes as ID classes. For OOD datasets, we consider two scenarios:

*(1) In-domain OOD*: To determine the OOD categories, we extensively analyze the performance of recent document classification models. Fig. 2(b) shows a detailed comparison of per-category accuracy on the RVL-CDIP test set. Naturally, for the classes the model performs poorly on, we may expect models to detect such inputs as OOD instead of assigning a specific ID class with low confidence. We observe that the 4 categories *letter*, *form*, *scientific report*, *presentation* result in the worst performance across most of models with different modality, which we use as OOD categories and construct the OOD datasets accordingly. The ID dataset is constructed from the remaining 12 categories. We refer to these OOD datasets as *in-domain*, as they are also constructed from RVL-CDIP.

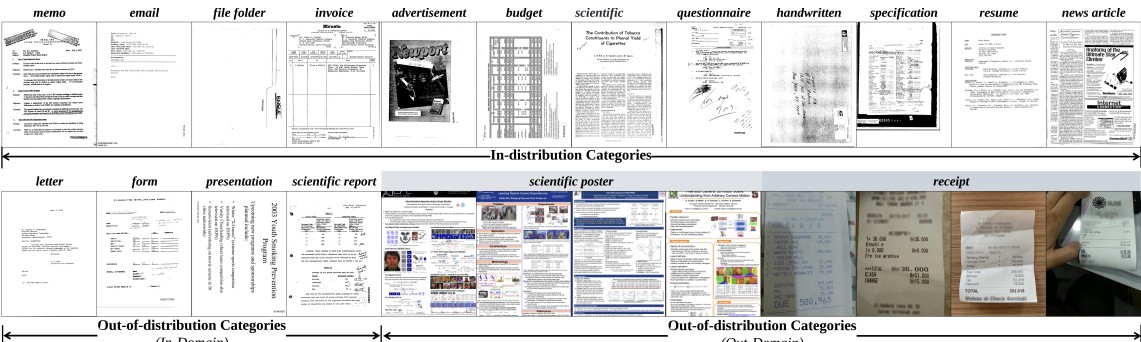

Figure 3: (**Top**) Examples of ID inputs sampled from RVL-CDIP (top). (**Bottom**) In-domain OOD from RVL-CDIP, and out-domain OOD from *Scientific Poster* and *Receipts*.

*(2) Out-domain OOD*: In the open-world setting, test inputs can have significantly different color schemes and layouts compared to ID samples. To mimic such scenarios, we use two public datasets as the *out-domain* OOD test sets. Specifically, NJU-Fudan Paper-Poster Dataset (Qiang et al., 2019) contains scientific posters in the digital PDF format, and we extract the document contents with [2]. CORD (Park et al., 2019) is a receipt understanding dataset that contains significantly different inputs from that in RVL-CDIP. As shown in Fig. 3, document images in CORD are receipt images without creases or warping, which requires the model to be capable of handling text information but also visual and spatial information.

In the following sections, we provide detailed analysis and share insights on various aspects of OOD detection performance for document understanding models under different OOD detection methods. Further details on the setup are provided in Appendix A.

### 3.1 Are pretrained models sufficient for OOD detection?

As shown in Sec. 2.1, most domain processing models deployed in the real world are pretrained on a large-scale dataset. Naturally, one may expect pre-trained models to be robust to OOD data when equipped with competitive OOD detection methods. To better understand the role of pretraining, we first provide more nuanced discussions on the following questions: 1) Are models equally robust to in-domain and out-domain OOD inputs? 2) How does model modality impact OOD detection performance?

We consider a wide range of models pretrained on pure-text/image data (*e.g.*, ImageNet, Wikipdeia, *etc*). A detailed description of these models can be found in Appendix A.1.2. During finetuning, we combine the pretrained model with a classifier and finetune on RVL-CDIP (ID). For models before and after finetuning, we extract the final feature outputs as the feature embeddings and use the same KNN+ score (Sun et al., 2022) for OOD detection. The results are shown in Figure 4. We observe the following trends. First, finetuning largely improves OOD detection performance for both in-domain and out-domain OOD data. Pretrained models, despite the fact that they have "seen" a diverse collection of data during pretraining, do not yield sufficient OOD robustness. The same trend holds broadly across models with different modalities. Second, the improvement of finetuning is less significant for out-domain OOD data. For example, the AUROC on Receipt (out-domain OOD) for pretrained ViT model is 97.13, whereas finetuning only improves by 0.79%. This suggests that pretrained models do have the potential to separate data from different domains due to the diversity of data used for pretraining, while it remains hard for pretrained models to perform finer-grained separation for in-domain OOD inputs. Therefore, finetuning is beneficial for improving both types of OOD detection performance as a consequence of improved feature representation.

---

[2]https://github.com/pymupdf/PyMuPDF

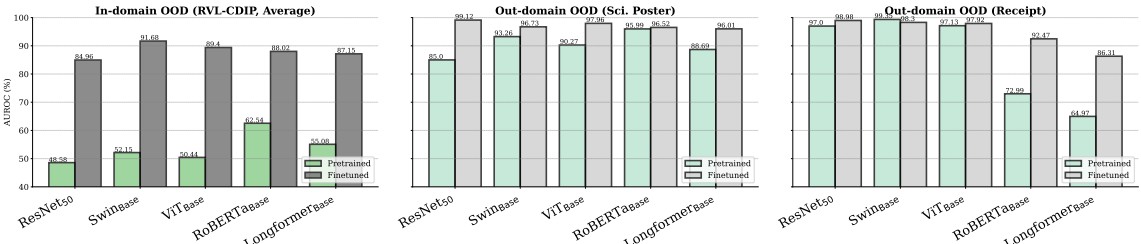

Figure 4: OOD detection performance for pretrained models w. and w.o. finetuning based on distance-based score KNN+(k=10). Finetuning significantly improves performance for both in and out-domain OOD.

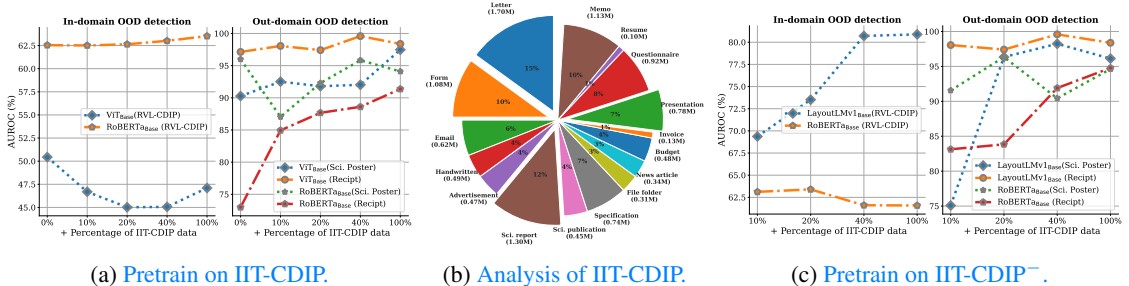

(a) Pretrain on IIT-CDIP.      (b) Analysis of IIT-CDIP.      (c) Pretrain on IIT-CDIP⁻.

Figure 5: The impact of pretraining data on zero-shot OOD detection performance. IIT-CDIP⁻ denotes the filtered pretraining data after removing the "*OOD*" categories.

To make the analysis more thorough, we have two additional in-domain OOD settings: (1) select the classes the model performs well on, as in-domain OOD categories; (2) randomly select classes as OOD categories. As shown in Appendix (Table 10 and Table 11), we can see that finetuning also improves both types of OOD detection, which further reaffirm our conclusion. We also visualize the optimal transport dataset distance (OTDD) (Alvarez-Melis & Fusi, 2020) between in-domain and out-domain OOD datasets in Appendix (Fig. 10(b) and Fig. 10(c)).Please refer to the Appendix for more details.

## 3.2    THE IMPACT OF PRETRAINING DATA ON ZERO-SHOT OOD DETECTION

In the previous section, we analyze the impacts of finetuning for OOD detection where the pretraining dataset is fixed and unrelated to documents. Next, we dive deeper and study the impacts of pretraining dataset on zero-shot OOD detection. For each model, we adopt the same pretraining objective while adjusting the amount of pretraining data. Specifically, we increase the data diversity by appending 10, 20, 40, and 100% of randomly sampled data from IIT-CDIP dataset (around 11M) and pretrain each model. After pretraining, we measure OOD detection performance with KNN+ score based on feature embeddings.

For out-domain OOD data (Fig. 5, right), increasing the amount of pretraining data can significantly improve the zero-shot OOD detection performance (w.o. finetuning) for models across different modalities. This further verifies our previous hypothesis that pretraining with diverse data is beneficial for coarse-grained OOD detection, such as inputs from different domains (*e.g.*, color schemes). On the other hand, for in-domain OOD inputs, even increasing the amount of pretraining data by over 40% provides negligible improvements (Fig. 5, left). This also suggests the necessity of finetuning for improving in-domain OOD detection.

We further explore zero-shot OOD detection by removing the potential OOD categories from IIT-CDIP. In practice, we first adopt the LayoutLMv1 finetuned on RVL-CDIP as the classifier for predicting labels for all IIT-CDIP document images. Fig. 5(b) shows the distribution of the predicted classes on IIT-CDIP. Next, we

remove the "*OOD*" categories from the IIT-CDIP data and pretrain two models (RoBERTa and LayoutLMv1) with 10, 20, 40, and 100% of randomly sampled data from the filtered IIT-CDIP. Fig. 5(c) shows our zero-shot OOD performance. Note that we do not show 0% in Fig. 5(c) since we pretrain LayoutLMv1 from scratch. For RoBERTa, we start from the public pretrained model and see a similar trend in Fig. 5(c) – the influence of pretraining for those well-pretrained language models is minor for in-domain OOD detection since there is a considerable gap between OCR words and pure-text data. *E.g.*, words in a document are spatially arranged, while words in text corpus are arranged sequentially. In contrast, pretraining data has a bigger impact on those models trained from scratch – the zero-shot performance of LayoutLMv1 increases when more pretraining data is added. We provide more details in the Appendix (Table 4 and Table 5).

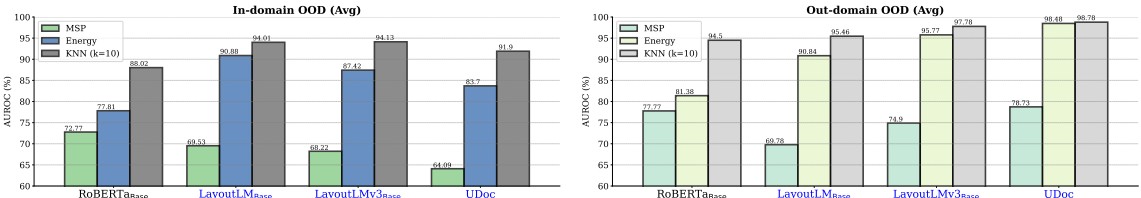

Figure 6: Comparison between representative feature-based scores and logit-based scores for spatial-aware and non-spatial-aware models. Spatial-aware models are colored in blue.

### 3.3 INVESTIGATING SPATIAL-AWARE MODELS FOR OOD DETECTION

In previous sections, we mainly focus on mainstream text-based and vision-based models to analyze the effects of finetuning and pretraining on in- and out-domain OOD detection. Next, we largely expand the scope of our study by incorporating models tailored to document processing, which we refer to as spatial-aware models, such as layoutLM, LayoutLMv3, and UDoc. Moreover, given finetuned models, we are able to compare the performance of logit-based scores and distance-based scores. Some key comparisons are shown in Figure 6. Please refer to the Appendix for full results.

**Distance-based vs. logit-based score** We can see that simple KNN-based score outperforms logit-based scores for both in-domain and out-domain OOD data. The trend holds consistently across models with different modalities. Moreover, spatial-aware models demonstrate both stronger OOD detection performance for in and out-domain OOD. For example, with the best scoring function (KNN+), compared to RoBERTa, LayoutLMv3 improves the average AUROC by 7.09% for out-domain OOD and 7.54%. The significant improvement suggests the value of spatial and visual information in improving OOD robustness for document data. Note that despite this paper mainly comparing the logit-based and distance-based scores, we need to be aware that Gradient-based score has also been proposed for OOD detection. We also report the GradNorm (Huang et al., 2021) OOD detection score in Appendix A.3 and achieves similar performance as logit-based scores.

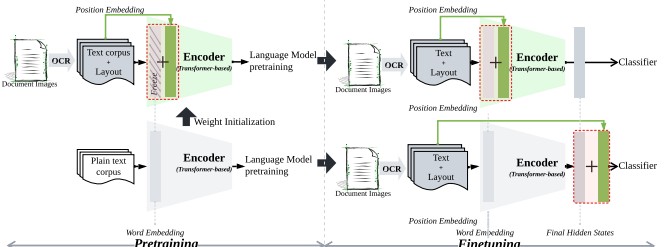

Figure 7: Illustration of our spatial-aware adapter design for language models. We have two adapter designs (marked in red box): *(1)* insert the adapter into the word embedding layer during pretraining and finetuning; *(2)* insert the adapter into the output layer for finetuning only. For the first design, we freeze the word embedding layer and learn the adapter and transformer layers.

### 3.4 TOWARDS SIMPLE AND EFFECTIVE SPATIAL-AWARE ADAPTORS

Spatial-aware models tailored for documents such as LayoutLM rely on spatial information and demonstrates superior OOD detection performance. This brings us a question: given the abundance of well-pretrained large-scale language models on text data such as RoBERTa, is there a simple and effective method that allows us to exploit the pretrained language model to document inputs for effective OOD detection? Next, we show that by enhancing transformer-based pretrained models with a spatial-aware adapter module, we can achieve good performance with minimal code edits.

**Spatial-aware adapter** Given a public pretrained RoBERTa model, depending on the position of the adaptor, we consider two architectures: 1) appending the adapter to the word embedding layer, denoted as Spatial-RoBERTa (pre). It requires pretraining and finetuning, as illustrated in the top row in Fig. 7; 2) appending the adapter to the final layer, denoted as Spatial-BoBERTa (post). As the model can utilize pretrained textual encoder, it only requires finetuning (illustrated in the bottom row). In the following, we only discuss Spatial-RoBERTa (pre). Full results for both Spatial-RoBERTa variants are in the Appendix.

We freeze the word embedding layer during pretraining for the following considerations: 1) word embeddings learned from large-scale corpus already cover most of those words from documents; 2) pretraining on documents without strong language dependency may not help improve word embeddings. For example, in semi-structured documents (*e.g.*, forms, receipts), language dependencies are not be as strong as rich-text documents (*e.g.*, letters, resumes), which may degenerate the learned word representations. In practice, each word has a normalized bounding box (x0, y0, x1, y1), where (x0, y0) / (x1, y1) corresponds to the position of the upper left / lower right in the bounding box. Each coordinate is fed into an embedding layer and outputs a position embedding. All position embeddings are added to the initial word embedding to form a new spatial-aware embedding.

**Spatial-RoBERTa significantly outperforms RoBERTa** To verify the effectiveness of Spatial-RoBERTa, We compare OOD detection performance for pre-trained and fine-tuned models. The results are shown in Fig. 8. Spatial-RoBERTa significantly improves the OOD detection performance, especially after finetuning. Specifically, compared to RoBERTa, Spatial-RoBERTa improves AUROC by 9.20% for in-domain OOD and 4.95% for out-domain OOD data. This further verifies the importance of spatial-awareness for OOD detection in the document domain.

**Spatial-RoBERTa is competitive for both ID classification and OOD detection** Beyond OOD detection performance, we also examine ID classification accuracy and plot the two metrics for all the models with different modalities in Fig. 9. We find that there exists a positive correlation between ID accuracy and OOD detection performance (measured by AUROC) for both in-domain and out-domain OOD. Moreover, spatial-aware models display superior ID accuracy and OOD robustness compared to text-only and vision-only models. Finally, our Spatial-RoBERTa provides a simple and effective solution that greatly improves upon RoBERTa and matches the performance of models with specific architecture such as LayoutLM. Specifically,

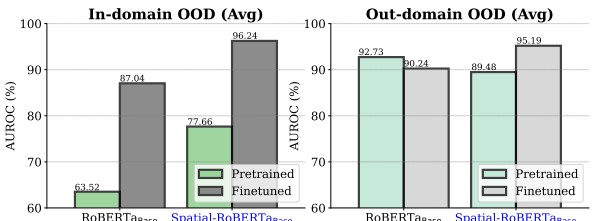

Figure 8: OOD detection performance between Spatial-RoBERTa and RoBERTa. All models are initialized with public pretrained checkpoints on pure-text data and further pretrained on IIT-CDIP with the same pretraining tasks. The only difference here is that Spatial-RoBERTa has an additional spatial-ware adapter and takes the word bounding boxes as the additional inputs.

Spatial-RoBERTa$_{Large}$ achieves 97.37 ID accuracy, which is even higher than LayoutLM (97.28) and UDoc (97.36). Furthermore, since our Spatial-RoBERTa$_{Base}$ and Spatial-RoBERTa$_{Large}$ freeze the word embed-

ding during pretraining, it can learn spatial-aware feature target document data while keeping the word embedding fixed, thus reduce the trainable model size and reduce the training cost.

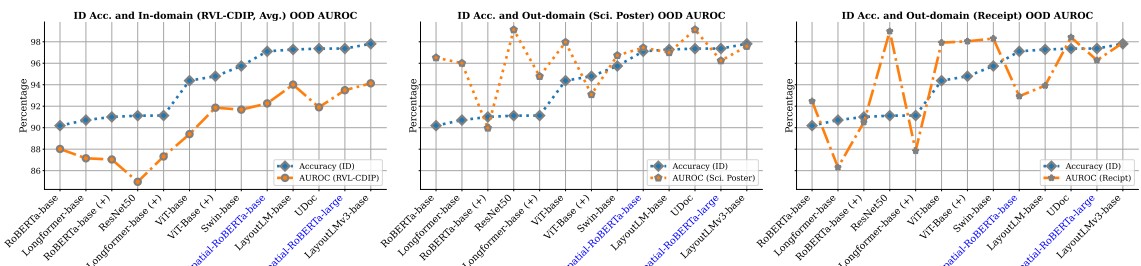

Figure 9: Correlation between ID accuracy and OOD detection performance. For most models, ID accuracy is positively correlated with OOD detection performance. Spatial-aware models display both higher ID accuracy and stronger OOD robustness (in AUROC).

## 4 CONCLUSION AND OUTLOOK

This paper presents a large-scale study of various methods for quantifying OOD robustness across different data modalities and models for document domains. Our key novelties include a large-scale investigation of OOD robustness in the document domain and a simple yet powerful spatial-aware adapter for transformer-based language models. We start from document classification and explore the pretrained models for document OOD robustness. A variety of substantial experiments in different settings demonstrates that pretraining datasets and tasks greatly impact OOD detection performance. Notably, OOD samples in the document domain are more accessible to identify in the feature space than in the logit space. Investigations from various perspectives explain certain intriguing phenomena and inspire more research on evaluating OOD robustness towards more reliable document understanding models.

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

# A APPENDIX

## A.1 DOCUMENT CLASSIFICATION

### A.1.1 DATASETS

RVL-CDIP consists of 320K/40K/40K training/validation/testing images under 16 categories. We select 12 categories and treat them as ID (In-Domain) data. We extract the text and layout information with Google OCR engine[3] which provides both tokens and text blocks along with their corresponding bounding boxes. Most recent models take the full IIT-CDIP as pretraining data and finetuning on RVL-CDIP. However, it is not reasonable for the OOD setting since RVL-CDIP is a subset of IIT-CDIP. To make OOD results more reliable, in our experiments, we exclude the RVL-CDIP from the IIT-CDIP during pretraining.

We measure the distance between in-domain and out-domain datasets via OTDD[4]. We first visualize the OTDD distance between ID and the OOD data (in-domain and out-domain datasets in our main paper) in Fig. 10(a). During analysis, we sample the maximum number of 1000 images from each data and calculate the distance between datasets. It can be seen that there is a clear gap between in-domain data and out-domain data. To make the analysis more thorough, we have two additional in-domain OOD settings: (1) select the classes the model performs well as OOD data; (2) randomly select classes as OOD data. Fig. 10(a) and Fig. 10(c) show the dataset distance. As for the other two selection strategies, we can see that the domain gap is not as clear as the subset we selected for the main experiments. Interestingly, for those rare-word documents, such as file-folder and advertisements, the dataset distances are larger than documents with rich words. The background colors and layouts may yield a big distinction for those documents.

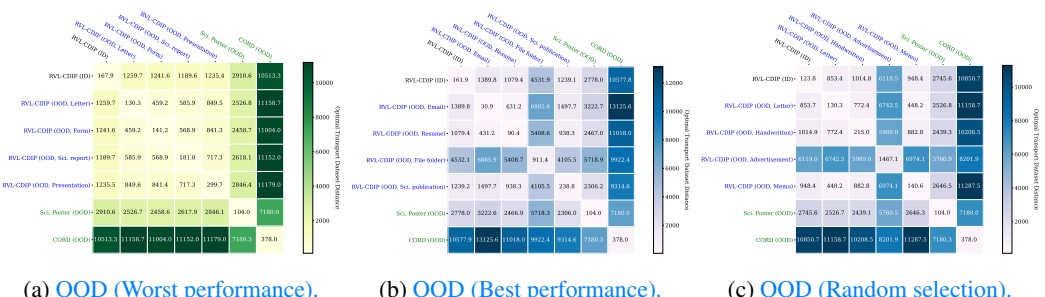

(a) OOD (Worst performance).  (b) OOD (Best performance).  (c) OOD (Random selection).

Figure 10: Visualization of optimal transport dataset distance for ID and OOD (in-domain and out-domain) datasets. We highlight the in-domain data in blue and the out-domain in green.

## A.1.2 MODELS

All models reported in Fig. 2(b), except for UDoc, are initialized with model pretrained weights from Huggingface[5] and finetune on the full RVL-CDIP training set. During finetuning, we train those models on RVL-CDIP with the cross-entropy loss. Models are optimized with Adam optimizer (Kingma & Ba, 2014) for 30 epochs with a batch size of 50 and a learning rate of $2 \times 10^{-5}$ on 8 A100 GPUs. We list the hyperparameters of models used in our paper as follows:

---

[3] https://cloud.google.com/vision/docs/ocr
[4] https://github.com/microsoft/otdd
[5] https://huggingface.co/models

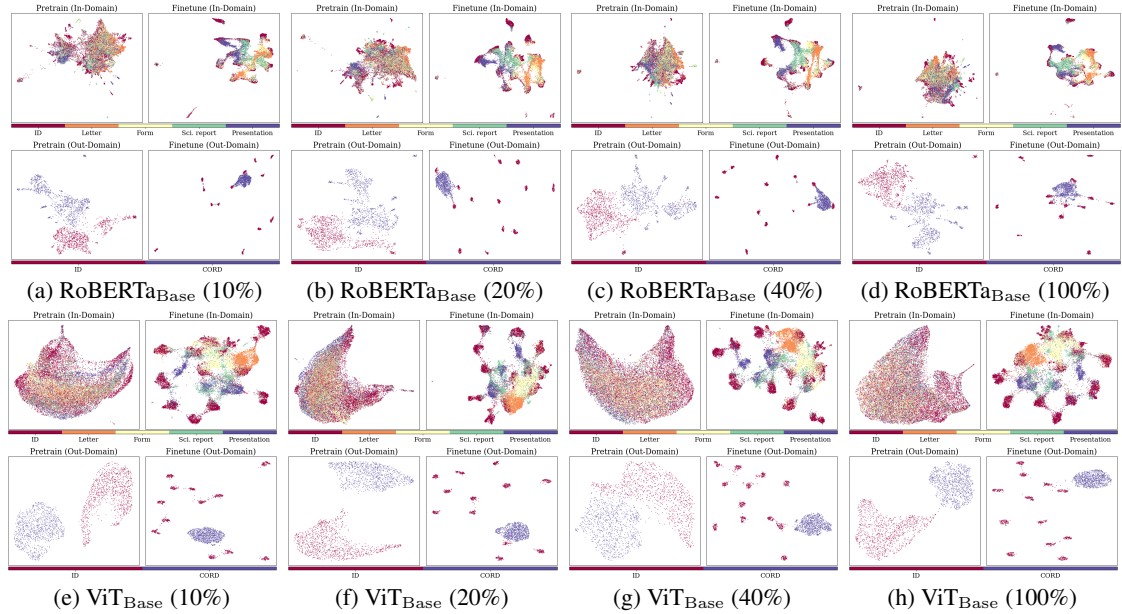

Figure 11: Feature visualization for pretrained (with different numbers of pretraining data) and finetuned models. We show both In-Domain (RVL-CDIP) and Out-Domain (CORD) OOD datasets.

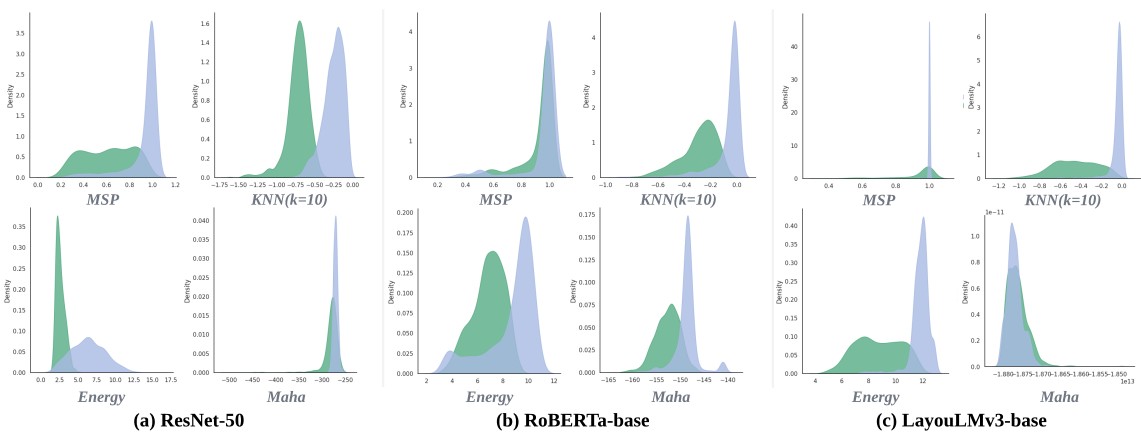

Figure 12: MSP, Energy, KNN, and Maha score histogram distributions of ID (*blue*) and OOD (*green*) inputs derived from finetuned ResNet-50, RoBERTa, and LayoutLMv3. The KNN scores calculated from both vision and language models naturally form smooth distributions. In contrast, MSP and Maha scores for both in- and out-of-distribution data concentrate on high values. Overall our experiments show that using feature space makes the scores more distinguishable between and out-of-distributions and, as a result, enables more effective OOD detection.

**Language-only:** *(1)* BERT and RoBERTa. We adopt the RoBERTa$_{\text{Base}}$ (12 layer, 768 hidden size) and BERT$_{\text{Base}}$ (12 layer, 768 hidden size) as the backbone and set the maximum sequence length to 512. For RoBERTa, the classifier is composed of two linear layers followed by a tanh activation function. *(2)* Longformer. We also adopts the Longformer$_{\text{Base}}$ (12 layer, 768 hidden size) as the backbone and sets the maximum sequence length to 4,096.

**Vision-only:** *(1)* ResNet50: This model adopts the ResNet50 (pretrained on ImageNet-1k) as the backbone. We finetune it at resolution 224×224. *(2)* ViT: This model adopts ViT$_{\text{Base}}$ (vit-base-patch16-224, pretrained on ImageNet-21k) as the visual backbone. We finetune it at resolution 224x224. *(3)* SwinB: This model adopt swin transformer (swin-base-patch4-window7-224-in22k, pretrained on ImageNet-21k) as backbone. We finetune it at resolution 224×224.

**Layout+Language:** *(1)* LayoutLMv1: This model adopts the LayoutLM (layoutlm-base-uncased, 12 layer, 768 hidden size, pretrained on IIT-CDIP ) as the backbone. We set the maximum sequence length to 512. *(2)* Spatial-RoBERTa$_{\text{Base}}$ (Pre): This model is combines our spatial-aware adapter to the pretrained RoBERTa$_{\text{Base}}$ model. The adapter is applied to the word embedding layer. During pretraining, we freeze the pretrained word embedding and optimize the spatial-aware adapter and transformers. During finetuning, we optimize all the parameters. *(3)* Spatial-RoBERTa$_{\text{Base}}$ (Post): Instead of inserting the spatial-aware adapter in the input layer, this model combines the spatial-aware adapter at the transformers' output layer.

**Vision+Language+Layout:** *(1)* LaytouLMv3: This model adopt the LayoutLMv3 (layoutlmv3-base, 12 layer, 768 hidden size, pretrained on IIT-CDIP) as the backbone. *21)* UDoc: This model follows the design of UDoc. The only difference is the sentence encoder, we adopt a smaller version of the pretrained sentence encoder (all-MiniLM-L6-v2, 6 layer, 384 hidden size) instead of the sentence encoder (bert-base-nli-mean-tokens, 12 layer, 768 hidden size) used in their paper.

## A.2 BEYOND DOCUMENT CLASSIFICATION

Going beyond document classification, we explore OOD detection for two entity-level tasks: document entity recognition and document object detection. Basic units such as text, tables, and figures in the document are the objects that need to be detected and recognized. Document entity recognition aims to predict the label for each semantic entity with given bounding boxes. Document object detection is an object detection task for document images. Specifically, we denote the input as $x$, the bounding box coordinates associated with object instances in the image as $\boldsymbol{b} \in \mathbb{R}^4$, and use the model with parameters $\theta$ to model the bounding box regression

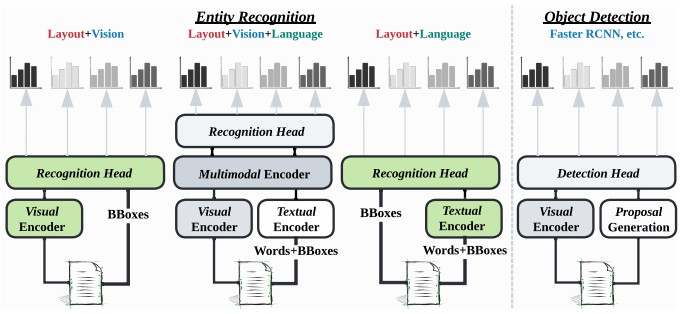

Figure 13: The network architectures in green blocks are our proposed models. We also show the modality information on top of each architecture.

$p_\theta(\boldsymbol{b}|x)$ and the label classification $p_\theta(y|x, \boldsymbol{b})$. Given a test input $\hat{x}$, the OOD detection scoring function for entity detection and recognition can be unified as $S(\hat{x}, \hat{\boldsymbol{b}})$, where $\hat{\boldsymbol{b}}$ denotes the object instance predicted by the object detector. In particular, for document entity recognition, since the bounding boxes are provided, the OOD score can be simplified as $S(\hat{x}, \bar{\boldsymbol{b}})$, where $\bar{\boldsymbol{b}}$ is the given object instance.

### A.2.1 DATASETS

**Document Entity Recognition**    The original FUNSD (Jaume et al., 2019) dataset contains 149/50 training/testing images. we treat entities with category *other*/*header* as the OOD entities. After doing the split, if we treat *other* as OOD, we have a total number of 8,330/1,019 ID/OOD entities in total. Otherwise, if we treat *header* as OOD, we have 8,981/368 ID/OOD entities in total.

**Document Object Detection**    PubLayNet (Zhong et al., 2019) contains 336K/11K training/validation images with 6 category labels (*text*, *title*, *list*, *figure*, and *table*). The original IIIT-AR-13K (Mondal et al., 2020) contains (*table*, *figure*, *natural image*, *logo*, and *signature*). In our paper, considering the overlap between IIIT-AR-13K and PubLayNet, we select those images that contain *Natural Image* as the OOD test set. After filtering, we have 2,880 OOD entities across 1,837 document images.

We consider three ID datasets in this experiment. *(1)* PubLayNet: This is the original PubLayNet dataset. We treat all the entities in training/validation images as ID entities. *(2)* Considering the domain shift between ID data (PubLayNet) and OOD data (IIIT-AR-13K). We combine the PubLayNet training data with the images from IIIT-AR-13K with overlapping annotations (*table* and *figure*) and train the object detection model.

### A.2.2 MODELS

**Document Entity Recognition**    Fig. 13 illustrates the entity recognition models used in this paper. We consider the entity on regions instead of tokens since regions contain richer semantic information. As for the pretrained model, we adopt UDoc (pretrained on IIT-CDIP) since it models the inputs at the region level. Based on UDoc framework, we develop the following models.

Vision/Vision+Layout: *(1)* ResNet-50: This model is composed of the ResNet-50 from pretrained UDoc. It adopts the RoI pooling followed by a classifier to extract the entity features. *(2)* ResNet-50+Position: This model also adapts UDoc pretrained ResNet-50. It further improves the RoI features to be spatial-aware by adding position embedding, where the position embeddings are mapped from bounding boxes via a linear mapping layer.

Language/Language+Layout: *(1)* Sentence BERT: This model adopts the language branch of UDoc and appends the classifier to the output of the sentence encoder. *(2)* Sentence BERT+Position: This model is close to the above model but adds position embedding to the sentence embeddings.

Vision+Language+Layout: *(1)* ResNet-50+sentence BERT: This model follows the same framework as UDoc, but replaces the sentence encoder in their original design with a more miniature sentence encoder (all-MiniLM-L6-v2). *(2)* SwinT+Sentence BERT: This model replaces the ResNet-50 visual backbone with a pretrained Swin tiny model (swin-tiny-patch4-window7-224) adopted from the Huggingface.

All the models are finetuned with cross-entropy loss for 100 epochs with a learning rate of $10^{-5}$ and batch size of 8 on one A100 GPU.

**Document Object Detection**    Two object detection models are considered in this paper: *(1)* Vanilla Faster-RCNN: This model is the Faster-RCNN with ResNet-50 visual backbone. *(2)* Faster-RCNN with VOS: This model enhances above model with VOS[6]. Following their paper, we use 1,000 samples for each ID class to estimate the class-conditional Gaussians. We train detection models with the Detectron2 framework (Wu et al., 2019). Models are trained for 180k iterations with a base learning rate of 0.01 and a batch size of 8. Mean average precision (MAP) @ intersection over union (IOU) [0.50:0.95] of bounding boxes is used to measure the performance.

---

[6]https://github.com/deeplearning-wisc/vos

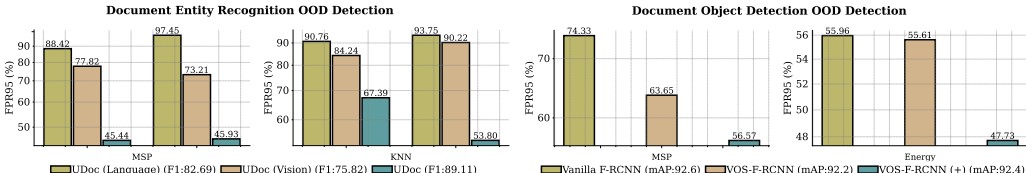

(a) Comparison of OOD detection methods on different models on two OOD classes: *other* and *header*.

(b) OOD detection results from different object detection methods.

Figure 14: Ablation on document entity recognition and object detection. Numbers are reported in FPR95.

### A.2.3 EXPERIMENTAL SETUP

For document entity recognition, we construct ID and OOD datasets from FUNSD. Each semantic entity includes a list of words, a label, and a bounding box. The standard label set for this dataset contains 4 categories: *question*, *answer*, *header*, and *other*. In this paper, we select the entity labeled as *other* or *header* category as OOD. And the entities belonging to the other three categories are ID. Instead of treating entity recognition as a named-entity recognition problem, we follow UDoc and solve this problem at the semantic region level. We replace the sentence encoder in UDoc with a smaller sentence encoder (all-MiniLM-L6-v2[7]) from Huggingface (Wolf et al., 2019). We also have the following model variants to verify the effectiveness of each modality combination: textual-only, visual-only, textual+spatial, visual+spatial, and visual+textual+spatial.

For document object detection, we use PubLayNet as the ID dataset. We construct the OOD dataset from IIIT-AR-13K. Unlike PubLayNet, where the documents are scientific articles, IIIT-AR-13K is a dataset for graphical object detection in business documents (*e.g.*, annual reports). Hence, there exists an obvious domain gap between these two datasets. We select *natural images* as the OOD entity and filter images that contain the OOD entity. We first adopt the vanilla Faster RCNN with ResNet-50 backbone for document object detection as the baseline model. We also enhance Faster RCNN with VOS (Du et al., 2022), a recent unknown-aware learning framework to improve OOD detection performance for natural images.

### A.2.4 OBSERVATIONS

To identify the entity type, models should not only understand the words but also require spatial and visual reasoning ability. We summarize our findings on document entity recognition in Fig. 14 (a) and describe them in more detail in Table 1. We can see that models can better predict the entity type with the help of the spatial position and also achieves better OOD robustness. Considering the weak language dependency between entities, it is not supervising that vision-based models achieve better performance than text-based models. We can see that UDoc with ResNet-50 achieves the best performance on two OOD test sets, illustrating that visual information plays a major role in increasing the discrimination of entities with similar semantics. We summarize our findings on document object detection in Fig. 14 (b) and describe them in more detail in Table 2. We can see that the OOD detection performance is further improved by introducing document images from IIIT-AR-13K with the same ID annotations as training data.

In Fig. 15, we visualize some document entity recognition OOD detection results. In Fig. 16, we visualize the prediction on sample OOD images, using object detection models trained without VOS (top) and with VOS (bottom), respectively. There is a clear difference between PubLayNet and IIIT-AR-13K – *natural image* annotations and entities rarely exist in PubLayNet. We can see that vanilla Faster RCNN trained on PubLayNet produces false positives when applied to the OOD document image from IIIT-AR-13K. After

---

[7]https://huggingface.co/sentence-transformers

introducing the unknown-aware learning method optimized for both ID and OOD, as shown in Table 2, the FPR95 reduces while preserving the mAP on the ID data. This experiment indicates that bringing uncertainty estimation into the entity detection training procedure can improve the reliability of the document object detection system.

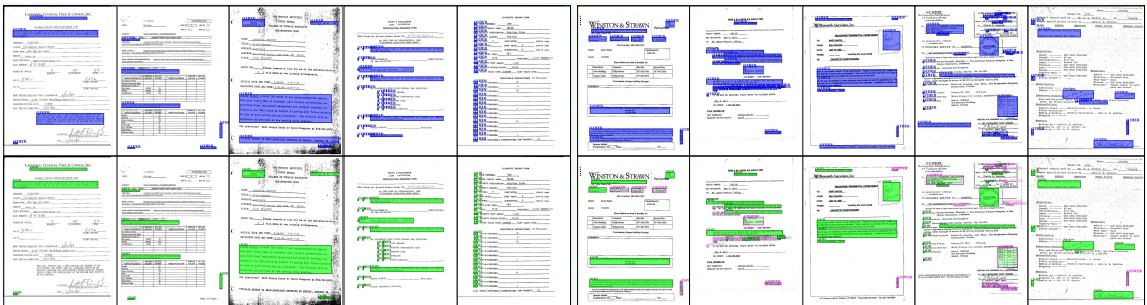

Figure 15: Visualization of detected OOD entities on the form images. The top part shows the entities in blue are entities annotated as *other*. The bottom part shows the detected OOD entities (green). We also show failure cases on the right part.

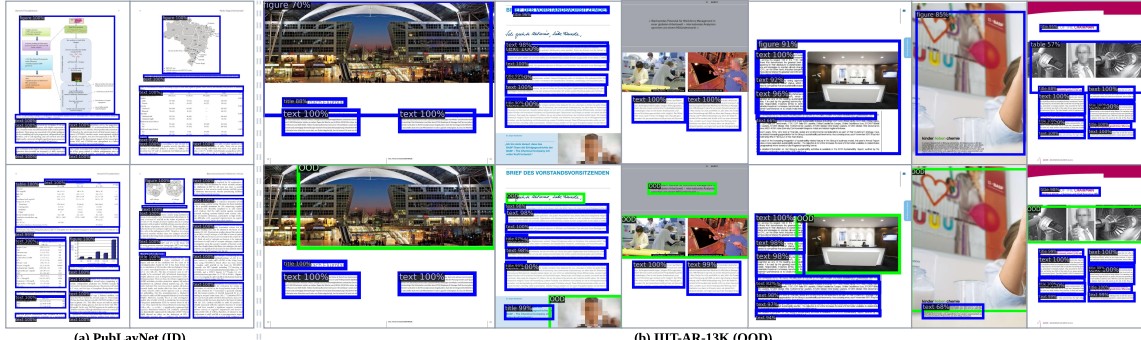

Figure 16: Visualization of detected objects on the OOD images (from IIIT-AR-13K) by a vanilla Faster-RCNN (top) and Faster-RCNN with VOS (bottom). Objects in blue boxes are detected and classified as one of the ID classes. The detected OOD objects (green) reduce false positives among detected objects. We also visualization of detected objects on the ID images. There is a clear difference between PubLayNet and IIIT-AR-13K – *natural image* annotations and entities rarely exist in PubLayNet.

### A.3 ADDITIONAL EXPERIMENTAL RESULTS

- Table 1 corresponds to the results shown in Fig. 15 and Fig. 14(a).
- Table 2 corresponds to the results shown in Fig. 16 and Fig. 14(b).
- Table 3 and Table 7 correspond to the results shown in Fig. 5(a).
- Table 4 and Table 5 correspond to the results shown in Fig. 5(c).
- Table 6 corresponds to the results shown in Fig. 8 and Fig. 9.
- Table 9 and Table 8 correspond to the results shown in Fig. 4 and Fig. 9.
- Table 10 and Table 11 correspond to the analysis in Sec. 3.1.
- Table 12 corresponds to the results shown in Fig. 9.

Table 1: Comparison with different models on FUNSD OOD setting. All models are initialized with UDoc pretrained on IIT-CDIP and finetuned on FUNSD data with ID entities. All values are percentages. A lower FPR95 or higher AUROC value indicates better performance.

| | Test F1 | Method | Other (OOD) FPR95 | AUROC | ID F1 | Header (OOD) FPR95 | AUROC | ID F1 | | Test F1 | Method | Other (OOD) FPR95 | AUROC | ID F1 | Header (OOD) FPR95 | AUROC | ID F1 |
|---|---|---|---|---|---|---|---|---|---|---|---|---|---|---|---|---|---|
| ResNet-50 | 75.15 | Maha$_{Norm}$ | 88.42 | 59.21 | | 92.12 | 41.49 | | ResNet-50+Position | 75.82 | Maha$_{Norm}$ | 74.48 | 69.86 | | 97.28 | 46.61 | |
| | | Maha$_{UnNorm}$ | 94.11 | 29.14 | | 99.46 | 24.06 | | | | Maha$_{UnNorm}$ | 89.11 | 33.51 | | 99.73 | 34.1 | |
| | | KNN$_{10}$ | 59.47 | 79.14 | | 81.79 | 63.97 | | | | KNN$_{10}$ | 73.21 | 73.19 | | 90.22 | 61.42 | |
| | | KNN$_{20}$ | 69.97 | 78.15 | | 81.25 | 63.66 | | | | KNN$_{20}$ | 72.91 | 73.44 | | 88.04 | 61.54 | |
| | | KNN$_{50}$ | 84.49 | 77.40 | | 82.61 | 62.86 | | | | KNN$_{50}$ | 75.96 | 74.43 | | 82.88 | 60.93 | |
| | | KNN$_{100}$ | 97.94 | 77.08 | 77.65 | 84.24 | 61.62 | 78.04 | | | KNN$_{100}$ | 79.69 | 74.85 | 77.65 | 83.70 | 59.39 | 77.98 |
| | | KNN$_{200}$ | 97.84 | 77.15 | | 94.29 | 59.74 | | | | KNN$_{200}$ | 86.06 | 75.14 | | 91.58 | 57.42 | |
| | | KNN$_{400}$ | 97.15 | 76.09 | | 94.84 | 57.53 | | | | KNN$_{400}$ | 87.93 | 74.92 | | 95.92 | 55.37 | |
| | | MSP | 50.54 | 75.80 | | 75.82 | 76.55 | | | | MSP | 77.82 | 67.60 | | 84.24 | 66.58 | |
| | | MaxLogit | 52.40 | 73.70 | | 73.64 | 76.72 | | | | MaxLogit | 76.94 | 67.05 | | 84.24 | 65.41 | |
| | | Energy | 52.50 | 73.70 | | 75.82 | 76.55 | | | | Energy | 76.64 | 66.93 | | 84.51 | 64.98 | |
| Sentence BERT | 77.15 | Maha$_{Norm}$ | 93.33 | 54.99 | | 88.32 | 67.06 | | Sentence BERT+Position | 82.69 | Maha$_{Norm}$ | 95.88 | 51.73 | | 92.66 | 64.25 | |
| | | Maha$_{UnNorm}$ | 93.33 | 55.05 | | 88.59 | 67.67 | | | | Maha$_{UnNorm}$ | 94.90 | 56.61 | | 96.47 | 50.46 | |
| | | KNN$_{10}$ | 93.72 | 48.44 | | 92.66 | 60.99 | | | | KNN$_{10}$ | 97.45 | 41.24 | | 93.75 | 62.38 | |
| | | KNN$_{20}$ | 93.92 | 47.65 | | 92.93 | 59.00 | | | | KNN$_{20}$ | 97.55 | 39.91 | | 93.48 | 61.51 | |
| | | KNN$_{50}$ | 93.62 | 48.94 | | 93.21 | 57.90 | | | | KNN$_{50}$ | 97.15 | 39.56 | | 92.39 | 61.76 | |
| | | KNN$_{100}$ | 93.92 | 48.79 | 82.12 | 93.21 | 55.07 | 82.41 | | | KNN$_{100}$ | 97.06 | 41.67 | 87.08 | 91.85 | 60.99 | 87.01 |
| | | KNN$_{200}$ | 93.92 | 47.85 | | 93.48 | 52.86 | | | | KNN$_{200}$ | 96.57 | 41.85 | | 89.67 | 59.08 | |
| | | KNN$_{400}$ | 94.11 | 46.21 | | 95.38 | 49.86 | | | | KNN$_{400}$ | 97.25 | 40.83 | | 90.22 | 54.03 | |
| | | MSP | 93.62 | 54.91 | | 94.29 | 52.14 | | | | MSP | 88.42 | 61.11 | | 90.76 | 59.58 | |
| | | MaxLogit | 93.72 | 54.75 | | 94.57 | 56.51 | | | | MaxLogit | 89.70 | 60.19 | | 88.86 | 60.92 | |
| | | Energy | 93.23 | 54.88 | | 93.21 | 58.22 | | | | Energy | 90.48 | 59.61 | | 89.95 | 61.12 | |
| ResNet-50+Sentence BERT | 89.11 | Maha$_{Norm}$ | 33.27 | 95.02 | | 31.25 | 94.65 | | SwinT+Sentence BERT | 86.00 | Maha$_{Norm}$ | 59.57 | 88.56 | | 78.80 | 77.33 | |
| | | Maha$_{UnNorm}$ | 94.70 | 27.16 | | 57.07 | 79.68 | | | | Maha$_{UnNorm}$ | 75.56 | 58.18 | | 92.66 | 56.17 | |
| | | KNN$_{10}$ | 45.93 | 87.85 | | 53.80 | 87.97 | | | | KNN$_{10}$ | 63.30 | 83.64 | | 81.52 | 64.08 | |
| | | KNN$_{20}$ | 53.58 | 86.71 | | 55.71 | 87.06 | | | | KNN$_{20}$ | 66.73 | 82.53 | | 81.52 | 61.50 | |
| | | KNN$_{50}$ | 73.21 | 84.36 | | 62.77 | 85.49 | | | | KNN$_{50}$ | 70.17 | 80.21 | | 82.34 | 57.77 | |
| | | KNN$_{100}$ | 89.70 | 83.01 | 93.13 | 69.02 | 83.60 | 93.18 | | | KNN$_{100}$ | 83.91 | 77.71 | 90.82 | 83.15 | 54.97 | 90.40 |
| | | KNN$_{200}$ | 96.66 | 81.90 | | 75.54 | 80.85 | | | | KNN$_{200}$ | 95.39 | 75.79 | | 95.38 | 50.57 | |
| | | KNN$_{400}$ | 98.82 | 81.00 | | 91.58 | 77.42 | | | | KNN$_{400}$ | 96.76 | 75.49 | | 99.73 | 47.45 | |
| | | MSP | 45.44 | 87.82 | | 67.39 | 72.85 | | | | MSP | 69.28 | 70.70 | | 80.71 | 52.02 | |
| | | MaxLogit | 45.53 | 90.58 | | 63.04 | 72.39 | | | | MaxLogit | 67.12 | 74.41 | | 81.79 | 52.77 | |
| | | Energy | 45.53 | 90.57 | | 63.86 | 72.37 | | | | Energy | 67.22 | 74.41 | | 81.79 | 52.77 | |

Table 2: Comparison with different training and detection methods.

| Models | ID | Method | IIIT-AR-13K (*Natural Image* as OOD) FPR95 | AUROC | AUPR | PubLayNet (ID) mAP |
|---|---|---|---|---|---|---|
| Vanilla Faster-RCNN | PubLayNet | MSP | 74.33 | 79.12 | 98.41 | 92.6 |
| | | Energy | 55.96 | 83.55 | 98.73 | |
| Faster-RCNN with VOS | PubLayNet | MSP | 63.65 | 79.37 | 98.57 | 92.2 |
| | | Energy | 55.61 | 80.60 | 98.67 | |
| Faster-RCNN with VOS | PubLayNet+IIIT-AR-13K(ID) | MSP | 56.57 | 82.94 | 98.59 | 92.4 |
| | | Energy | 47.73 | 84.04 | 98.67 | |

Table 3: OOD detection performance for document classification with different number of pretraining data from IIT-CDIP. ID (Acc) denotes the ID accuracy obtained by testing on ID test data. We report the KNN-based scores for both pretrained and finetuned models. *Sci. Poster* denotes the document images converted from NJU-Fudan Paper-Poster Dataset. *Receipt* denotes the receipt images collected from the CORD receipt understanding dataset. For in-domain OOD test data, we also report the averaged scores.

| | ID Acc | Method | OOD Dataset (In-Domain) | | | | | | | | | | OOD Dataset (Out-Domain) | | | |
| | | | Sci. Report | | Presentation | | Form | | Letter | | *Average* | | Sci. Poster | | Receipt | |
| | | | FPR95 | AUROC | FPR95 | AUROC | FPR95 | AUROC | FPR95 | AUROC | FPR95 | AUROC | FPR95 | AUROC | FPR95 | AUROC |
|---|---|---|---|---|---|---|---|---|---|---|---|---|---|---|---|---|
| RoBERTa$_{Base}$ (10%) | | **Pretrain on 10% IIT-CDIP→ Finetune on RVL-CDIP ID data** | | | | | | | | | | | | | | |
| | | MSP | 92.75 | 69.24 | 92.21 | 66.93 | 94.65 | 65.40 | 92.00 | 70.09 | 92.90 | 67.92 | 96.51 | 66.93 | 99.10 | 52.90 |
| | | MaxLogit | 98.36 | 77.85 | 97.23 | 78.51 | 98.76 | 72.84 | 98.86 | 78.08 | 98.30 | 76.82 | 100.00 | 78.69 | 100.00 | 63.74 |
| | | Energy | 98.60 | 77.81 | 97.55 | 78.49 | 98.96 | 72.79 | 98.94 | 78.00 | 98.51 | 76.77 | 100.00 | 78.68 | 100.00 | 63.70 |
| | | GradNorm | 98.04 | 79.26 | 97.07 | 76.85 | 98.56 | 72.83 | 98.62 | 80.55 | 98.07 | 77.37 | 100.00 | 85.23 | 100.00 | 64.10 |
| | 90.59 | Maha$_{Norm}$ | 97.72 | 38.35 | 97.11 | 36.97 | 96.45 | 45.02 | 93.43 | 46.56 | 96.18 | 41.72 | 98.84 | 55.19 | 96.80 | 48.45 |
| | | Maha$_{UnNorm}$ | 97.76 | 39.23 | 97.27 | 38.31 | 96.45 | 45.97 | 93.59 | 46.60 | 96.27 | 42.53 | 98.84 | 56.11 | 96.80 | 49.28 |
| | | KNN$_{10}$ | 63.21 | 88.18 | 65.81 | 88.05 | 73.02 | 84.63 | 67.74 | 88.92 | 67.45 | 87.44 | 69.77 | 88.49 | 90.50 | 84.44 |
| | | KNN$_{20}$ | 63.53 | 88.07 | 65.89 | 87.90 | 72.75 | 84.48 | 67.33 | 88.81 | 67.38 | 87.32 | 68.60 | 88.13 | 91.10 | 84.09 |
| | | KNN$_{50}$ | 64.17 | 87.89 | 66.97 | 87.77 | 73.34 | 84.23 | 67.21 | 88.60 | 67.92 | 87.12 | 72.09 | 87.47 | 91.60 | 83.59 |
| | | KNN$_{100}$ | 64.49 | 87.64 | 67.78 | 87.55 | 73.46 | 83.94 | 67.29 | 88.37 | 68.26 | 86.88 | 72.09 | 86.83 | 91.50 | 83.21 |
| | | KNN$_{200}$ | 65.25 | 87.29 | 68.34 | 87.27 | 74.02 | 83.52 | 67.45 | 88.10 | 68.76 | 86.54 | 72.09 | 86.09 | 91.60 | 82.77 |
| | | KNN$_{400}$ | 66.13 | 86.84 | 69.43 | 86.98 | 75.14 | 82.94 | 68.55 | 87.81 | 69.81 | 86.14 | 74.42 | 85.29 | 91.70 | 82.27 |
| | | **Pretrain on 10% IIT-CDIP (no finetune)** | | | | | | | | | | | | | | |
| | | KNN$_{10}$ | 88.07 | 66.94 | 92.13 | 66.62 | 94.13 | 61.90 | 94.40 | 54.57 | 92.18 | 62.51 | 67.44 | 87.04 | 62.10 | 84.94 |
| | | KNN$_{20}$ | 88.59 | 66.02 | 92.65 | 65.25 | 94.13 | 60.83 | 94.72 | 53.79 | 92.52 | 61.47 | 77.91 | 85.38 | 64.60 | 83.86 |
| | — | KNN$_{50}$ | 89.75 | 64.40 | 93.53 | 63.12 | 94.37 | 58.98 | 95.17 | 52.33 | 93.20 | 59.71 | 83.72 | 82.97 | 69.20 | 82.29 |
| | | KNN$_{100}$ | 90.23 | 62.94 | 93.85 | 61.28 | 94.41 | 57.45 | 95.13 | 51.28 | 93.40 | 58.24 | 83.72 | 80.91 | 70.10 | 81.05 |
| | | KNN$_{200}$ | 90.55 | 60.99 | 94.17 | 58.98 | 94.53 | 55.50 | 95.50 | 49.91 | 93.69 | 56.34 | 83.72 | 77.86 | 71.00 | 79.31 |
| | | KNN$_{400}$ | 91.43 | 59.73 | 94.54 | 57.24 | 95.01 | 54.40 | 95.74 | 49.57 | 94.18 | 55.24 | 87.21 | 75.05 | 73.40 | 77.74 |
| RoBERTa$_{Base}$ (20%) | | **Pretrain on 20% IIT-CDIP→ Finetune on RVL-CDIP ID data** | | | | | | | | | | | | | | |
| | | MSP | 94.28 | 68.02 | 94.46 | 65.98 | 96.01 | 62.98 | 94.81 | 65.98 | 94.89 | 65.74 | 95.35 | 63.55 | 99.10 | 54.99 |
| | | MaxLogit | 97.36 | 77.82 | 97.19 | 79.16 | 98.40 | 72.64 | 98.34 | 77.68 | 97.82 | 76.82 | 100.00 | 77.36 | 99.60 | 66.63 |
| | | Energy | 98.04 | 77.80 | 97.43 | 79.15 | 98.76 | 72.61 | 98.58 | 77.64 | 98.20 | 76.80 | 100.00 | 77.32 | 99.60 | 66.61 |
| | | GradNorm | 97.36 | 80.68 | 96.83 | 76.04 | 98.44 | 73.29 | 97.89 | 81.37 | 97.63 | 77.85 | 100.00 | 86.18 | 99.50 | 67.49 |
| | 90.71 | Maha$_{Norm}$ | 98.80 | 36.58 | 98.27 | 35.70 | 97.73 | 39.46 | 98.54 | 36.60 | 98.34 | 37.08 | 98.84 | 49.45 | 95.40 | 52.28 |
| | | Maha$_{UnNorm}$ | 98.92 | 36.39 | 98.39 | 35.38 | 97.77 | 39.26 | 98.54 | 34.94 | 98.40 | 36.49 | 98.84 | 49.82 | 95.40 | 52.25 |
| | | KNN$_{10}$ | 63.57 | 88.30 | 67.06 | 87.06 | 73.66 | 83.92 | 73.09 | 87.80 | 69.34 | 86.77 | 69.77 | 88.01 | 87.60 | 83.81 |
| | | KNN$_{20}$ | 63.85 | 88.20 | 67.46 | 86.90 | 73.94 | 83.73 | 72.93 | 87.70 | 69.54 | 86.64 | 69.77 | 87.63 | 88.20 | 83.53 |
| | | KNN$_{50}$ | 63.89 | 88.02 | 67.54 | 86.71 | 74.38 | 83.55 | 72.24 | 87.46 | 69.51 | 86.43 | 70.93 | 87.09 | 88.20 | 83.12 |
| | | KNN$_{100}$ | 64.85 | 87.81 | 67.62 | 86.45 | 74.90 | 83.25 | 72.65 | 87.24 | 70.00 | 86.19 | 72.09 | 86.65 | 88.30 | 82.89 |
| | | KNN$_{200}$ | 66.45 | 87.51 | 68.78 | 86.13 | 75.86 | 82.83 | 73.38 | 86.97 | 71.12 | 85.86 | 73.26 | 86.09 | 88.60 | 82.57 |
| | | KNN$_{400}$ | 67.69 | 87.16 | 69.35 | 85.78 | 76.38 | 82.20 | 73.50 | 86.71 | 71.73 | 85.46 | 74.42 | 85.37 | 88.50 | 82.03 |
| | | **Pretrain on 20% IIT-CDIP (no finetune)** | | | | | | | | | | | | | | |
| | | KNN$_{10}$ | 87.15 | 68.27 | 90.88 | 66.89 | 92.26 | 62.39 | 95.01 | 53.02 | 91.32 | 62.64 | 43.02 | 92.29 | 57.00 | 87.67 |
| | | KNN$_{20}$ | 87.31 | 67.35 | 92.04 | 65.54 | 91.54 | 61.40 | 94.97 | 52.33 | 91.46 | 61.66 | 47.67 | 91.18 | 62.60 | 86.61 |
| | — | KNN$_{50}$ | 88.39 | 65.71 | 92.69 | 63.45 | 92.18 | 59.57 | 95.25 | 50.97 | 92.13 | 59.92 | 56.98 | 89.64 | 65.70 | 85.20 |
| | | KNN$_{100}$ | 88.83 | 64.20 | 93.13 | 61.61 | 92.22 | 57.99 | 95.45 | 49.95 | 92.41 | 58.44 | 58.14 | 88.36 | 66.90 | 84.17 |
| | | KNN$_{200}$ | 89.07 | 62.15 | 93.21 | 59.28 | 92.66 | 55.89 | 95.45 | 48.35 | 92.60 | 56.42 | 61.63 | 86.59 | 67.50 | 82.68 |
| | | KNN$_{400}$ | 90.71 | 60.75 | 93.85 | 57.44 | 93.38 | 54.58 | 95.86 | 47.79 | 93.45 | 55.14 | 69.77 | 84.71 | 70.10 | 81.21 |
| RoBERTa$_{Base}$ (40%) | | **Pretrain on 40% IIT-CDIP→ Finetune on RVL-CDIP ID data** | | | | | | | | | | | | | | |
| | | MSP | 92.67 | 70.09 | 93.93 | 65.69 | 95.05 | 63.19 | 95.50 | 65.54 | 94.29 | 66.13 | 95.35 | 63.63 | 95.40 | 64.97 |
| | | MaxLogit | 98.08 | 78.72 | 97.87 | 79.85 | 98.44 | 71.63 | 98.30 | 75.41 | 98.17 | 76.40 | 98.84 | 78.07 | 98.90 | 75.65 |
| | | Energy | 98.48 | 78.69 | 97.91 | 79.83 | 98.68 | 71.61 | 98.50 | 75.40 | 98.39 | 76.38 | 100.00 | 78.04 | 98.50 | 75.60 |
| | | GradNorm | 98.04 | 81.03 | 97.47 | 76.73 | 98.44 | 72.77 | 97.40 | 79.11 | 97.84 | 77.41 | 100.00 | 87.47 | 97.60 | 77.12 |
| | 90.76 | Maha$_{Norm}$ | 97.60 | 37.27 | 96.26 | 39.41 | 97.21 | 40.50 | 92.82 | 42.83 | 95.97 | 40.00 | 77.91 | 59.29 | 62.70 | 70.78 |
| | | Maha$_{UnNorm}$ | 97.88 | 37.22 | 96.30 | 40.07 | 97.29 | 40.39 | 94.44 | 39.14 | 96.48 | 39.20 | 77.91 | 59.87 | 62.70 | 71.13 |
| | | KNN$_{10}$ | 60.57 | 88.79 | 68.86 | 86.36 | 75.26 | 83.55 | 73.90 | 87.12 | 69.65 | 86.46 | 67.44 | 89.90 | 72.70 | 89.49 |
| | | KNN$_{20}$ | 61.37 | 88.72 | 69.06 | 86.24 | 75.46 | 83.43 | 73.46 | 87.00 | 69.84 | 86.35 | 68.60 | 89.66 | 73.50 | 89.25 |
| | | KNN$_{50}$ | 62.21 | 88.52 | 69.18 | 86.08 | 75.66 | 83.21 | 73.42 | 86.71 | 70.12 | 86.13 | 70.93 | 89.20 | 74.70 | 88.89 |
| | | KNN$_{100}$ | 63.77 | 88.30 | 69.79 | 85.84 | 76.02 | 82.93 | 74.19 | 86.46 | 70.94 | 85.88 | 74.42 | 88.84 | 75.30 | 88.69 |
| | | KNN$_{200}$ | 64.77 | 88.04 | 70.15 | 85.55 | 76.66 | 82.54 | 74.80 | 86.16 | 71.60 | 85.57 | 74.42 | 88.36 | 76.10 | 88.46 |
| | | KNN$_{400}$ | 65.45 | 87.71 | 71.07 | 85.21 | 77.29 | 82.13 | 74.96 | 85.84 | 72.19 | 85.17 | 74.42 | 87.71 | 75.60 | 88.13 |
| | | **Pretrain on 40% IIT-CDIP (no finetune)** | | | | | | | | | | | | | | |
| | | KNN$_{10}$ | 85.71 | 69.08 | 90.84 | 68.68 | 90.46 | 62.52 | 94.76 | 51.76 | 90.44 | 63.01 | 25.58 | 95.83 | 57.30 | 88.60 |
| | | KNN$_{20}$ | 85.27 | 68.21 | 91.64 | 67.48 | 89.74 | 61.32 | 94.81 | 51.01 | 90.36 | 62.00 | 29.07 | 95.22 | 62.30 | 87.61 |
| | — | KNN$_{50}$ | 86.19 | 66.60 | 92.21 | 65.54 | 90.30 | 59.35 | 94.93 | 49.60 | 90.91 | 60.27 | 41.86 | 94.32 | 66.80 | 86.25 |
| | | KNN$_{100}$ | 87.19 | 65.04 | 92.57 | 63.83 | 90.50 | 57.74 | 95.09 | 48.44 | 91.34 | 58.76 | 45.35 | 93.66 | 68.30 | 85.14 |
| | | KNN$_{200}$ | 87.67 | 63.12 | 92.93 | 61.78 | 91.18 | 55.74 | 95.09 | 46.80 | 91.72 | 56.86 | 48.84 | 92.90 | 69.70 | 83.74 |
| | | KNN$_{400}$ | 88.59 | 61.46 | 93.77 | 60.14 | 92.18 | 54.01 | 95.45 | 45.54 | 92.50 | 55.29 | 50.00 | 91.98 | 72.90 | 82.29 |
| RoBERTa$_{Base}$ (100%) | | **Pretrain on 100% IIT-CDIP→ Finetune on RVL-CDIP ID data** | | | | | | | | | | | | | | |
| | | MSP | 93.23 | 68.88 | 94.54 | 65.83 | 96.65 | 63.11 | 94.12 | 68.28 | 94.64 | 66.53 | 98.84 | 62.52 | 95.10 | 71.25 |
| | | MaxLogit | 97.84 | 78.86 | 97.95 | 80.23 | 98.48 | 74.01 | 98.25 | 77.59 | 98.13 | 77.67 | 100.00 | 78.73 | 98.90 | 79.36 |
| | | Energy | 98.20 | 78.84 | 97.95 | 80.22 | 98.52 | 74.00 | 98.78 | 77.55 | 98.36 | 77.65 | 100.00 | 78.72 | 98.70 | 79.29 |
| | | GradNorm | 97.88 | 80.81 | 97.91 | 76.37 | 98.28 | 75.25 | 98.25 | 80.09 | 98.08 | 78.13 | 100.00 | 86.10 | 98.30 | 77.50 |
| | 91.00 | Maha$_{Norm}$ | 98.32 | 36.79 | 98.88 | 28.27 | 98.32 | 37.34 | 94.93 | 36.82 | 97.61 | 34.80 | 97.67 | 45.20 | 99.10 | 38.95 |
| | | Maha$_{UnNorm}$ | 98.48 | 36.40 | 99.00 | 28.07 | 98.36 | 36.60 | 95.21 | 35.42 | 97.76 | 34.24 | 97.67 | 45.77 | 99.10 | 39.13 |
| | | KNN$_{10}$ | 62.57 | 88.26 | 68.90 | 86.96 | 72.39 | 84.73 | 70.37 | 88.23 | 68.56 | 87.04 | 72.09 | 89.97 | 65.90 | 90.51 |
| | | KNN$_{20}$ | 63.41 | 88.11 | 69.59 | 86.88 | 73.10 | 84.56 | 70.70 | 88.11 | 69.20 | 86.92 | 74.42 | 89.58 | 67.20 | 90.37 |
| | | KNN$_{50}$ | 63.85 | 87.87 | 69.79 | 86.79 | 73.90 | 84.30 | 71.14 | 87.87 | 69.67 | 86.71 | 76.74 | 88.95 | 67.90 | 90.22 |
| | | KNN$_{100}$ | 65.13 | 87.61 | 70.27 | 86.58 | 74.86 | 84.00 | 71.75 | 87.65 | 70.50 | 86.46 | 79.07 | 88.44 | 68.30 | 90.19 |
| | | KNN$_{200}$ | 66.57 | 87.26 | 71.68 | 86.27 | 76.34 | 83.58 | 72.81 | 87.38 | 71.85 | 86.12 | 82.56 | 87.77 | 69.20 | 90.09 |
| | | KNN$_{400}$ | 67.37 | 86.82 | 71.96 | 85.91 | 77.41 | 83.00 | 72.52 | 87.09 | 72.32 | 85.70 | 83.72 | 87.06 | 69.00 | 89.89 |
| | | **Pretrain on 100% IIT-CDIP (no finetune)** | | | | | | | | | | | | | | |
| | | KNN$_{10}$ | 84.43 | 70.20 | 90.20 | 68.54 | 90.98 | 63.18 | 94.72 | 52.16 | 90.08 | 63.52 | 27.91 | 94.10 | 46.00 | 91.37 |
| | | KNN$_{20}$ | 84.51 | 69.30 | 91.28 | 67.35 | 90.38 | 61.96 | 94.72 | 51.43 | 90.22 | 62.51 | 33.72 | 93.39 | 51.50 | 90.55 |
| | — | KNN$_{50}$ | 85.67 | 67.75 | 91.92 | 65.35 | 90.82 | 59.79 | 94.89 | 49.77 | 90.82 | 60.66 | 39.53 | 92.28 | 56.70 | 89.32 |
| | | KNN$_{100}$ | 86.55 | 66.08 | 92.97 | 63.46 | 91.46 | 58.00 | 95.41 | 48.39 | 91.60 | 58.98 | 44.19 | 91.29 | 61.60 | 88.18 |
| | | KNN$_{200}$ | 86.91 | 64.20 | 93.61 | 61.34 | 91.86 | 55.95 | 95.66 | 46.65 | 92.01 | 57.04 | 47.67 | 90.30 | 65.70 | 87.03 |
| | | KNN$_{400}$ | 87.99 | 62.45 | 94.09 | 59.48 | 92.26 | 54.05 | 95.90 | 45.07 | 92.56 | 55.26 | 53.49 | 89.19 | 69.40 | 85.65 |

Table 4: OOD detection performance for document classification with different number of pretraining data from IIT-CDIP⁻ (remove *pseudo* OOD categories).

| | ID Acc | Method | OOD Dataset (In-Domain) | | | | | | | | | | OOD Dataset (Out-Domain) | | | |
| | | | Sci. Report | | Presentation | | Form | | Letter | | *Average* | | Sci. Poster | | Receipt | |
| | | | FPR95 | AUROC | FPR95 | AUROC | FPR95 | AUROC | FPR95 | AUROC | FPR95 | AUROC | FPR95 | AUROC | FPR95 | AUROC |
|---|---|---|---|---|---|---|---|---|---|---|---|---|---|---|---|---|
| | | **Pretrain on 10% IIT-CDIP⁻ → Finetune on RVL-CDIP ID data** | | | | | | | | | | | | | | |
| RoBERTa_Base (10%) | 90.62 | MSP | 90.07 | 69.00 | 89.92 | 68.86 | 92.58 | 64.16 | 91.07 | 66.78 | 90.91 | 67.20 | 96.51 | 54.47 | 96.70 | 59.63 |
| | | MaxLogit | 97.76 | 78.40 | 97.71 | 80.58 | 98.64 | 71.26 | 98.70 | 76.38 | 98.20 | 76.66 | 100.00 | 73.51 | 99.80 | 73.32 |
| | | Energy | 98.16 | 78.35 | 97.75 | 80.55 | 98.84 | 71.20 | 98.90 | 76.32 | 98.41 | 76.60 | 100.00 | 73.46 | 99.80 | 73.31 |
| | | GradNorm | 97.68 | 79.92 | 97.27 | 79.42 | 98.56 | 71.31 | 98.50 | 79.44 | 98.00 | 77.52 | 100.00 | 82.62 | 99.60 | 75.85 |
| | | Maha_Norm | 97.36 | 38.99 | 97.75 | 37.09 | 97.37 | 41.81 | 92.53 | 46.51 | 96.25 | 41.10 | 97.67 | 55.81 | 96.20 | 48.88 |
| | | Maha_UnNorm | 97.40 | 39.19 | 97.87 | 37.86 | 97.41 | 42.54 | 92.49 | 46.82 | 96.29 | 41.60 | 97.67 | 56.33 | 96.10 | 49.50 |
| | | KNN_10 | 65.85 | 87.89 | 66.69 | 88.12 | 75.98 | 82.82 | 74.55 | 86.85 | 70.77 | 86.42 | 87.21 | 85.16 | 83.90 | 87.91 |
| | | KNN_20 | 66.33 | 87.80 | 66.85 | 88.04 | 75.94 | 82.70 | 73.94 | 86.75 | 70.76 | 86.32 | 87.21 | 84.63 | 83.60 | 87.71 |
| | | KNN_50 | 66.77 | 87.66 | 67.30 | 88.00 | 76.02 | 82.49 | 73.66 | 86.52 | 70.94 | 86.17 | 88.37 | 83.73 | 83.90 | 87.34 |
| | | KNN_100 | 67.25 | 87.42 | 67.74 | 87.84 | 76.18 | 82.18 | 73.99 | 86.26 | 71.29 | 85.92 | 89.53 | 82.85 | 83.90 | 86.98 |
| | | KNN_200 | 68.29 | 87.05 | 69.02 | 87.60 | 77.17 | 81.71 | 74.47 | 85.89 | 72.24 | 85.56 | 89.53 | 81.76 | 84.50 | 86.50 |
| | | KNN_400 | 70.14 | 86.55 | 70.03 | 87.37 | 78.13 | 81.03 | 75.53 | 85.51 | 73.46 | 85.12 | 90.70 | 80.78 | 85.60 | 85.87 |
| | | **Pretrain on 10% IIT-CDIP⁻ (no finetune)** | | | | | | | | | | | | | | |
| | – | KNN_10 | 86.35 | 65.48 | 85.74 | 70.84 | 92.94 | 59.55 | 93.14 | 56.62 | 89.54 | 63.12 | 29.07 | 95.42 | 87.60 | 83.13 |
| | | KNN_20 | 86.87 | 64.48 | 87.14 | 69.68 | 93.30 | 58.41 | 93.30 | 55.91 | 90.15 | 62.12 | 37.21 | 94.75 | 88.00 | 81.44 |
| | | KNN_50 | 87.75 | 62.73 | 88.99 | 67.80 | 93.50 | 56.54 | 93.75 | 54.52 | 91.00 | 60.40 | 47.67 | 93.71 | 90.30 | 78.97 |
| | | KNN_100 | 88.43 | 61.17 | 89.59 | 66.05 | 93.62 | 54.91 | 93.99 | 53.40 | 91.41 | 58.88 | 48.84 | 93.09 | 91.50 | 77.00 |
| | | KNN_200 | 88.95 | 59.28 | 90.32 | 63.95 | 94.29 | 53.09 | 94.32 | 52.07 | 91.97 | 57.10 | 52.33 | 92.40 | 92.50 | 74.24 |
| | | KNN_400 | 89.59 | 58.00 | 91.04 | 62.19 | 94.65 | 52.10 | 94.68 | 51.64 | 92.49 | 55.98 | 56.98 | 91.80 | 93.50 | 71.66 |
| | | **Pretrain on 20% IIT-CDIP⁻ → Finetune on RVL-CDIP ID data** | | | | | | | | | | | | | | |
| RoBERTa_Base (20%) | 90.65 | MSP | 96.04 | 67.58 | 94.90 | 68.32 | 96.05 | 64.92 | 96.23 | 68.62 | 95.80 | 67.36 | 100.00 | 61.49 | 98.70 | 56.38 |
| | | MaxLogit | 97.96 | 76.92 | 97.59 | 80.68 | 98.48 | 72.31 | 98.74 | 77.72 | 98.19 | 76.91 | 100.00 | 75.91 | 99.50 | 69.21 |
| | | Energy | 98.16 | 76.89 | 98.23 | 80.65 | 98.88 | 72.26 | 99.07 | 77.67 | 98.58 | 76.87 | 100.00 | 75.89 | 99.50 | 69.18 |
| | | GradNorm | 97.84 | 78.23 | 97.31 | 78.57 | 98.00 | 71.44 | 98.46 | 80.03 | 97.90 | 77.07 | 100.00 | 85.80 | 99.00 | 69.54 |
| | | Maha_Norm | 97.84 | 45.53 | 98.31 | 40.63 | 97.25 | 47.35 | 96.31 | 50.09 | 97.43 | 45.90 | 96.51 | 38.30 | 100.00 | 51.69 |
| | | Maha_UnNorm | 97.88 | 45.43 | 98.27 | 41.24 | 97.25 | 48.10 | 96.31 | 49.86 | 97.43 | 46.16 | 96.51 | 40.44 | 100.00 | 51.42 |
| | | KNN_10 | 66.05 | 87.60 | 67.70 | 87.94 | 73.42 | 83.10 | 73.50 | 87.96 | 70.17 | 86.65 | 77.91 | 90.19 | 90.10 | 84.32 |
| | | KNN_20 | 66.17 | 87.50 | 68.38 | 87.83 | 73.90 | 82.93 | 73.66 | 87.82 | 70.53 | 86.52 | 77.91 | 89.84 | 89.80 | 84.13 |
| | | KNN_50 | 67.21 | 87.26 | 68.46 | 87.73 | 74.18 | 82.63 | 73.66 | 87.58 | 70.88 | 86.30 | 79.07 | 89.24 | 89.60 | 83.80 |
| | | KNN_100 | 68.78 | 86.98 | 69.14 | 87.53 | 75.50 | 82.30 | 74.27 | 87.36 | 71.92 | 86.04 | 82.56 | 88.68 | 89.80 | 83.59 |
| | | KNN_200 | 69.66 | 86.59 | 69.35 | 87.27 | 75.74 | 81.85 | 73.78 | 87.08 | 72.13 | 85.70 | 86.05 | 87.98 | 89.70 | 83.26 |
| | | KNN_400 | 70.74 | 86.14 | 69.63 | 87.06 | 76.02 | 81.24 | 74.03 | 86.87 | 72.60 | 85.33 | 88.37 | 87.24 | 89.30 | 82.88 |
| | | **Pretrain on 20% IIT-CDIP⁻ (no finetune)** | | | | | | | | | | | | | | |
| | – | KNN_10 | 85.63 | 66.10 | 85.17 | 70.34 | 92.58 | 60.29 | 93.43 | 56.85 | 89.20 | 63.40 | 30.23 | 95.72 | 83.20 | 83.84 |
| | | KNN_20 | 86.31 | 65.17 | 85.98 | 69.13 | 93.30 | 59.09 | 93.47 | 56.05 | 89.77 | 62.36 | 34.88 | 95.08 | 84.90 | 82.16 |
| | | KNN_50 | 87.31 | 63.50 | 87.63 | 67.11 | 93.38 | 57.17 | 94.16 | 54.60 | 90.62 | 60.60 | 44.19 | 94.07 | 87.50 | 79.74 |
| | | KNN_100 | 87.83 | 62.06 | 88.27 | 65.31 | 93.62 | 55.65 | 94.32 | 53.56 | 91.01 | 59.14 | 48.84 | 93.48 | 88.80 | 77.77 |
| | | KNN_200 | 88.43 | 60.26 | 89.23 | 63.16 | 94.29 | 53.87 | 94.52 | 52.29 | 91.62 | 57.40 | 50.00 | 92.79 | 89.60 | 74.96 |
| | | KNN_400 | 89.03 | 59.06 | 89.96 | 61.33 | 94.73 | 52.89 | 94.76 | 51.90 | 92.12 | 56.30 | 54.65 | 92.10 | 90.50 | 72.33 |
| | | **Pretrain on 40% IIT-CDIP⁻ → Finetune on RVL-CDIP ID data** | | | | | | | | | | | | | | |
| RoBERTa_Base (40%) | 90.72 | MSP | 93.84 | 68.86 | 93.69 | 67.62 | 95.41 | 63.91 | 94.20 | 66.91 | 94.28 | 66.41 | 96.51 | 63.32 | 98.90 | 54.02 |
| | | MaxLogit | 97.16 | 78.56 | 96.87 | 80.18 | 98.68 | 71.84 | 98.58 | 74.44 | 97.82 | 76.26 | 100.00 | 76.72 | 99.10 | 65.41 |
| | | Energy | 97.40 | 78.53 | 97.15 | 80.17 | 98.68 | 71.79 | 98.78 | 74.39 | 98.00 | 76.22 | 100.00 | 76.67 | 99.50 | 65.39 |
| | | GradNorm | 97.24 | 80.59 | 96.95 | 78.01 | 98.52 | 72.12 | 98.34 | 77.16 | 97.76 | 76.97 | 100.00 | 86.94 | 99.70 | 67.46 |
| | | Maha_Norm | 97.68 | 41.34 | 98.83 | 32.09 | 98.64 | 43.30 | 96.23 | 49.91 | 97.84 | 41.66 | 100.00 | 58.44 | 99.20 | 28.39 |
| | | Maha_UnNorm | 97.72 | 41.59 | 98.83 | 32.93 | 98.60 | 43.70 | 96.23 | 50.06 | 97.84 | 42.07 | 100.00 | 59.11 | 99.20 | 30.87 |
| | | KNN_10 | 66.89 | 87.91 | 68.58 | 86.90 | 77.61 | 82.31 | 76.58 | 85.39 | 72.41 | 85.63 | 75.58 | 89.45 | 86.40 | 84.23 |
| | | KNN_20 | 67.57 | 87.80 | 68.90 | 86.79 | 77.77 | 82.19 | 76.30 | 85.22 | 72.64 | 85.50 | 80.23 | 89.17 | 86.80 | 83.85 |
| | | KNN_50 | 67.97 | 87.58 | 69.67 | 86.62 | 78.01 | 81.98 | 76.66 | 84.85 | 73.08 | 85.27 | 80.23 | 88.63 | 87.20 | 83.21 |
| | | KNN_100 | 69.46 | 87.34 | 71.23 | 86.47 | 79.01 | 81.72 | 77.48 | 84.57 | 74.30 | 85.02 | 82.56 | 88.19 | 88.00 | 82.72 |
| | | KNN_200 | 70.86 | 87.01 | 72.20 | 86.19 | 79.73 | 81.31 | 78.29 | 84.19 | 75.27 | 84.68 | 82.56 | 87.58 | 88.50 | 82.05 |
| | | KNN_400 | 72.46 | 86.58 | 73.16 | 85.91 | 80.65 | 80.70 | 78.98 | 83.84 | 76.31 | 84.26 | 83.72 | 86.86 | 89.00 | 81.14 |
| | | **Pretrain on 40% IIT-CDIP⁻ (no finetune)** | | | | | | | | | | | | | | |
| | – | KNN_10 | 88.79 | 66.14 | 88.35 | 68.92 | 93.50 | 60.30 | 95.54 | 51.09 | 91.54 | 61.61 | 37.21 | 95.37 | 55.90 | 91.90 |
| | | KNN_20 | 89.59 | 65.07 | 89.80 | 67.61 | 93.89 | 59.10 | 95.58 | 50.17 | 92.21 | 60.49 | 46.51 | 94.41 | 61.50 | 91.00 |
| | | KNN_50 | 90.59 | 63.39 | 91.64 | 65.68 | 93.77 | 57.35 | 95.66 | 48.63 | 92.92 | 58.76 | 53.49 | 93.06 | 66.40 | 89.72 |
| | | KNN_100 | 91.19 | 61.79 | 92.37 | 63.90 | 93.66 | 55.78 | 95.62 | 47.42 | 93.21 | 57.22 | 65.12 | 91.99 | 68.30 | 88.72 |
| | | KNN_200 | 92.07 | 60.06 | 92.85 | 61.94 | 93.62 | 54.02 | 95.86 | 46.08 | 93.60 | 55.53 | 74.42 | 90.94 | 71.00 | 87.63 |
| | | KNN_400 | 92.39 | 58.83 | 93.41 | 60.48 | 94.09 | 52.79 | 96.14 | 45.47 | 94.01 | 54.39 | 77.91 | 89.99 | 72.30 | 86.59 |
| | | **Pretrain on 100% IIT-CDIP⁻ → Finetune on RVL-CDIP ID data** | | | | | | | | | | | | | | |
| RoBERTa_Base (100%) | 90.74 | MSP | 94.12 | 68.24 | 94.29 | 66.18 | 95.93 | 63.83 | 95.21 | 65.66 | 94.89 | 65.98 | 98.84 | 59.25 | 96.50 | 65.42 |
| | | MaxLogit | 97.24 | 78.15 | 97.19 | 80.27 | 98.36 | 72.16 | 98.38 | 75.82 | 97.79 | 76.60 | 100.00 | 73.28 | 99.30 | 75.58 |
| | | Energy | 97.32 | 78.13 | 97.51 | 80.26 | 98.64 | 72.12 | 98.70 | 75.78 | 98.04 | 76.57 | 100.00 | 73.27 | 99.50 | 75.52 |
| | | GradNorm | 97.16 | 80.07 | 97.39 | 77.86 | 98.40 | 71.83 | 98.05 | 79.08 | 97.75 | 77.21 | 100.00 | 86.32 | 99.40 | 73.52 |
| | | Maha_Norm | 98.60 | 38.30 | 98.43 | 31.77 | 98.52 | 39.68 | 97.52 | 44.21 | 98.27 | 38.49 | 98.84 | 53.25 | 99.40 | 33.41 |
| | | Maha_UnNorm | 98.60 | 37.83 | 98.55 | 31.90 | 98.64 | 39.00 | 97.65 | 43.11 | 98.36 | 37.98 | 98.84 | 53.82 | 99.40 | 34.34 |
| | | KNN_10 | 66.81 | 87.86 | 69.67 | 86.91 | 77.49 | 82.60 | 74.59 | 86.28 | 72.14 | 85.91 | 81.40 | 87.74 | 76.90 | 88.49 |
| | | KNN_20 | 66.73 | 87.75 | 70.31 | 86.78 | 77.89 | 82.51 | 75.28 | 86.13 | 72.55 | 85.79 | 81.40 | 87.43 | 77.70 | 88.39 |
| | | KNN_50 | 67.25 | 87.54 | 70.59 | 86.62 | 77.85 | 82.32 | 75.41 | 85.84 | 72.78 | 85.58 | 83.72 | 86.85 | 77.80 | 88.23 |
| | | KNN_100 | 68.13 | 87.34 | 71.47 | 86.39 | 78.05 | 82.08 | 76.14 | 85.60 | 73.45 | 85.35 | 83.72 | 86.39 | 78.50 | 88.21 |
| | | KNN_200 | 70.18 | 86.99 | 72.96 | 86.09 | 79.41 | 81.69 | 77.44 | 85.28 | 75.00 | 85.01 | 88.37 | 85.71 | 79.00 | 88.14 |
| | | KNN_400 | 71.66 | 86.54 | 73.68 | 85.77 | 80.61 | 81.09 | 77.52 | 84.99 | 75.87 | 84.60 | 88.37 | 84.87 | 79.20 | 87.99 |
| | | **Pretrain on 100% IIT-CDIP⁻ (no finetune)** | | | | | | | | | | | | | | |
| | – | KNN_10 | 87.95 | 66.44 | 84.49 | 72.34 | 95.01 | 58.47 | 96.23 | 49.07 | 90.92 | 61.58 | 31.40 | 96.19 | 41.60 | 94.78 |
| | | KNN_20 | 88.91 | 65.39 | 85.70 | 71.25 | 95.33 | 57.19 | 96.59 | 48.06 | 91.63 | 60.47 | 34.88 | 95.50 | 48.40 | 94.12 |
| | | KNN_50 | 90.59 | 63.69 | 87.14 | 69.45 | 95.53 | 54.93 | 97.08 | 46.26 | 92.58 | 58.58 | 43.02 | 94.51 | 55.20 | 93.05 |
| | | KNN_100 | 91.75 | 62.08 | 88.55 | 67.85 | 95.89 | 53.05 | 97.20 | 44.81 | 93.35 | 56.95 | 50.00 | 93.60 | 61.10 | 92.04 |
| | | KNN_200 | 91.91 | 60.13 | 89.35 | 66.23 | 96.17 | 50.78 | 97.40 | 42.85 | 93.71 | 55.00 | 53.49 | 92.75 | 64.30 | 91.08 |
| | | KNN_400 | 92.19 | 58.33 | 90.12 | 64.47 | 96.49 | 48.74 | 97.56 | 41.40 | 94.09 | 53.24 | 58.14 | 91.74 | 67.60 | 89.83 |

Table 5: OOD detection performance for document classification with different number of pretraining data from IIT-CDIP⁻ (remove *pseudo* OOD categories).

| | ID Acc | Method | OOD Dataset (In-Domain) | | | | | | | | | | OOD Dataset (Out-Domain) | | | |
|---|---|---|---|---|---|---|---|---|---|---|---|---|---|---|---|---|
| | | | Sci. Report | | Presentation | | Form | | Letter | | *Average* | | Sci. Poster | | Receipt | |
| | | | FPR95 | AUROC | FPR95 | AUROC | FPR95 | AUROC | FPR95 | AUROC | FPR95 | AUROC | FPR95 | AUROC | FPR95 | AUROC |
| **Pretrain on 10% IIT-CDIP⁻ → Finetune on RVL-CDIP ID data** | | | | | | | | | | | | | | | | |
| $\text{LayoutLMv1}_{\text{Base}}$ (10%) | 95.89 | MSP | 42.43 | 76.31 | 56.05 | 69.39 | 54.31 | 70.25 | 47.00 | 73.93 | 49.95 | 72.47 | 43.02 | 76.55 | 44.10 | 75.68 |
| | | MaxLogit | 41.91 | 91.27 | 55.04 | 89.33 | 54.19 | 85.20 | 44.97 | 90.93 | 49.03 | 89.18 | 38.37 | 94.27 | 41.30 | 91.38 |
| | | Energy | 41.83 | 91.29 | 54.92 | 89.35 | 54.11 | 85.22 | 45.01 | 90.97 | 48.97 | 89.21 | 38.37 | 94.29 | 41.10 | 91.42 |
| | | GradNorm | 39.15 | 91.80 | 54.04 | 86.93 | 51.88 | 86.05 | 42.49 | 91.65 | 46.89 | 89.11 | 38.37 | 91.79 | 41.40 | 91.82 |
| | | $\text{Maha}_{\text{Norm}}$ | 29.94 | 94.77 | 45.44 | 91.53 | 44.29 | 91.30 | 41.64 | 92.85 | 40.33 | 92.61 | 22.09 | 96.26 | 28.60 | 96.03 |
| | | $\text{Maha}_{\text{UnNorm}}$ | 40.11 | 85.82 | 52.87 | 78.32 | 55.11 | 78.51 | 62.01 | 71.56 | 52.52 | 78.55 | 25.58 | 90.88 | 40.50 | 87.49 |
| | | $\text{KNN}_{10}$ | 31.63 | 94.25 | 46.52 | 90.98 | 46.77 | 90.49 | 40.83 | 92.79 | 41.44 | 92.13 | 24.42 | 95.95 | 30.30 | 95.66 |
| | | $\text{KNN}_{20}$ | 32.03 | 94.11 | 46.65 | 90.89 | 47.01 | 90.32 | 41.60 | 92.63 | 41.82 | 91.99 | 26.74 | 95.76 | 31.80 | 95.44 |
| | | $\text{KNN}_{50}$ | 34.39 | 93.75 | 49.34 | 90.46 | 49.36 | 89.94 | 44.52 | 92.23 | 44.40 | 91.60 | 33.72 | 95.33 | 33.20 | 95.38 |
| | | $\text{KNN}_{100}$ | 36.15 | 93.47 | 51.27 | 90.19 | 51.36 | 89.65 | 46.63 | 91.99 | 46.35 | 91.32 | 33.72 | 95.10 | 35.10 | 95.16 |
| | | $\text{KNN}_{200}$ | 38.67 | 93.18 | 53.68 | 89.95 | 53.99 | 89.33 | 49.31 | 91.78 | 48.91 | 91.06 | 34.88 | 94.88 | 38.90 | 94.93 |
| | | $\text{KNN}_{400}$ | 40.11 | 92.87 | 54.92 | 89.75 | 55.55 | 88.98 | 50.81 | 91.59 | 50.35 | 90.80 | 34.88 | 94.64 | 40.20 | 94.69 |
| **Pretrain on 10% IIT-CDIP⁻ (no finetune)** | | | | | | | | | | | | | | | | |
| – | | $\text{KNN}_{10}$ | 90.95 | 72.30 | 94.66 | 65.49 | 90.94 | 72.38 | 94.40 | 67.32 | 92.74 | 69.37 | 48.84 | 91.56 | 56.00 | 75.08 |
| | | $\text{KNN}_{20}$ | 91.59 | 70.54 | 94.98 | 63.91 | 91.66 | 70.74 | 94.81 | 65.95 | 93.26 | 67.78 | 53.49 | 90.41 | 57.60 | 73.51 |
| | | $\text{KNN}_{50}$ | 93.07 | 67.76 | 95.54 | 61.24 | 92.78 | 68.27 | 95.25 | 64.01 | 94.16 | 65.32 | 55.81 | 88.37 | 58.50 | 71.06 |
| | | $\text{KNN}_{100}$ | 93.55 | 65.41 | 95.90 | 59.13 | 93.10 | 66.19 | 95.54 | 62.41 | 94.52 | 63.28 | 67.44 | 86.44 | 60.20 | 69.09 |
| | | $\text{KNN}_{200}$ | 94.44 | 62.66 | 96.14 | 56.66 | 93.74 | 63.76 | 95.78 | 60.49 | 95.02 | 60.89 | 75.58 | 84.12 | 63.00 | 66.97 |
| | | $\text{KNN}_{400}$ | 95.32 | 58.89 | 96.67 | 53.42 | 94.53 | 60.24 | 96.14 | 57.55 | 95.66 | 57.53 | 80.23 | 80.73 | 66.80 | 64.45 |
| **Pretrain on 20% IIT-CDIP⁻ → Finetune on RVL-CDIP ID data** | | | | | | | | | | | | | | | | |
| $\text{LayoutLMv1}_{\text{Base}}$ (20%) | 95.84 | MSP | 49.20 | 76.78 | 61.51 | 70.13 | 62.37 | 69.49 | 55.52 | 73.64 | 57.15 | 72.51 | 50.00 | 77.99 | 50.70 | 75.90 |
| | | MaxLogit | 41.03 | 91.57 | 54.00 | 88.45 | 56.42 | 85.70 | 47.00 | 90.19 | 49.61 | 88.98 | 38.37 | 93.62 | 41.80 | 90.56 |
| | | Energy | 40.95 | 91.60 | 53.76 | 88.47 | 56.19 | 85.72 | 46.79 | 90.22 | 49.42 | 89.00 | 38.37 | 93.65 | 41.70 | 90.59 |
| | | GradNorm | 37.15 | 91.89 | 54.16 | 84.99 | 53.03 | 86.28 | 43.95 | 90.94 | 47.07 | 88.52 | 40.70 | 90.41 | 42.40 | 90.91 |
| | | $\text{Maha}_{\text{Norm}}$ | 30.18 | 94.80 | 46.16 | 90.95 | 45.41 | 91.24 | 39.73 | 93.09 | 40.37 | 92.52 | 26.74 | 96.03 | 32.70 | 95.56 |
| | | $\text{Maha}_{\text{UnNorm}}$ | 45.20 | 83.07 | 52.11 | 77.99 | 54.03 | 79.46 | 51.58 | 81.19 | 50.73 | 80.43 | 20.93 | 92.15 | 31.70 | 92.64 |
| | | $\text{KNN}_{10}$ | 31.63 | 94.17 | 47.69 | 90.29 | 47.49 | 90.50 | 40.54 | 92.92 | 41.84 | 91.97 | 31.40 | 95.65 | 34.50 | 95.15 |
| | | $\text{KNN}_{20}$ | 32.55 | 94.03 | 47.89 | 90.22 | 48.32 | 90.34 | 40.91 | 92.76 | 42.42 | 91.84 | 33.72 | 95.45 | 35.40 | 94.97 |
| | | $\text{KNN}_{50}$ | 35.71 | 93.67 | 49.74 | 89.82 | 51.04 | 89.99 | 44.12 | 92.39 | 45.15 | 91.47 | 36.05 | 95.01 | 36.30 | 94.92 |
| | | $\text{KNN}_{100}$ | 36.75 | 93.38 | 50.30 | 89.60 | 51.68 | 89.71 | 44.97 | 92.17 | 45.92 | 91.22 | 36.05 | 94.73 | 36.50 | 94.71 |
| | | $\text{KNN}_{200}$ | 38.87 | 93.08 | 52.15 | 89.38 | 53.51 | 89.39 | 47.48 | 91.94 | 48.00 | 90.95 | 38.37 | 94.50 | 38.00 | 94.46 |
| | | $\text{KNN}_{400}$ | 40.87 | 92.76 | 53.48 | 89.21 | 55.07 | 89.05 | 48.94 | 91.73 | 49.59 | 90.69 | 39.53 | 94.23 | 40.20 | 94.18 |
| **Pretrain on 20% IIT-CDIP⁻ (no finetune)** | | | | | | | | | | | | | | | | |
| – | | $\text{KNN}_{10}$ | 90.39 | 75.25 | 79.59 | 79.43 | 93.14 | 72.41 | 97.12 | 66.99 | 90.06 | 73.52 | 50.00 | 91.36 | 24.70 | 96.34 |
| | | $\text{KNN}_{20}$ | 90.63 | 73.75 | 80.47 | 78.51 | 93.81 | 70.58 | 97.16 | 65.54 | 90.52 | 72.10 | 55.81 | 89.91 | 26.90 | 95.94 |
| | | $\text{KNN}_{50}$ | 91.67 | 71.19 | 82.56 | 76.90 | 94.45 | 67.82 | 97.36 | 62.98 | 91.51 | 69.72 | 67.44 | 87.29 | 29.10 | 95.31 |
| | | $\text{KNN}_{100}$ | 91.95 | 69.19 | 83.73 | 75.55 | 95.33 | 65.37 | 97.36 | 60.84 | 92.09 | 67.74 | 74.42 | 84.78 | 30.30 | 94.75 |
| | | $\text{KNN}_{200}$ | 92.75 | 66.89 | 85.05 | 74.05 | 95.53 | 62.48 | 97.28 | 58.40 | 92.65 | 65.46 | 76.74 | 81.65 | 33.00 | 94.08 |
| | | $\text{KNN}_{400}$ | 93.43 | 63.13 | 86.78 | 71.67 | 96.45 | 57.77 | 97.44 | 54.21 | 93.52 | 61.70 | 82.56 | 76.97 | 36.50 | 93.11 |
| **Pretrain on 40% IIT-CDIP⁻ → Finetune on RVL-CDIP ID data** | | | | | | | | | | | | | | | | |
| $\text{LayoutLMv1}_{\text{Base}}$ (40%) | 96.01 | MSP | 51.76 | 75.76 | 62.39 | 69.63 | 63.37 | 68.75 | 54.22 | 74.03 | 57.94 | 72.04 | 55.81 | 71.69 | 42.50 | 80.56 |
| | | MaxLogit | 42.03 | 91.29 | 54.24 | 89.47 | 57.30 | 84.44 | 45.66 | 90.02 | 49.81 | 88.80 | 52.33 | 93.08 | 33.00 | 92.89 |
| | | Energy | 41.87 | 91.31 | 54.20 | 89.49 | 57.26 | 84.47 | 45.50 | 90.05 | 49.71 | 88.83 | 52.33 | 93.13 | 32.50 | 92.92 |
| | | GradNorm | 38.19 | 91.66 | 53.64 | 86.85 | 55.03 | 85.63 | 43.18 | 91.45 | 47.51 | 88.90 | 52.33 | 92.39 | 34.40 | 92.95 |
| | | $\text{Maha}_{\text{Norm}}$ | 28.90 | 94.98 | 44.72 | 91.14 | 44.97 | 91.47 | 36.16 | 93.61 | 38.69 | 92.80 | 24.42 | 96.20 | 22.90 | 96.28 |
| | | $\text{Maha}_{\text{UnNorm}}$ | 39.83 | 87.21 | 51.91 | 78.81 | 53.15 | 83.13 | 46.83 | 85.38 | 47.93 | 83.63 | 34.88 | 88.86 | 26.00 | 92.78 |
| | | $\text{KNN}_{10}$ | 31.47 | 94.43 | 47.13 | 90.63 | 48.20 | 90.45 | 38.11 | 93.30 | 41.23 | 92.20 | 27.91 | 95.78 | 24.70 | 96.09 |
| | | $\text{KNN}_{20}$ | 32.59 | 94.29 | 47.61 | 90.55 | 49.60 | 90.27 | 39.25 | 93.14 | 42.26 | 92.06 | 32.56 | 95.60 | 25.50 | 95.95 |
| | | $\text{KNN}_{50}$ | 34.87 | 93.93 | 49.50 | 90.10 | 52.11 | 89.88 | 42.29 | 92.77 | 44.69 | 91.66 | 38.37 | 95.16 | 26.40 | 95.95 |
| | | $\text{KNN}_{100}$ | 36.55 | 93.65 | 50.38 | 89.82 | 53.55 | 89.57 | 43.71 | 92.51 | 46.05 | 91.39 | 43.02 | 94.89 | 27.70 | 95.77 |
| | | $\text{KNN}_{200}$ | 39.07 | 93.37 | 52.11 | 89.54 | 55.47 | 89.25 | 45.82 | 92.26 | 48.12 | 91.10 | 46.51 | 94.62 | 30.60 | 95.57 |
| | | $\text{KNN}_{400}$ | 41.47 | 93.07 | 53.92 | 89.25 | 57.38 | 88.85 | 47.77 | 92.02 | 50.14 | 90.80 | 50.00 | 94.40 | 33.90 | 95.34 |
| **Pretrain on 40% IIT-CDIP⁻ (no finetune)** | | | | | | | | | | | | | | | | |
| – | | $\text{KNN}_{10}$ | 87.07 | 80.44 | 71.76 | 83.72 | 86.75 | 82.31 | 96.10 | 76.36 | 85.42 | 80.71 | 75.58 | 84.96 | 5.90 | 98.24 |
| | | $\text{KNN}_{20}$ | 88.95 | 79.03 | 74.93 | 82.31 | 88.99 | 81.11 | 96.71 | 75.01 | 87.40 | 79.36 | 80.23 | 82.56 | 7.20 | 97.93 |
| | | $\text{KNN}_{50}$ | 91.47 | 77.23 | 80.39 | 79.90 | 91.78 | 79.75 | 97.40 | 72.60 | 90.26 | 77.37 | 87.21 | 78.19 | 9.00 | 97.92 |
| | | $\text{KNN}_{100}$ | 90.75 | 75.27 | 84.77 | 77.48 | 91.74 | 78.31 | 97.16 | 70.26 | 91.10 | 75.33 | 89.53 | 74.11 | 14.20 | 97.49 |
| | | $\text{KNN}_{200}$ | 88.71 | 72.34 | 89.19 | 74.45 | 89.19 | 76.20 | 96.88 | 67.80 | 90.99 | 72.57 | 91.86 | 69.01 | 23.00 | 96.74 |
| | | $\text{KNN}_{400}$ | 89.87 | 67.31 | 94.82 | 69.64 | 89.86 | 71.98 | 97.08 | 62.73 | 92.91 | 67.92 | 93.02 | 62.02 | 41.90 | 94.95 |
| **Pretrain on 100% IIT-CDIP⁻ → Finetune on RVL-CDIP ID data** | | | | | | | | | | | | | | | | |
| $\text{LayoutLMv1}_{\text{Base}}$ (100%) | 96.38 | MSP | 43.43 | 76.12 | 57.21 | 69.16 | 58.38 | 68.56 | 46.14 | 74.76 | 51.29 | 72.15 | 38.37 | 78.67 | 28.30 | 83.78 |
| | | MaxLogit | 35.19 | 91.29 | 50.22 | 88.98 | 53.19 | 84.54 | 39.98 | 90.71 | 44.64 | 88.88 | 24.42 | 96.39 | 21.40 | 95.57 |
| | | Energy | 35.23 | 91.32 | 50.22 | 89.00 | 53.19 | 84.55 | 39.98 | 90.73 | 44.65 | 88.90 | 24.42 | 96.44 | 21.40 | 95.58 |
| | | GradNorm | 30.30 | 92.54 | 48.61 | 88.18 | 48.96 | 86.58 | 36.16 | 92.63 | 41.01 | 89.98 | 19.77 | 96.71 | 19.20 | 96.35 |
| | | $\text{Maha}_{\text{Norm}}$ | 24.62 | 95.29 | 43.55 | 92.03 | 42.30 | 91.60 | 32.95 | 94.01 | 35.86 | 93.23 | 11.63 | 97.85 | 20.20 | 96.60 |
| | | $\text{Maha}_{\text{UnNorm}}$ | 35.43 | 88.79 | 47.13 | 81.21 | 50.32 | 82.13 | 43.71 | 86.20 | 44.15 | 84.58 | 13.95 | 97.15 | 58.50 | 82.43 |
| | | $\text{KNN}_{10}$ | 26.50 | 94.95 | 43.47 | 91.69 | 45.09 | 90.95 | 34.09 | 93.86 | 37.29 | 92.86 | 19.77 | 97.39 | 17.80 | 96.37 |
| | | $\text{KNN}_{20}$ | 27.22 | 94.83 | 44.07 | 91.58 | 45.41 | 90.79 | 34.62 | 93.71 | 37.83 | 92.73 | 19.77 | 97.22 | 18.40 | 96.26 |
| | | $\text{KNN}_{50}$ | 29.46 | 94.49 | 46.28 | 91.12 | 47.69 | 90.45 | 37.50 | 93.33 | 40.23 | 92.35 | 17.44 | 97.04 | 18.70 | 96.80 |
| | | $\text{KNN}_{100}$ | 32.15 | 94.26 | 48.17 | 90.85 | 50.64 | 90.21 | 40.38 | 93.12 | 42.83 | 92.11 | 19.77 | 96.88 | 20.70 | 96.74 |
| | | $\text{KNN}_{200}$ | 34.35 | 94.00 | 49.50 | 90.58 | 52.43 | 89.93 | 42.05 | 92.92 | 44.58 | 91.85 | 20.93 | 96.71 | 21.90 | 96.65 |
| | | $\text{KNN}_{400}$ | 35.75 | 93.68 | 50.30 | 90.15 | 53.79 | 89.52 | 42.98 | 92.79 | 45.70 | 91.54 | 23.26 | 96.54 | 23.30 | 96.53 |
| **Pretrain on 100% IIT-CDIP⁻ (no finetune)** | | | | | | | | | | | | | | | | |
| – | | $\text{KNN}_{10}$ | 78.74 | 81.67 | 74.45 | 80.86 | 80.53 | 83.71 | 95.01 | 77.33 | 82.18 | 80.89 | 38.37 | 94.62 | 17.70 | 96.12 |
| | | $\text{KNN}_{20}$ | 82.39 | 80.13 | 77.86 | 79.31 | 83.48 | 82.75 | 95.45 | 75.93 | 84.80 | 79.53 | 44.19 | 93.42 | 14.60 | 96.13 |
| | | $\text{KNN}_{50}$ | 86.03 | 77.65 | 82.80 | 76.60 | 86.91 | 81.30 | 96.10 | 73.07 | 87.96 | 77.16 | 54.65 | 91.09 | 9.60 | 97.21 |
| | | $\text{KNN}_{100}$ | 89.11 | 75.51 | 88.03 | 74.08 | 90.62 | 79.78 | 96.71 | 70.43 | 91.12 | 74.95 | 66.28 | 88.50 | 18.00 | 96.82 |
| | | $\text{KNN}_{200}$ | 91.47 | 72.38 | 93.01 | 70.49 | 92.22 | 77.38 | 96.55 | 66.66 | 93.31 | 71.73 | 75.58 | 84.60 | 34.50 | 95.79 |
| | | $\text{KNN}_{400}$ | 93.07 | 67.61 | 96.95 | 65.71 | 91.46 | 73.15 | 97.36 | 61.59 | 94.71 | 67.02 | 80.23 | 78.79 | 51.20 | 93.83 |

Table 6: OOD detection performance for document classification. Spatial-RoBERTa$_{Base}$ (Pre) denotes applying the spatial-aware adapter in the word embedding layer. Spatial-RoBERTa$_{Base}$ (Post) denots applying the spatial-aware adaptor at the output layer.

| | ID Acc | Method | OOD Dataset (In-Domain) | | | | | | | | | | OOD Dataset (Out-Domain) | | | |
| | | | Sci. Report | | Presentation | | Form | | Letter | | *Average* | | Sci. Poster | | Receipt | |
| | | | FPR95 | AUROC | FPR95 | AUROC | FPR95 | AUROC | FPR95 | AUROC | FPR95 | AUROC | FPR95 | AUROC | FPR95 | AUROC |
| **RoBERTa$_{Base}$** | | **Finetune on RVL-CDIP (ID)** | | | | | | | | | | | | | | |
| | 90.19 | MSP | 91.19 | 73.70 | 90.84 | 73.49 | 91.82 | 71.53 | 91.03 | 72.35 | 91.22 | 72.77 | 93.02 | 80.94 | 97.60 | 74.59 |
| | | MaxLogit | 96.88 | 79.04 | 96.87 | 79.38 | 98.04 | 75.85 | 98.54 | 77.45 | 97.58 | 77.93 | 100.00 | 82.76 | 99.40 | 79.99 |
| | | Energy | 97.48 | 78.96 | 97.23 | 79.31 | 98.40 | 75.71 | 99.07 | 77.25 | 98.04 | 77.81 | 100.00 | 82.71 | 99.20 | 80.06 |
| | | Maha$_{Norm}$ | 55.48 | 89.30 | 56.33 | 89.11 | 66.48 | 86.06 | 75.12 | 87.52 | 63.35 | 88.00 | 13.95 | 97.72 | 52.40 | 94.25 |
| | | Maha$_{UnNorm}$ | 52.04 | 85.14 | 56.81 | 82.19 | 56.26 | 83.13 | 72.44 | 76.56 | 59.39 | 81.76 | 5.81 | 98.60 | 52.30 | 90.33 |
| | | KNN$_{10}$ | 53.20 | 88.94 | 58.50 | 88.62 | 61.37 | 86.25 | 63.72 | 88.29 | 59.20 | 88.02 | 22.09 | 96.52 | 68.60 | 92.47 |
| | | KNN$_{20}$ | 53.44 | 88.81 | 58.90 | 88.50 | 61.65 | 86.07 | 63.60 | 88.15 | 59.40 | 87.88 | 27.91 | 96.38 | 71.70 | 92.02 |
| | | KNN$_{50}$ | 53.84 | 88.52 | 59.42 | 88.42 | 62.01 | 85.81 | 64.16 | 87.80 | 59.86 | 87.64 | 32.56 | 96.07 | 74.30 | 91.37 |
| | | KNN$_{100}$ | 55.56 | 88.10 | 60.67 | 88.20 | 63.69 | 85.41 | 64.77 | 87.42 | 61.17 | 87.28 | 34.88 | 95.67 | 76.50 | 90.81 |
| | | KNN$_{200}$ | 57.85 | 87.45 | 63.00 | 87.82 | 65.16 | 84.81 | 66.60 | 86.87 | 63.15 | 86.74 | 39.53 | 95.13 | 79.30 | 90.16 |
| | | KNN$_{400}$ | 59.73 | 86.63 | 63.80 | 87.35 | 66.52 | 83.97 | 67.41 | 86.29 | 64.37 | 86.06 | 41.86 | 94.43 | 80.20 | 89.33 |
| | | **No finetune** | | | | | | | | | | | | | | |
| | – | KNN$_{10}$ | 93.11 | 63.52 | 88.15 | 66.34 | 94.57 | 66.92 | 98.42 | 53.37 | 93.56 | 62.54 | 25.58 | 95.99 | 86.00 | 72.99 |
| | | KNN$_{20}$ | 92.99 | 63.18 | 88.39 | 65.78 | 94.57 | 66.08 | 98.42 | 52.10 | 93.59 | 61.78 | 26.74 | 95.71 | 87.30 | 70.44 |
| | | KNN$_{50}$ | 92.67 | 62.41 | 89.31 | 64.72 | 94.17 | 64.74 | 98.34 | 50.07 | 93.62 | 60.48 | 26.74 | 95.02 | 90.80 | 66.04 |
| | | KNN$_{100}$ | 92.67 | 61.57 | 89.59 | 63.57 | 94.01 | 63.45 | 98.17 | 48.33 | 93.61 | 59.23 | 29.07 | 94.34 | 92.80 | 61.62 |
| | | KNN$_{200}$ | 93.03 | 60.31 | 90.68 | 61.86 | 94.09 | 61.75 | 98.17 | 46.20 | 93.99 | 57.53 | 34.88 | 93.36 | 94.40 | 55.69 |
| | | KNN$_{400}$ | 93.92 | 58.73 | 91.84 | 59.75 | 94.37 | 59.69 | 98.21 | 43.90 | 94.58 | 55.52 | 41.86 | 92.09 | 95.70 | 49.01 |
| **Spatial-RoBERTa$_{Base}$ (Pre)** | | **Pretrain on IIT-CDIP → Finetune on RVL-CDIP (ID)** | | | | | | | | | | | | | | |
| | 97.11 | MSP | 46.80 | 74.52 | 54.64 | 70.58 | 56.26 | 69.72 | 54.30 | 70.74 | 53.00 | 71.39 | 44.19 | 75.79 | 57.20 | 69.23 |
| | | MaxLogit | 39.43 | 88.64 | 46.48 | 89.92 | 49.96 | 85.75 | 48.30 | 87.66 | 46.04 | 87.99 | 33.72 | 93.42 | 50.60 | 88.70 |
| | | Energy | 39.43 | 88.66 | 46.48 | 89.94 | 50.00 | 85.76 | 48.30 | 87.67 | 46.05 | 88.01 | 33.72 | 93.45 | 50.60 | 88.71 |
| | | Maha$_{Norm}$ | 98.88 | 25.24 | 98.63 | 28.53 | 98.68 | 31.70 | 98.01 | 33.95 | 98.55 | 29.86 | 95.35 | 29.12 | 98.20 | 24.56 |
| | | Maha$_{UnNorm}$ | 99.12 | 18.75 | 99.00 | 25.71 | 98.88 | 29.42 | 98.17 | 31.51 | 98.79 | 26.35 | 96.51 | 21.15 | 99.90 | 20.58 |
| | | KNN$_{10}$ | 31.91 | 94.41 | 42.19 | 92.65 | 46.65 | 89.31 | 42.09 | 92.65 | 40.71 | 92.26 | 10.47 | 97.45 | 52.10 | 92.93 |
| | | KNN$_{20}$ | 32.31 | 94.28 | 42.59 | 92.64 | 47.01 | 89.21 | 43.43 | 92.53 | 41.34 | 92.16 | 11.63 | 97.31 | 53.30 | 92.80 |
| | | KNN$_{50}$ | 34.39 | 93.99 | 43.83 | 92.36 | 49.04 | 88.93 | 45.41 | 92.19 | 43.17 | 91.87 | 12.79 | 97.01 | 53.10 | 92.51 |
| | | KNN$_{100}$ | 35.15 | 93.76 | 44.27 | 92.15 | 49.48 | 88.65 | 46.14 | 91.97 | 43.76 | 91.63 | 15.12 | 96.81 | 49.70 | 92.44 |
| | | KNN$_{200}$ | 36.07 | 93.48 | 45.04 | 91.84 | 49.96 | 88.26 | 47.04 | 91.66 | 44.53 | 91.31 | 17.44 | 96.59 | 49.90 | 92.22 |
| | | KNN$_{400}$ | 37.43 | 93.16 | 46.00 | 91.50 | 50.88 | 87.75 | 48.30 | 91.35 | 45.65 | 90.94 | 20.93 | 96.36 | 49.90 | 91.85 |
| | | **Pretrain on IIT-CDIP (no finetune)** | | | | | | | | | | | | | | |
| | – | KNN$_{10}$ | 78.82 | 78.92 | 79.99 | 73.89 | 77.69 | 81.32 | 91.48 | 76.52 | 82.00 | 77.66 | 10.47 | 98.08 | 87.30 | 80.89 |
| | | KNN$_{20}$ | 79.74 | 77.95 | 82.64 | 72.17 | 79.81 | 80.40 | 92.13 | 75.11 | 83.58 | 76.41 | 16.28 | 97.60 | 92.10 | 76.94 |
| | | KNN$_{50}$ | 80.42 | 76.87 | 85.13 | 69.62 | 82.12 | 78.93 | 92.98 | 73.01 | 85.16 | 74.61 | 22.09 | 96.66 | 95.20 | 70.53 |
| | | KNN$_{100}$ | 81.43 | 75.70 | 86.90 | 67.19 | 83.40 | 77.12 | 93.38 | 71.07 | 86.28 | 72.77 | 27.91 | 95.86 | 96.60 | 64.56 |
| | | KNN$_{200}$ | 84.51 | 73.69 | 89.27 | 63.93 | 85.51 | 74.38 | 94.07 | 68.53 | 88.34 | 70.13 | 53.49 | 94.43 | 98.10 | 57.57 |
| | | KNN$_{400}$ | 86.23 | 70.31 | 90.64 | 59.79 | 87.51 | 70.39 | 94.12 | 65.01 | 89.62 | 66.38 | 58.14 | 92.80 | 99.10 | 48.98 |
| **Spatial-RoBERTa$_{Base}$ (Post)** | | **Finetune on RVL-CDIP (ID)** | | | | | | | | | | | | | | |
| | 97.10 | MSP | 58.05 | 78.37 | 76.46 | 65.44 | 65.80 | 75.00 | 61.81 | 77.59 | 65.53 | 74.10 | 54.65 | 81.65 | 93.50 | 52.85 |
| | | MaxLogit | 49.20 | 89.82 | 72.36 | 80.28 | 57.82 | 87.28 | 52.52 | 90.04 | 57.98 | 86.86 | 34.88 | 94.88 | 91.60 | 73.37 |
| | | Energy | 47.56 | 89.87 | 71.96 | 80.30 | 56.58 | 87.32 | 51.18 | 90.10 | 56.82 | 86.90 | 34.88 | 95.04 | 91.30 | 73.39 |
| | | Maha$_{Norm}$ | 98.40 | 32.06 | 98.71 | 34.37 | 98.24 | 33.23 | 95.98 | 34.72 | 97.83 | 33.60 | 96.51 | 30.41 | 100.00 | 18.31 |
| | | Maha$_{UnNorm}$ | 98.44 | 30.88 | 98.79 | 33.59 | 98.20 | 31.69 | 95.94 | 32.94 | 97.84 | 32.28 | 96.51 | 29.45 | 100.00 | 16.88 |
| | | KNN$_{10}$ | 37.43 | 93.37 | 64.08 | 86.83 | 49.44 | 89.82 | 46.92 | 92.17 | 49.47 | 90.55 | 26.74 | 96.38 | 90.10 | 80.21 |
| | | KNN$_{20}$ | 38.27 | 93.25 | 65.33 | 86.52 | 50.80 | 89.66 | 48.09 | 91.99 | 50.62 | 90.35 | 26.74 | 96.23 | 91.20 | 79.57 |
| | | KNN$_{50}$ | 40.43 | 92.98 | 67.38 | 86.02 | 52.83 | 89.38 | 50.65 | 91.58 | 52.82 | 89.99 | 26.74 | 95.89 | 92.10 | 78.48 |
| | | KNN$_{100}$ | 41.99 | 92.77 | 67.94 | 85.62 | 53.87 | 89.17 | 51.22 | 91.33 | 53.76 | 89.72 | 29.07 | 95.67 | 92.60 | 77.68 |
| | | KNN$_{200}$ | 42.87 | 92.52 | 68.86 | 85.19 | 53.95 | 88.93 | 52.07 | 91.03 | 54.44 | 89.42 | 31.40 | 95.42 | 93.20 | 76.81 |
| | | KNN$_{400}$ | 44.04 | 92.26 | 69.75 | 84.75 | 54.91 | 88.49 | 53.17 | 90.71 | 55.47 | 89.10 | 32.56 | 95.15 | 93.60 | 75.81 |
| **Spatial-RoBERTa$_{Large}$ (Pre)** | | **Pretrain on IIT-CDIP → Finetune on RVL-CDIP (ID)** | | | | | | | | | | | | | | |
| | 97.37 | MSP | 62.37 | 67.82 | 71.27 | 63.36 | 72.87 | 62.54 | 70.25 | 63.84 | 69.19 | 64.39 | 76.74 | 60.61 | 67.00 | 65.48 |
| | | MaxLogit | 33.39 | 90.15 | 39.25 | 89.87 | 42.30 | 88.12 | 37.05 | 91.66 | 38.00 | 89.95 | 31.40 | 92.41 | 27.70 | 94.23 |
| | | Energy | 33.39 | 90.16 | 39.25 | 89.88 | 42.30 | 88.13 | 37.05 | 91.66 | 38.00 | 89.96 | 31.40 | 92.42 | 27.70 | 94.22 |
| | | Maha$_{Norm}$ | 86.59 | 43.75 | 95.18 | 33.10 | 95.49 | 40.19 | 94.64 | 43.55 | 92.98 | 40.15 | 94.19 | 24.87 | 94.00 | 34.27 |
| | | Maha$_{UnNorm}$ | 86.59 | 43.74 | 95.14 | 33.13 | 95.57 | 40.18 | 94.72 | 43.55 | 93.00 | 40.15 | 94.19 | 24.95 | 94.00 | 34.33 |
| | | KNN$_{10}$ | 28.18 | 94.47 | 42.43 | 93.01 | 37.43 | 91.74 | 31.13 | 94.72 | 34.79 | 93.49 | 25.58 | 96.24 | 18.60 | 96.28 |
| | | KNN$_{20}$ | 28.78 | 94.32 | 42.43 | 92.90 | 38.07 | 91.58 | 32.02 | 94.55 | 35.33 | 93.34 | 25.58 | 96.02 | 18.60 | 96.33 |
| | | KNN$_{50}$ | 30.22 | 93.95 | 43.71 | 92.69 | 40.06 | 91.26 | 34.54 | 94.10 | 37.13 | 93.00 | 26.74 | 95.52 | 21.40 | 96.14 |
| | | KNN$_{100}$ | 30.86 | 93.71 | 44.11 | 92.56 | 40.66 | 91.05 | 35.47 | 93.88 | 37.78 | 92.80 | 26.74 | 95.22 | 21.70 | 96.11 |
| | | KNN$_{200}$ | 31.43 | 93.48 | 44.31 | 92.49 | 41.50 | 90.84 | 36.36 | 93.66 | 38.40 | 92.62 | 26.74 | 94.96 | 22.40 | 96.08 |
| | | KNN$_{400}$ | 31.87 | 93.27 | 44.36 | 92.51 | 42.10 | 90.67 | 37.30 | 93.43 | 38.91 | 92.47 | 29.07 | 94.70 | 22.70 | 96.07 |
| | | **Pretrain on IIT-CDIP (no finetune)** | | | | | | | | | | | | | | |
| | – | KNN$_{10}$ | 68.49 | 80.43 | 88.23 | 69.83 | 71.75 | 83.11 | 88.11 | 73.32 | 79.14 | 76.67 | 75.58 | 84.36 | 49.80 | 92.02 |
| | | KNN$_{20}$ | 71.74 | 78.77 | 90.24 | 67.41 | 75.66 | 81.38 | 89.04 | 71.14 | 81.67 | 74.68 | 81.40 | 81.55 | 62.20 | 90.29 |
| | | KNN$_{50}$ | 75.46 | 76.49 | 92.81 | 63.82 | 80.17 | 78.72 | 90.42 | 67.84 | 84.72 | 71.72 | 82.56 | 77.15 | 78.20 | 87.49 |
| | | KNN$_{100}$ | 77.62 | 74.59 | 94.42 | 60.94 | 83.16 | 76.25 | 91.80 | 65.30 | 86.75 | 69.27 | 84.88 | 73.34 | 88.20 | 84.96 |
| | | KNN$_{200}$ | 81.43 | 71.91 | 96.06 | 57.24 | 87.67 | 73.01 | 93.51 | 61.99 | 89.67 | 66.04 | 87.21 | 68.70 | 94.90 | 81.47 |
| | | KNN$_{400}$ | 87.15 | 69.12 | 97.71 | 54.08 | 92.06 | 69.76 | 95.29 | 59.03 | 93.05 | 63.00 | 89.53 | 64.54 | 97.80 | 78.05 |

Table 7: OOD detection performance for document classification with the different number of pretraining data from IIT-CDIP.

| | ID Acc | Method | Sci. Report FPR95 | Sci. Report AUROC | Presentation FPR95 | Presentation AUROC | Form FPR95 | Form AUROC | Letter FPR95 | Letter AUROC | Average FPR95 | Average AUROC | Sci. Poster FPR95 | Sci. Poster AUROC | Receipt FPR95 | Receipt AUROC |
|---|---|---|---|---|---|---|---|---|---|---|---|---|---|---|---|---|
| | | | | | | | OOD Dataset (In-Domain) | | | | | | OOD Dataset (Out-Domain) | | | |
| **Pretrain on 10% IIT-CDIP→ Finetune on RVL-CDIP (ID)** | | | | | | | | | | | | | | | | |
| $\text{ViT}_{\text{Base}}$ (10%) | 94.89 | MSP | 55.80 | 88.37 | 48.61 | 91.38 | 63.93 | 83.83 | 55.52 | 88.55 | 55.96 | 88.03 | 52.05 | 89.60 | 34.10 | 95.04 |
| | | MaxLogit | 50.36 | 91.51 | 37.77 | 94.30 | 62.37 | 87.97 | 53.69 | 92.11 | 51.05 | 91.47 | 38.36 | 94.24 | 28.60 | 96.06 |
| | | Energy | 50.56 | 91.48 | 37.08 | 94.33 | 63.49 | 87.89 | 55.19 | 92.00 | 51.58 | 91.42 | 38.36 | 94.29 | 29.40 | 95.96 |
| | | GradNorm | 55.56 | 79.75 | 45.96 | 84.79 | 66.92 | 74.07 | 58.44 | 81.07 | 56.72 | 79.92 | 47.95 | 82.04 | 34.90 | 91.68 |
| | | $\text{Maha}_{\text{Norm}}$ | 96.92 | 64.51 | 97.35 | 62.51 | 94.29 | 66.66 | 91.88 | 70.25 | 95.11 | 65.98 | 98.63 | 54.58 | 98.30 | 64.77 |
| | | $\text{Maha}_{\text{UnNorm}}$ | 96.92 | 64.51 | 97.35 | 62.51 | 94.29 | 66.67 | 91.88 | 70.25 | 95.11 | 65.98 | 98.63 | 54.59 | 98.30 | 64.77 |
| | | $\text{KNN}_{10}$ | 50.40 | 92.60 | 43.51 | 93.92 | 51.60 | 90.54 | 74.47 | 88.87 | 55.00 | 91.48 | 20.55 | 97.19 | 9.20 | 98.21 |
| | | $\text{KNN}_{20}$ | 49.80 | 92.70 | 40.38 | 94.43 | 53.39 | 90.26 | 74.72 | 88.77 | 54.57 | 91.54 | 23.29 | 96.98 | 10.40 | 98.05 |
| | | $\text{KNN}_{50}$ | 46.72 | 92.89 | 34.27 | 95.24 | 56.07 | 89.92 | 74.55 | 88.45 | 52.90 | 91.62 | 27.40 | 96.56 | 12.80 | 97.80 |
| | | $\text{KNN}_{100}$ | 45.48 | 92.89 | 29.33 | 95.67 | 57.62 | 89.56 | 75.04 | 88.25 | 51.87 | 91.59 | 30.14 | 96.21 | 15.00 | 97.57 |
| | | $\text{KNN}_{200}$ | 44.48 | 92.86 | 24.35 | 96.01 | 59.18 | 89.22 | 75.37 | 88.20 | 50.84 | 91.57 | 34.25 | 95.93 | 16.90 | 97.36 |
| | | $\text{KNN}_{400}$ | 44.32 | 92.76 | 22.98 | 96.17 | 60.81 | 88.82 | 75.24 | 88.09 | 50.84 | 91.46 | 35.62 | 95.67 | 19.30 | 97.19 |
| **Pretrain on IIT-CDIP (no finetune)** | | | | | | | | | | | | | | | | |
| – | | $\text{KNN}_{10}$ | 98.92 | 43.08 | 97.67 | 49.00 | 99.52 | 54.41 | 99.35 | 40.26 | 98.86 | 46.69 | 93.15 | 92.51 | 6.90 | 98.06 |
| | | $\text{KNN}_{20}$ | 98.88 | 42.47 | 97.75 | 48.57 | 99.52 | 53.75 | 99.35 | 39.56 | 98.88 | 46.09 | 94.52 | 92.24 | 8.60 | 97.91 |
| | | $\text{KNN}_{50}$ | 98.80 | 41.70 | 97.83 | 48.04 | 99.52 | 52.91 | 99.35 | 38.62 | 98.88 | 45.32 | 95.89 | 91.80 | 10.60 | 97.66 |
| | | $\text{KNN}_{100}$ | 98.76 | 41.20 | 97.79 | 47.70 | 99.48 | 52.32 | 99.35 | 38.01 | 98.84 | 44.81 | 98.63 | 91.31 | 14.50 | 97.41 |
| | | $\text{KNN}_{200}$ | 98.48 | 40.72 | 97.63 | 47.38 | 99.44 | 51.74 | 99.31 | 37.34 | 98.72 | 44.30 | 98.63 | 90.83 | 20.60 | 97.01 |
| | | $\text{KNN}_{400}$ | 98.28 | 40.11 | 97.59 | 47.02 | 99.48 | 51.03 | 99.31 | 36.42 | 98.66 | 43.64 | 98.63 | 90.38 | 25.40 | 96.51 |
| **Pretrain on 20% IIT-CDIP→ Finetune on RVL-CDIP (ID)** | | | | | | | | | | | | | | | | |
| $\text{ViT}_{\text{Base}}$ (20%) | 94.62 | MSP | 54.36 | 89.01 | 51.63 | 91.31 | 64.57 | 85.23 | 60.51 | 88.67 | 57.77 | 88.56 | 60.27 | 89.34 | 44.20 | 93.73 |
| | | MaxLogit | 44.32 | 92.16 | 38.21 | 94.18 | 64.92 | 87.63 | 58.56 | 91.33 | 51.50 | 91.32 | 45.21 | 92.63 | 39.70 | 94.36 |
| | | Energy | 44.36 | 92.17 | 37.89 | 94.24 | 66.56 | 87.51 | 60.39 | 91.22 | 52.30 | 91.28 | 46.58 | 92.62 | 41.50 | 94.18 |
| | | GradNorm | 90.51 | 54.92 | 92.04 | 51.67 | 94.29 | 45.41 | 98.13 | 32.36 | 93.74 | 46.09 | 95.89 | 40.44 | 89.70 | 59.01 |
| | | $\text{Maha}_{\text{Norm}}$ | 96.88 | 48.94 | 97.23 | 45.89 | 96.81 | 51.12 | 95.98 | 53.68 | 96.72 | 49.91 | 97.26 | 58.22 | 99.10 | 39.24 |
| | | $\text{Maha}_{\text{UnNorm}}$ | 96.88 | 48.94 | 97.23 | 45.89 | 96.81 | 51.12 | 95.98 | 53.68 | 96.72 | 49.91 | 97.26 | 58.22 | 99.10 | 39.24 |
| | | $\text{KNN}_{10}$ | 52.20 | 92.58 | 45.84 | 93.73 | 53.79 | 90.75 | 77.84 | 87.02 | 57.42 | 91.02 | 17.81 | 97.33 | 16.90 | 97.40 |
| | | $\text{KNN}_{20}$ | 51.60 | 92.66 | 43.55 | 94.15 | 55.63 | 90.46 | 78.04 | 86.79 | 57.20 | 91.02 | 19.18 | 97.06 | 19.40 | 97.11 |
| | | $\text{KNN}_{50}$ | 50.12 | 92.86 | 39.98 | 94.82 | 58.02 | 90.18 | 78.77 | 86.54 | 56.72 | 91.10 | 19.18 | 96.63 | 23.10 | 96.68 |
| | | $\text{KNN}_{100}$ | 48.04 | 92.91 | 34.75 | 95.28 | 60.38 | 89.88 | 78.98 | 86.42 | 55.54 | 91.12 | 20.55 | 96.27 | 26.20 | 96.35 |
| | | $\text{KNN}_{200}$ | 47.80 | 92.94 | 28.65 | 95.64 | 61.57 | 89.59 | 78.69 | 86.48 | 54.18 | 91.16 | 26.03 | 95.95 | 28.00 | 96.11 |
| | | $\text{KNN}_{400}$ | 46.88 | 92.92 | 26.64 | 95.79 | 62.81 | 89.30 | 77.60 | 86.50 | 53.48 | 91.13 | 32.88 | 95.67 | 28.70 | 96.16 |
| **Pretrain on IIT-CDIP (no finetune)** | | | | | | | | | | | | | | | | |
| – | | $\text{KNN}_{10}$ | 98.16 | 41.13 | 97.51 | 47.12 | 99.48 | 53.05 | 99.31 | 38.79 | 98.62 | 45.02 | 94.52 | 91.80 | 8.00 | 97.41 |
| | | $\text{KNN}_{20}$ | 98.12 | 40.71 | 97.51 | 46.79 | 99.48 | 52.52 | 99.31 | 38.31 | 98.60 | 44.58 | 94.52 | 91.48 | 8.70 | 97.25 |
| | | $\text{KNN}_{50}$ | 98.04 | 40.10 | 97.55 | 46.31 | 99.48 | 51.84 | 99.39 | 37.63 | 98.62 | 43.97 | 95.89 | 91.01 | 11.50 | 96.99 |
| | | $\text{KNN}_{100}$ | 98.00 | 39.74 | 97.55 | 45.98 | 99.48 | 51.14 | 99.39 | 37.26 | 98.60 | 43.58 | 97.26 | 90.55 | 14.60 | 96.70 |
| | | $\text{KNN}_{200}$ | 98.00 | 39.42 | 97.51 | 45.67 | 99.48 | 50.82 | 99.35 | 36.91 | 98.58 | 43.20 | 97.26 | 90.06 | 20.40 | 96.32 |
| | | $\text{KNN}_{400}$ | 97.92 | 39.16 | 97.55 | 45.37 | 99.44 | 50.32 | 99.31 | 36.56 | 98.56 | 42.85 | 98.63 | 89.62 | 31.90 | 95.97 |
| **Pretrain on 40% IIT-CDIP→ Finetune on RVL-CDIP (ID)** | | | | | | | | | | | | | | | | |
| $\text{ViT}_{\text{Base}}$ (40%) | 94.63 | MSP | 55.48 | 88.65 | 52.27 | 91.54 | 64.49 | 85.52 | 58.08 | 89.20 | 57.58 | 88.73 | 67.12 | 84.62 | 45.80 | 93.82 |
| | | MaxLogit | 47.12 | 91.74 | 40.06 | 94.09 | 61.05 | 88.68 | 56.57 | 92.01 | 51.20 | 91.63 | 69.86 | 89.81 | 32.90 | 95.46 |
| | | Energy | 47.12 | 91.73 | 39.94 | 94.10 | 62.33 | 88.62 | 58.60 | 91.88 | 52.00 | 91.58 | 69.86 | 89.65 | 32.70 | 95.44 |
| | | GradNorm | 47.00 | 85.76 | 41.90 | 89.64 | 60.69 | 81.37 | 53.73 | 87.06 | 50.83 | 85.96 | 64.38 | 81.12 | 34.00 | 92.93 |
| | | $\text{Maha}_{\text{Norm}}$ | 90.59 | 54.32 | 90.56 | 54.22 | 97.33 | 49.20 | 97.04 | 53.16 | 93.88 | 52.72 | 87.67 | 59.36 | 98.90 | 47.51 |
| | | $\text{Maha}_{\text{UnNorm}}$ | 90.59 | 54.32 | 90.56 | 54.22 | 97.33 | 49.20 | 97.04 | 53.16 | 93.88 | 52.72 | 87.67 | 59.36 | 98.90 | 47.51 |
| | | $\text{KNN}_{10}$ | 53.28 | 92.13 | 48.33 | 92.99 | 46.45 | 92.20 | 75.61 | 88.87 | 55.92 | 91.55 | 34.25 | 95.53 | 6.80 | 98.56 |
| | | $\text{KNN}_{20}$ | 52.76 | 92.24 | 45.88 | 93.57 | 48.12 | 91.95 | 74.84 | 88.75 | 55.40 | 91.63 | 32.88 | 95.21 | 7.80 | 98.36 |
| | | $\text{KNN}_{50}$ | 51.28 | 92.52 | 40.94 | 94.51 | 50.52 | 91.70 | 75.08 | 88.46 | 54.46 | 91.80 | 35.62 | 94.67 | 10.90 | 98.04 |
| | | $\text{KNN}_{100}$ | 50.32 | 92.62 | 36.16 | 95.12 | 53.35 | 91.36 | 75.93 | 88.24 | 53.94 | 91.84 | 39.73 | 94.25 | 13.60 | 97.76 |
| | | $\text{KNN}_{200}$ | 49.56 | 92.69 | 30.90 | 95.55 | 55.35 | 91.09 | 76.34 | 88.17 | 53.04 | 91.88 | 47.95 | 93.86 | 16.00 | 97.51 |
| | | $\text{KNN}_{400}$ | 47.56 | 92.69 | 27.56 | 95.76 | 56.62 | 90.84 | 76.14 | 88.05 | 51.97 | 91.84 | 54.79 | 93.53 | 17.20 | 97.35 |
| **Pretrain on IIT-CDIP (no finetune)** | | | | | | | | | | | | | | | | |
| – | | $\text{KNN}_{10}$ | 97.56 | 40.60 | 97.03 | 46.28 | 99.24 | 53.76 | 99.15 | 39.62 | 98.24 | 45.06 | 82.19 | 92.02 | 1.00 | 99.59 |
| | | $\text{KNN}_{20}$ | 97.56 | 40.00 | 96.95 | 45.86 | 99.24 | 53.18 | 99.15 | 39.12 | 98.22 | 44.54 | 82.19 | 91.63 | 1.00 | 99.55 |
| | | $\text{KNN}_{50}$ | 97.56 | 39.24 | 96.99 | 45.20 | 99.24 | 52.39 | 99.15 | 38.49 | 98.24 | 43.83 | 86.30 | 91.07 | 1.00 | 99.50 |
| | | $\text{KNN}_{100}$ | 97.60 | 38.78 | 97.03 | 44.79 | 99.24 | 51.76 | 99.15 | 38.15 | 98.26 | 43.37 | 90.41 | 90.67 | 1.20 | 99.45 |
| | | $\text{KNN}_{200}$ | 97.56 | 38.39 | 96.99 | 44.41 | 99.28 | 51.13 | 99.07 | 37.83 | 98.22 | 42.94 | 91.78 | 90.28 | 1.30 | 99.38 |
| | | $\text{KNN}_{400}$ | 97.56 | 38.06 | 97.11 | 44.07 | 99.28 | 50.50 | 99.07 | 37.51 | 98.26 | 42.54 | 91.78 | 89.84 | 1.50 | 99.28 |
| **Pretrain on 100% IIT-CDIP→ Finetune on RVL-CDIP (ID)** | | | | | | | | | | | | | | | | |
| $\text{ViT}_{\text{Base}}$ (100%) | 94.79 | MSP | 54.28 | 88.80 | 49.14 | 91.80 | 64.60 | 84.44 | 58.85 | 88.78 | 56.72 | 88.46 | 61.64 | 89.44 | 41.00 | 94.27 |
| | | MaxLogit | 44.96 | 92.13 | 38.01 | 94.52 | 63.97 | 87.97 | 56.49 | 91.81 | 50.86 | 91.61 | 68.49 | 90.65 | 34.60 | 95.26 |
| | | Energy | 45.72 | 92.11 | 38.01 | 94.55 | 65.84 | 87.86 | 57.91 | 91.70 | 51.87 | 91.56 | 72.60 | 90.41 | 34.80 | 95.14 |
| | | GradNorm | 48.72 | 84.21 | 44.36 | 87.50 | 63.49 | 78.07 | 56.25 | 84.79 | 53.20 | 83.64 | 60.27 | 82.96 | 35.60 | 91.24 |
| | | $\text{Maha}_{\text{Norm}}$ | 97.56 | 58.38 | 96.50 | 60.98 | 94.57 | 60.56 | 97.52 | 56.98 | 96.54 | 59.22 | 94.52 | 71.56 | 92.50 | 62.59 |
| | | $\text{Maha}_{\text{UnNorm}}$ | 97.56 | 58.38 | 96.50 | 60.98 | 94.53 | 60.56 | 97.52 | 56.98 | 96.53 | 59.22 | 94.52 | 71.56 | 92.50 | 62.59 |
| | | $\text{KNN}_{10}$ | 45.16 | 93.14 | 39.13 | 94.62 | 51.68 | 90.85 | 73.58 | 88.81 | 52.39 | 91.86 | 50.68 | 93.09 | 10.40 | 98.04 |
| | | $\text{KNN}_{20}$ | 44.88 | 93.14 | 36.64 | 95.04 | 53.35 | 90.59 | 74.27 | 88.67 | 52.28 | 91.86 | 50.68 | 92.67 | 12.00 | 97.81 |
| | | $\text{KNN}_{50}$ | 43.67 | 93.19 | 31.18 | 95.60 | 56.74 | 90.29 | 75.28 | 88.49 | 51.72 | 91.89 | 57.53 | 92.23 | 15.60 | 97.45 |
| | | $\text{KNN}_{100}$ | 43.63 | 93.15 | 27.52 | 95.94 | 58.74 | 90.02 | 76.18 | 88.38 | 51.52 | 91.87 | 61.64 | 92.01 | 18.90 | 97.18 |
| | | $\text{KNN}_{200}$ | 44.16 | 93.05 | 25.23 | 96.11 | 60.57 | 89.72 | 76.06 | 88.33 | 51.50 | 91.80 | 61.64 | 91.74 | 20.90 | 96.95 |
| | | $\text{KNN}_{400}$ | 44.56 | 92.92 | 25.27 | 96.12 | 61.69 | 89.39 | 76.42 | 88.23 | 51.98 | 91.66 | 61.64 | 91.33 | 22.00 | 96.81 |
| **Pretrain on IIT-CDIP (no finetune)** | | | | | | | | | | | | | | | | |
| – | | $\text{KNN}_{10}$ | 97.04 | 42.35 | 93.97 | 50.17 | 97.41 | 52.68 | 98.01 | 43.19 | 96.61 | 47.10 | 12.33 | 97.47 | 3.10 | 98.38 |
| | | $\text{KNN}_{20}$ | 97.16 | 41.99 | 94.01 | 49.96 | 97.81 | 52.01 | 98.09 | 42.73 | 96.77 | 46.67 | 15.07 | 96.95 | 3.00 | 98.31 |
| | | $\text{KNN}_{50}$ | 96.96 | 41.62 | 94.34 | 49.56 | 98.00 | 51.20 | 98.05 | 42.24 | 96.84 | 46.16 | 21.92 | 96.08 | 2.70 | 98.18 |
| | | $\text{KNN}_{100}$ | 97.00 | 41.48 | 94.90 | 49.31 | 98.12 | 50.65 | 98.13 | 42.03 | 97.04 | 45.87 | 36.99 | 95.29 | 2.30 | 98.27 |
| | | $\text{KNN}_{200}$ | 97.12 | 41.31 | 95.30 | 49.01 | 98.04 | 50.06 | 98.46 | 41.87 | 97.23 | 45.56 | 52.05 | 94.41 | 1.30 | 98.42 |
| | | $\text{KNN}_{400}$ | 97.16 | 41.17 | 95.62 | 48.60 | 98.28 | 49.42 | 98.54 | 41.75 | 97.40 | 45.24 | 60.27 | 93.33 | 1.10 | 98.53 |

Table 8: OOD detection performance for document classification. Longformer$_{4096}$ denotes the original model adopted from the Huggingface model hub. Longformer$_{4096}$ (+) denotes the additional pretraining on IIT-CDIP.

| | ID Acc | Method | OOD Dataset (In-Domain) | | | | | | | | | | OOD Dataset (Out-Domain) | | | |
| | | | Sci. Report | | Presentation | | Form | | Letter | | Average | | Sci. Poster | | Receipt | |
| | | | FPR95 | AUROC | FPR95 | AUROC | FPR95 | AUROC | FPR95 | AUROC | FPR95 | AUROC | FPR95 | AUROC | FPR95 | AUROC |
|---|---|---|---|---|---|---|---|---|---|---|---|---|---|---|---|---|
| **Longformer$_{4096}$** | | **Finetune on RVL-CDIP (ID)** | | | | | | | | | | | | | | |
| | 90.71 | MSP | 95.00 | 64.32 | 95.62 | 62.17 | 95.89 | 60.53 | 93.95 | 66.89 | 95.12 | 63.48 | 88.37 | 77.50 | 98.60 | 54.72 |
| | | MaxLogit | 97.12 | 72.84 | 97.07 | 75.22 | 98.24 | 70.39 | 95.82 | 77.57 | 97.06 | 74.00 | 90.70 | 86.62 | 99.60 | 68.10 |
| | | Energy | 97.48 | 72.82 | 97.35 | 75.21 | 98.36 | 70.37 | 96.59 | 77.56 | 97.44 | 73.99 | 91.86 | 86.63 | 99.80 | 68.08 |
| | | Maha$_{Norm}$ | 58.21 | 88.86 | 64.93 | 87.16 | 69.19 | 84.31 | 57.55 | 89.54 | 62.47 | 87.47 | 19.77 | 97.02 | 78.50 | 87.44 |
| | | Maha$_{UnNorm}$ | 61.21 | 85.11 | 67.01 | 81.93 | 71.99 | 78.19 | 63.19 | 81.94 | 65.85 | 81.79 | 24.42 | 94.13 | 76.80 | 86.47 |
| | | KNN$_{10}$ | 58.45 | 88.21 | 65.65 | 86.88 | 67.80 | 83.99 | 56.78 | 89.53 | 62.17 | 87.15 | 27.91 | 96.01 | 82.10 | 86.31 |
| | | KNN$_{20}$ | 58.97 | 88.04 | 65.57 | 86.60 | 68.12 | 83.80 | 57.35 | 89.34 | 62.50 | 86.94 | 29.07 | 95.82 | 82.60 | 85.93 |
| | | KNN$_{50}$ | 60.25 | 87.64 | 66.57 | 86.25 | 68.91 | 83.41 | 58.81 | 88.96 | 63.64 | 86.56 | 30.23 | 95.46 | 82.70 | 85.27 |
| | | KNN$_{100}$ | 61.97 | 87.19 | 68.14 | 85.81 | 70.15 | 82.95 | 60.47 | 88.60 | 65.18 | 86.14 | 34.88 | 95.04 | 82.80 | 84.75 |
| | | KNN$_{200}$ | 64.29 | 86.71 | 69.43 | 85.40 | 71.79 | 82.42 | 62.74 | 88.31 | 67.06 | 85.71 | 43.02 | 94.63 | 83.60 | 84.27 |
| | | KNN$_{400}$ | 66.33 | 86.11 | 71.03 | 84.87 | 73.82 | 81.64 | 64.94 | 88.07 | 69.03 | 85.17 | 45.35 | 94.27 | 84.20 | 83.50 |
| | **No finetune** | | | | | | | | | | | | | | | |
| | – | KNN$_{10}$ | 98.04 | 55.45 | 97.63 | 59.97 | 98.76 | 51.75 | 98.13 | 53.16 | 98.14 | 55.08 | 70.93 | 88.69 | 100.00 | 64.97 |
| | | KNN$_{20}$ | 98.12 | 55.19 | 97.67 | 59.64 | 98.80 | 51.27 | 98.17 | 52.71 | 98.19 | 54.70 | 70.93 | 88.51 | 100.00 | 64.08 |
| | | KNN$_{50}$ | 98.00 | 54.82 | 97.63 | 59.13 | 98.80 | 50.57 | 98.30 | 52.07 | 98.18 | 54.15 | 73.26 | 88.29 | 100.00 | 62.82 |
| | | KNN$_{100}$ | 97.92 | 54.48 | 97.67 | 58.62 | 98.84 | 50.00 | 98.34 | 51.62 | 98.19 | 53.68 | 74.42 | 88.14 | 100.00 | 61.70 |
| | | KNN$_{200}$ | 97.96 | 53.92 | 97.75 | 57.85 | 98.92 | 49.30 | 98.38 | 51.05 | 98.25 | 53.03 | 74.42 | 87.99 | 100.00 | 60.18 |
| | | KNN$_{400}$ | 97.84 | 53.04 | 97.71 | 56.77 | 98.92 | 48.33 | 98.42 | 50.27 | 98.22 | 52.10 | 70.93 | 87.86 | 100.00 | 57.80 |
| **Longformer$_{4096}$ (+)** | | **Pretrain on IIT-CDIP → Finetune on RVL-CDIP (ID)** | | | | | | | | | | | | | | |
| | 91.13 | MSP | 95.20 | 64.08 | 95.62 | 61.38 | 96.05 | 59.47 | 94.48 | 63.13 | 95.34 | 62.02 | 90.70 | 67.26 | 98.00 | 55.52 |
| | | MaxLogit | 96.96 | 75.41 | 96.54 | 76.03 | 97.89 | 70.15 | 96.71 | 74.56 | 97.02 | 74.04 | 100.00 | 78.65 | 99.70 | 72.88 |
| | | Energy | 97.28 | 75.40 | 96.54 | 76.03 | 97.28 | 70.14 | 97.16 | 74.55 | 97.32 | 74.03 | 100.00 | 78.59 | 99.70 | 72.86 |
| | | Maha$_{Norm}$ | 98.92 | 42.39 | 97.83 | 48.23 | 99.40 | 49.47 | 97.40 | 52.81 | 98.39 | 48.22 | 100.00 | 40.91 | 99.70 | 65.88 |
| | | Maha$_{UnNorm}$ | 99.20 | 37.22 | 97.91 | 48.12 | 99.44 | 43.39 | 97.93 | 45.09 | 98.62 | 43.46 | 100.00 | 40.00 | 99.70 | 68.48 |
| | | KNN$_{10}$ | 58.73 | 89.25 | 66.21 | 87.57 | 72.03 | 83.76 | 63.68 | 88.72 | 65.16 | 87.32 | 48.84 | 94.78 | 86.40 | 87.84 |
| | | KNN$_{20}$ | 58.61 | 89.18 | 65.97 | 87.45 | 71.67 | 83.69 | 63.39 | 88.61 | 64.91 | 87.23 | 48.84 | 94.62 | 85.30 | 87.70 |
| | | KNN$_{50}$ | 61.17 | 88.96 | 66.97 | 87.29 | 72.83 | 83.47 | 65.83 | 88.33 | 66.70 | 87.01 | 55.81 | 94.25 | 85.20 | 87.39 |
| | | KNN$_{100}$ | 61.73 | 88.79 | 66.93 | 87.11 | 73.30 | 83.24 | 66.15 | 88.15 | 67.03 | 86.82 | 55.81 | 94.00 | 84.70 | 87.21 |
| | | KNN$_{200}$ | 62.89 | 88.60 | 67.34 | 86.94 | 73.74 | 82.93 | 66.96 | 87.99 | 67.73 | 86.62 | 59.30 | 93.77 | 84.80 | 87.00 |
| | | KNN$_{400}$ | 63.41 | 88.39 | 67.38 | 86.83 | 74.14 | 82.53 | 67.09 | 87.82 | 68.00 | 86.39 | 59.30 | 93.57 | 83.90 | 86.71 |
| | **Pretrain on IIT-CDIP (no finetune)** | | | | | | | | | | | | | | | |
| | – | KNN$_{10}$ | 95.48 | 61.40 | 98.07 | 53.66 | 97.73 | 55.55 | 98.66 | 48.70 | 97.49 | 54.83 | 81.40 | 91.12 | 97.40 | 46.27 |
| | | KNN$_{20}$ | 95.56 | 60.92 | 97.95 | 52.95 | 97.49 | 54.97 | 98.50 | 48.21 | 97.38 | 54.26 | 84.88 | 90.62 | 97.50 | 45.55 |
| | | KNN$_{50}$ | 95.60 | 59.94 | 97.95 | 51.77 | 97.41 | 53.97 | 98.62 | 47.29 | 97.40 | 53.24 | 87.21 | 89.95 | 98.20 | 44.18 |
| | | KNN$_{100}$ | 95.60 | 59.04 | 97.99 | 50.74 | 97.21 | 52.99 | 98.58 | 46.51 | 97.34 | 52.32 | 88.37 | 89.52 | 98.50 | 43.09 |
| | | KNN$_{200}$ | 95.68 | 58.13 | 98.03 | 49.68 | 97.45 | 52.01 | 98.58 | 45.79 | 97.44 | 51.40 | 89.53 | 88.84 | 98.50 | 41.98 |
| | | KNN$_{400}$ | 96.00 | 57.52 | 98.15 | 48.88 | 97.65 | 51.31 | 98.70 | 45.45 | 97.62 | 50.79 | 91.86 | 88.46 | 98.60 | 41.02 |

Table 9: OOD detection performance for document classification. All models are pretrained on ImageNet.

| | ID Acc | Method | OOD Dataset (In-Domain) | | | | | | | | | | OOD Dataset (Out-Domain) | | | |
|---|---|---|---|---|---|---|---|---|---|---|---|---|---|---|---|---|
| | | | Sci. Report | | Presentation | | Form | | Letter | | *Average* | | Sci. Poster | | Receipt | |
| | | | FPR95 | AUROC | FPR95 | AUROC | FPR95 | AUROC | FPR95 | AUROC | FPR95 | AUROC | FPR95 | AUROC | FPR95 | AUROC |
| **ResNet-50** | | **Pretrain on ImageNet→ Finetune on RVL-CDIP (ID)** | | | | | | | | | | | | | | |
| | | MSP | 64.49 | 87.87 | 55.89 | 90.94 | 66.60 | 87.31 | 77.88 | 80.87 | 66.22 | 86.75 | 51.16 | 92.76 | 63.10 | 90.36 |
| | | MaxLogit | 64.89 | 88.59 | 47.97 | 92.81 | 65.40 | 87.52 | 77.56 | 81.87 | 63.96 | 87.70 | 41.86 | 94.62 | 54.00 | 93.29 |
| | | Energy | 67.09 | 88.30 | 47.81 | 92.86 | 66.68 | 87.24 | 78.53 | 81.75 | 65.03 | 87.54 | 39.53 | 94.73 | 48.50 | 93.68 |
| | | Maha$_{Norm}$ | 77.78 | 83.36 | 67.66 | 86.40 | 84.48 | 81.43 | 95.01 | 72.93 | 81.23 | 81.03 | 1.16 | 99.75 | 0.00 | 99.95 |
| | | Maha$_{UnNorm}$ | 97.16 | 39.28 | 93.09 | 43.28 | 94.37 | 50.89 | 98.99 | 38.08 | 95.90 | 42.88 | 50.00 | 82.75 | 55.50 | 76.69 |
| | 91.12 | KNN$_{10}$ | 73.38 | 86.82 | 67.98 | 87.46 | 71.31 | 87.84 | 92.90 | 77.74 | 76.39 | 84.96 | 6.98 | 99.12 | 5.20 | 98.98 |
| | | KNN$_{20}$ | 74.90 | 86.41 | 66.29 | 87.79 | 73.82 | 87.21 | 93.95 | 76.51 | 77.24 | 84.48 | 6.98 | 98.96 | 5.50 | 98.85 |
| | | KNN$_{50}$ | 76.66 | 86.04 | 66.41 | 88.48 | 78.29 | 86.39 | 95.50 | 74.76 | 79.22 | 83.92 | 5.81 | 98.68 | 5.90 | 98.70 |
| | | KNN$_{100}$ | 77.54 | 85.61 | 65.41 | 88.99 | 82.16 | 85.43 | 96.23 | 73.37 | 80.33 | 83.35 | 6.98 | 98.34 | 6.30 | 98.51 |
| | | KNN$_{200}$ | 77.34 | 84.97 | 64.89 | 89.41 | 84.44 | 84.40 | 96.88 | 71.70 | 80.89 | 82.62 | 9.30 | 97.89 | 9.60 | 98.24 |
| | | KNN$_{400}$ | 77.98 | 84.01 | 64.48 | 89.61 | 87.11 | 83.04 | 97.93 | 69.24 | 81.88 | 81.48 | 15.12 | 97.29 | 12.20 | 97.81 |
| | | **Pretrain on ImageNet** | | | | | | | | | | | | | | |
| | | KNN$_{10}$ | 96.96 | 51.14 | 94.62 | 51.75 | 98.76 | 53.84 | 99.59 | 37.60 | 97.48 | 48.58 | 83.56 | 85.00 | 20.80 | 97.00 |
| | | KNN$_{20}$ | 96.96 | 50.37 | 94.34 | 51.54 | 98.92 | 52.98 | 99.59 | 36.60 | 97.45 | 47.87 | 83.56 | 84.49 | 22.70 | 96.71 |
| | | KNN$_{50}$ | 96.92 | 49.29 | 94.29 | 51.30 | 99.00 | 51.84 | 99.59 | 35.15 | 97.45 | 46.90 | 83.56 | 84.03 | 26.70 | 96.21 |
| | – | KNN$_{100}$ | 97.12 | 48.60 | 94.54 | 51.25 | 99.16 | 51.11 | 99.55 | 34.36 | 97.59 | 46.33 | 82.19 | 83.31 | 29.40 | 95.67 |
| | | KNN$_{200}$ | 97.48 | 47.85 | 94.58 | 51.12 | 99.08 | 50.41 | 99.68 | 33.44 | 97.70 | 45.70 | 83.56 | 82.24 | 33.30 | 94.92 |
| | | KNN$_{400}$ | 97.76 | 47.00 | 95.18 | 50.95 | 99.04 | 49.57 | 99.68 | 32.27 | 97.92 | 44.95 | 83.56 | 80.92 | 40.10 | 93.86 |
| **Swin$_{Base}$** | | **Pretrain on ImageNet→ Finetune on RVL-CDIP (ID)** | | | | | | | | | | | | | | |
| | | MSP | 47.64 | 88.09 | 49.90 | 88.11 | 58.22 | 83.14 | 50.28 | 88.90 | 51.51 | 87.06 | 49.32 | 91.31 | 36.50 | 93.63 |
| | | MaxLogit | 42.39 | 93.11 | 42.47 | 93.45 | 58.62 | 88.79 | 45.90 | 93.18 | 47.34 | 92.13 | 50.68 | 92.50 | 32.20 | 95.65 |
| | | Energy | 43.15 | 93.05 | 42.95 | 93.40 | 59.02 | 88.70 | 46.71 | 93.07 | 47.96 | 92.06 | 52.05 | 92.38 | 33.60 | 95.49 |
| | | Maha$_{Norm}$ | 99.92 | 28.31 | 99.88 | 25.72 | 99.96 | 32.66 | 99.96 | 22.89 | 99.93 | 27.40 | 100.00 | 35.29 | 100.00 | 29.82 |
| | | Maha$_{UnNorm}$ | 99.96 | 28.98 | 100.00 | 26.25 | 99.96 | 33.32 | 100.00 | 23.69 | 99.98 | 28.06 | 100.00 | 35.87 | 100.00 | 30.43 |
| | 95.74 | KNN$_{10}$ | 49.44 | 92.82 | 46.73 | 92.87 | 42.90 | 92.57 | 72.69 | 88.45 | 52.94 | 91.68 | 16.44 | 96.73 | 6.10 | 98.30 |
| | | KNN$_{20}$ | 48.84 | 92.95 | 43.27 | 93.51 | 44.53 | 92.32 | 72.28 | 88.35 | 52.23 | 91.78 | 17.81 | 96.52 | 7.40 | 98.10 |
| | | KNN$_{50}$ | 46.44 | 93.26 | 39.25 | 94.57 | 47.41 | 92.09 | 73.34 | 87.87 | 51.61 | 91.95 | 26.03 | 96.15 | 8.60 | 97.80 |
| | | KNN$_{100}$ | 43.76 | 93.42 | 35.03 | 95.29 | 50.08 | 91.72 | 75.77 | 87.42 | 51.16 | 91.96 | 28.77 | 95.94 | 11.30 | 97.55 |
| | | KNN$_{200}$ | 41.59 | 93.56 | 27.84 | 95.86 | 52.19 | 91.45 | 76.66 | 87.13 | 49.57 | 92.00 | 27.40 | 95.75 | 13.20 | 97.31 |
| | | KNN$_{400}$ | 40.79 | 93.68 | 24.71 | 96.14 | 53.27 | 91.17 | 77.52 | 86.97 | 49.07 | 91.99 | 26.03 | 95.72 | 16.00 | 97.18 |
| | | **Pretrain on ImageNet** | | | | | | | | | | | | | | |
| | | KNN$_{10}$ | 98.56 | 52.75 | 95.06 | 55.14 | 99.36 | 58.85 | 99.80 | 41.86 | 98.20 | 52.15 | 65.75 | 93.26 | 2.10 | 99.35 |
| | | KNN$_{20}$ | 98.44 | 51.86 | 95.18 | 54.72 | 99.32 | 57.88 | 99.80 | 40.66 | 98.18 | 51.28 | 68.49 | 92.52 | 2.60 | 99.22 |
| | | KNN$_{50}$ | 98.52 | 50.69 | 95.38 | 54.13 | 99.16 | 56.61 | 99.76 | 39.01 | 98.20 | 50.11 | 78.08 | 91.14 | 3.40 | 98.99 |
| | – | KNN$_{100}$ | 98.72 | 49.96 | 95.66 | 53.80 | 99.16 | 55.84 | 99.76 | 38.16 | 98.32 | 49.44 | 79.45 | 89.89 | 4.30 | 98.77 |
| | | KNN$_{200}$ | 98.68 | 49.22 | 95.90 | 53.39 | 99.20 | 55.14 | 99.76 | 37.31 | 98.38 | 48.76 | 84.93 | 88.40 | 5.40 | 98.47 |
| | | KNN$_{400}$ | 98.84 | 48.36 | 96.22 | 52.74 | 99.40 | 54.41 | 99.76 | 36.26 | 98.56 | 47.94 | 84.93 | 86.74 | 8.20 | 98.05 |
| **ViT$_{Base}$** | | **Pretrain on ImageNet→ Finetune on RVL-CDIP (ID)** | | | | | | | | | | | | | | |
| | | MSP | 56.81 | 89.14 | 52.19 | 91.80 | 67.48 | 84.26 | 59.90 | 88.77 | 59.10 | 88.49 | 47.67 | 92.98 | 59.50 | 91.99 |
| | | MaxLogit | 50.76 | 91.37 | 44.60 | 93.75 | 68.04 | 86.94 | 55.15 | 91.81 | 54.64 | 90.97 | 40.70 | 94.20 | 52.40 | 93.16 |
| | | Energy | 51.16 | 91.31 | 44.52 | 93.75 | 69.43 | 86.81 | 56.09 | 91.77 | 55.30 | 90.91 | 38.37 | 94.11 | 53.20 | 93.11 |
| | | Maha$_{Norm}$ | 90.63 | 70.10 | 91.84 | 65.75 | 89.55 | 70.83 | 97.81 | 57.37 | 92.46 | 66.01 | 100.00 | 58.34 | 82.20 | 77.09 |
| | | Maha$_{UnNorm}$ | 90.63 | 70.10 | 91.80 | 65.75 | 89.55 | 70.83 | 97.81 | 57.37 | 92.45 | 66.01 | 100.00 | 58.34 | 82.20 | 77.09 |
| | 94.38 | KNN$_{10}$ | 62.57 | 90.12 | 57.73 | 90.91 | 53.67 | 90.36 | 84.50 | 86.19 | 64.62 | 89.40 | 12.79 | 97.96 | 13.00 | 97.92 |
| | | KNN$_{20}$ | 63.01 | 90.24 | 56.01 | 91.51 | 55.03 | 90.02 | 84.38 | 86.01 | 64.61 | 89.44 | 15.12 | 97.76 | 14.90 | 97.67 |
| | | KNN$_{50}$ | 61.97 | 90.62 | 53.23 | 92.62 | 58.26 | 89.57 | 84.25 | 85.64 | 64.43 | 89.61 | 16.28 | 97.38 | 19.80 | 97.24 |
| | | KNN$_{100}$ | 60.29 | 90.85 | 49.70 | 93.53 | 60.38 | 89.07 | 84.01 | 85.43 | 63.60 | 89.72 | 16.28 | 97.05 | 23.60 | 96.82 |
| | | KNN$_{200}$ | 58.45 | 91.04 | 45.04 | 94.36 | 62.89 | 88.54 | 84.17 | 85.33 | 62.64 | 89.82 | 22.09 | 96.75 | 27.50 | 96.39 |
| | | KNN$_{400}$ | 58.01 | 91.11 | 40.42 | 94.94 | 65.08 | 87.94 | 83.93 | 85.19 | 61.86 | 89.80 | 24.42 | 96.47 | 31.60 | 96.03 |
| | | **Pretrain on ImageNet** | | | | | | | | | | | | | | |
| | | KNN$_{10}$ | 98.48 | 52.15 | 95.02 | 56.94 | 99.48 | 53.77 | 99.47 | 38.90 | 98.11 | 50.44 | 93.15 | 90.27 | 20.40 | 97.13 |
| | | KNN$_{20}$ | 98.48 | 51.41 | 95.06 | 56.61 | 99.44 | 52.92 | 99.55 | 37.61 | 98.13 | 49.64 | 94.52 | 89.44 | 22.60 | 96.80 |
| | | KNN$_{50}$ | 98.32 | 50.43 | 94.86 | 56.21 | 99.40 | 51.86 | 99.59 | 35.82 | 98.04 | 48.58 | 97.26 | 88.23 | 26.60 | 96.25 |
| | – | KNN$_{100}$ | 98.40 | 49.76 | 95.06 | 55.90 | 99.44 | 51.15 | 99.59 | 34.59 | 98.12 | 47.85 | 98.63 | 87.24 | 31.20 | 95.76 |
| | | KNN$_{200}$ | 98.60 | 49.01 | 95.46 | 55.55 | 99.48 | 50.46 | 99.55 | 33.24 | 98.27 | 47.07 | 98.63 | 86.08 | 36.30 | 95.15 |
| | | KNN$_{400}$ | 98.64 | 48.04 | 95.50 | 55.01 | 99.44 | 49.60 | 99.55 | 31.52 | 98.28 | 46.04 | 100.00 | 84.82 | 43.80 | 94.44 |

Table 10: OOD detection performance for document classification (select OOD categories achieve the best performance across most of the models with different modalities).

| | | OOD Dataset (In-Domain) | | | | | | | | | | OOD Dataset (Out-Domain) | | | |
|---|---|---|---|---|---|---|---|---|---|---|---|---|---|---|---|
| ID Acc | Method | Email | | Resume | | File folder | | Sci. publication | | Average | | Sci. Poster | | Receipt | |
| | | FPR95 | AUROC | FPR95 | AUROC | FPR95 | AUROC | FPR95 | AUROC | FPR95 | AUROC | FPR95 | AUROC | FPR95 | AUROC |
| **RoBERTa_Base — Pretrain on pure-text data→ Finetune on RVL-CDIP (ID)** | | | | | | | | | | | | | | | |
| | MSP | 96.22 | 60.38 | 90.67 | 71.72 | 93.82 | 59.47 | 93.86 | 65.51 | 93.64 | 64.27 | 91.86 | 70.57 | 93.00 | 69.99 |
| | MaxLogit | 99.21 | 66.57 | 95.80 | 73.66 | 95.47 | 66.81 | 97.09 | 65.63 | 96.89 | 68.17 | 94.19 | 77.17 | 94.60 | 74.69 |
| | Energy | 99.60 | 66.53 | 95.64 | 73.57 | 95.14 | 66.82 | 97.21 | 65.35 | 97.15 | 68.07 | 94.19 | 77.44 | 94.90 | 74.90 |
| | Maha_Norm | 99.24 | 40.24 | 98.80 | 35.35 | 99.72 | 24.11 | 99.19 | 32.50 | 99.24 | 33.05 | 100.00 | 42.36 | 99.70 | 31.42 |
| | Maha_UnNorm | 99.24 | 40.68 | 98.80 | 35.49 | 99.72 | 23.41 | 99.19 | 32.51 | 99.24 | 33.02 | 100.00 | 42.81 | 99.70 | 31.49 |
| 86.13 | KNN_10 | 83.70 | 82.77 | 69.02 | 84.28 | 88.32 | 74.06 | 86.11 | 74.02 | 81.79 | 78.78 | 43.02 | 92.74 | 72.00 | 88.87 |
| | KNN_20 | 84.50 | 82.35 | 69.06 | 84.21 | 88.20 | 73.71 | 86.72 | 74.02 | 82.12 | 78.57 | 48.84 | 92.38 | 73.80 | 88.31 |
| | KNN_50 | 84.98 | 81.57 | 68.86 | 84.06 | 88.08 | 73.01 | 87.08 | 73.94 | 82.25 | 78.14 | 54.65 | 91.92 | 75.40 | 87.44 |
| | KNN_100 | 86.25 | 80.88 | 70.26 | 83.80 | 88.28 | 72.40 | 87.44 | 73.89 | 83.06 | 77.74 | 58.14 | 91.50 | 78.20 | 86.68 |
| | KNN_200 | 86.72 | 80.01 | 71.22 | 83.43 | 88.32 | 71.46 | 87.61 | 73.56 | 83.47 | 77.11 | 59.30 | 91.07 | 78.80 | 85.94 |
| **Pretrain on pure-text data** | | | | | | | | | | | | | | | |
| | KNN_10 | 86.09 | 75.63 | 95.12 | 58.62 | 97.71 | 59.75 | 98.95 | 50.54 | 94.47 | 61.14 | 10.47 | 98.46 | 89.80 | 63.01 |
| | KNN_20 | 86.29 | 74.92 | 95.00 | 58.14 | 97.71 | 58.88 | 99.03 | 49.49 | 94.51 | 60.36 | 12.79 | 98.35 | 90.80 | 60.59 |
| | KNN_50 | 87.32 | 73.55 | 94.64 | 57.53 | 97.83 | 57.56 | 99.15 | 48.11 | 94.73 | 59.19 | 12.79 | 98.11 | 93.30 | 56.61 |
| | KNN_100 | 89.27 | 72.48 | 94.28 | 57.12 | 97.99 | 56.52 | 99.11 | 47.37 | 95.16 | 58.37 | 11.63 | 97.89 | 94.30 | 52.98 |
| | KNN_200 | 90.26 | 71.07 | 93.96 | 56.45 | 98.11 | 55.37 | 98.75 | 46.50 | 95.27 | 57.35 | 13.95 | 97.64 | 95.70 | 48.45 |
| **Longformer_4096 — Pretrain on pure-text data→ Finetune on RVL-CDIP (ID)** | | | | | | | | | | | | | | | |
| | MSP | 96.90 | 60.55 | 96.20 | 59.14 | 96.31 | 55.72 | 97.82 | 55.12 | 96.81 | 57.63 | 95.35 | 80.44 | 99.60 | 52.82 |
| | MaxLogit | 98.97 | 68.97 | 97.60 | 65.64 | 95.67 | 63.42 | 98.63 | 62.87 | 97.72 | 65.23 | 97.67 | 88.42 | 99.70 | 71.54 |
| | Energy | 99.44 | 68.96 | 97.92 | 65.63 | 95.83 | 63.42 | 98.71 | 62.83 | 97.98 | 65.21 | 97.67 | 88.46 | 99.90 | 71.55 |
| | Maha_Norm | 97.85 | 41.33 | 96.68 | 45.36 | 89.93 | 41.96 | 96.93 | 65.93 | 95.35 | 48.64 | 94.19 | 60.62 | 97.80 | 53.44 |
| | Maha_UnNorm | 97.97 | 35.08 | 95.96 | 44.92 | 98.80 | 27.08 | 88.17 | 68.37 | 95.22 | 43.86 | 88.37 | 63.31 | 95.50 | 57.51 |
| 88.34 | KNN_10 | 68.28 | 88.72 | 69.62 | 83.36 | 78.17 | 85.08 | 90.88 | 74.98 | 76.74 | 83.04 | 16.28 | 96.90 | 81.60 | 86.94 |
| | KNN_20 | 68.04 | 88.61 | 70.10 | 83.22 | 77.53 | 84.92 | 90.75 | 74.95 | 76.60 | 82.92 | 16.28 | 96.84 | 81.80 | 86.49 |
| | KNN_50 | 69.28 | 88.29 | 70.98 | 82.92 | 78.29 | 84.46 | 90.96 | 74.82 | 77.38 | 82.62 | 19.77 | 96.59 | 83.40 | 85.71 |
| | KNN_100 | 69.28 | 88.15 | 71.34 | 82.69 | 78.49 | 84.21 | 90.43 | 74.86 | 77.39 | 82.48 | 22.09 | 96.38 | 83.90 | 85.17 |
| | KNN_200 | 69.44 | 88.06 | 72.14 | 82.43 | 78.65 | 84.00 | 89.58 | 74.74 | 77.45 | 82.31 | 23.26 | 96.16 | 84.70 | 84.60 |
| **Pretrain on pure-text data** | | | | | | | | | | | | | | | |
| | KNN_10 | 97.42 | 47.77 | 95.72 | 50.09 | 97.67 | 46.58 | 99.52 | 38.61 | 97.58 | 45.76 | 45.35 | 93.92 | 100.00 | 63.03 |
| | KNN_20 | 97.46 | 46.91 | 95.60 | 49.80 | 97.71 | 46.02 | 99.52 | 38.21 | 97.57 | 45.24 | 46.51 | 93.77 | 100.00 | 61.92 |
| | KNN_50 | 97.58 | 45.68 | 95.56 | 49.45 | 97.75 | 45.19 | 99.52 | 37.72 | 97.60 | 44.51 | 50.00 | 93.60 | 100.00 | 60.35 |
| | KNN_100 | 97.66 | 44.78 | 95.60 | 49.17 | 97.87 | 44.63 | 99.56 | 37.57 | 97.67 | 44.04 | 51.16 | 93.48 | 100.00 | 58.89 |
| | KNN_200 | 97.85 | 43.79 | 95.72 | 48.71 | 97.87 | 44.06 | 99.60 | 37.28 | 97.76 | 43.46 | 51.16 | 93.42 | 100.00 | 57.02 |
| **ResNet-50 — Pretrain on ImageNet→ Finetune on RVL-CDIP (ID)** | | | | | | | | | | | | | | | |
| | MSP | 60.53 | 87.26 | 69.53 | 87.00 | 27.86 | 95.13 | 94.05 | 75.79 | 62.99 | 86.30 | 91.78 | 74.40 | 27.80 | 95.47 |
| | MaxLogit | 59.98 | 89.27 | 72.61 | 88.02 | 30.04 | 95.41 | 93.39 | 75.38 | 64.00 | 87.02 | 80.82 | 79.89 | 30.00 | 95.29 |
| | Energy | 63.71 | 89.14 | 75.64 | 87.55 | 45.71 | 94.15 | 92.77 | 75.02 | 64.96 | 86.46 | 78.08 | 81.07 | 62.20 | 93.44 |
| | Maha_Norm | 76.11 | 83.47 | 91.76 | 78.43 | 53.19 | 87.52 | 96.50 | 62.02 | 79.39 | 77.86 | 4.11 | 99.12 | 0.20 | 99.81 |
| | Maha_UnNorm | 99.56 | 11.67 | 99.88 | 30.74 | 99.13 | 8.54 | 99.65 | 21.10 | 99.56 | 18.01 | 69.86 | 76.26 | 91.40 | 39.85 |
| 85.25 | KNN_10 | 72.46 | 85.68 | 85.69 | 85.30 | 68.62 | 76.01 | 96.15 | 55.35 | 80.73 | 75.59 | 36.99 | 94.56 | 2.20 | 99.37 |
| | KNN_20 | 76.15 | 84.55 | 88.65 | 84.22 | 66.13 | 80.67 | 96.54 | 56.31 | 81.87 | 76.44 | 38.36 | 93.81 | 2.70 | 99.28 |
| | KNN_50 | 80.37 | 82.61 | 92.00 | 82.49 | 60.98 | 86.77 | 96.93 | 59.06 | 82.57 | 77.73 | 47.95 | 92.42 | 3.80 | 99.11 |
| | KNN_100 | 84.70 | 80.54 | 95.15 | 80.64 | 51.29 | 91.78 | 97.16 | 61.19 | 82.08 | 78.54 | 50.68 | 91.01 | 4.70 | 98.91 |
| | KNN_200 | 89.07 | 78.01 | 96.65 | 78.76 | 38.58 | 93.84 | 97.63 | 62.68 | 80.48 | 78.32 | 57.53 | 89.28 | 6.00 | 98.60 |
| **Pretrain on ImageNet** | | | | | | | | | | | | | | | |
| | KNN_10 | 99.72 | 40.94 | 99.76 | 21.52 | 52.47 | 91.03 | 98.33 | 45.40 | 87.54 | 49.72 | 84.93 | 84.38 | 20.40 | 97.12 |
| | KNN_20 | 99.68 | 41.18 | 99.65 | 20.68 | 50.61 | 91.63 | 98.41 | 44.65 | 87.09 | 49.54 | 86.30 | 83.94 | 23.40 | 96.87 |
| | KNN_50 | 99.64 | 41.58 | 99.65 | 19.48 | 46.97 | 92.36 | 98.37 | 43.49 | 86.16 | 49.23 | 84.93 | 83.70 | 26.90 | 96.43 |
| | KNN_100 | 99.64 | 42.19 | 99.65 | 18.98 | 44.91 | 92.84 | 98.33 | 42.86 | 85.63 | 49.22 | 84.93 | 83.12 | 29.20 | 95.98 |
| | KNN_200 | 99.64 | 42.88 | 99.65 | 18.31 | 43.33 | 93.25 | 98.29 | 42.26 | 85.23 | 49.18 | 84.93 | 82.53 | 32.80 | 95.37 |
| **Swin_Base — Pretrain on ImageNet→ Finetune on RVL-CDIP (ID)** | | | | | | | | | | | | | | | |
| | MSP | 70.23 | 81.87 | 67.68 | 85.31 | 43.97 | 92.68 | 83.78 | 79.40 | 66.42 | 84.82 | 86.30 | 78.23 | 54.10 | 91.62 |
| | MaxLogit | 54.73 | 87.04 | 46.51 | 92.30 | 17.25 | 96.51 | 90.86 | 74.11 | 52.34 | 87.49 | 82.19 | 83.20 | 34.40 | 94.82 |
| | Energy | 54.05 | 87.11 | 44.38 | 92.49 | 16.38 | 96.63 | 91.29 | 73.59 | 51.53 | 87.46 | 84.93 | 83.07 | 33.80 | 94.82 |
| | Maha_Norm | 99.72 | 28.96 | 99.92 | 18.92 | 98.61 | 23.55 | 99.73 | 44.10 | 99.50 | 28.88 | 100.00 | 53.54 | 99.90 | 38.03 |
| | Maha_UnNorm | 99.05 | 29.74 | 100.00 | 20.22 | 99.76 | 22.62 | 99.81 | 43.98 | 99.66 | 29.14 | 97.26 | 53.69 | 100.00 | 39.09 |
| 91.25 | KNN_10 | 56.08 | 90.66 | 48.80 | 92.84 | 38.31 | 93.31 | 91.02 | 66.91 | 58.55 | 85.93 | 27.40 | 96.03 | 3.30 | 98.84 |
| | KNN_20 | 54.61 | 90.95 | 49.98 | 92.68 | 27.58 | 95.24 | 91.44 | 66.54 | 55.90 | 86.85 | 26.03 | 96.35 | 4.00 | 98.76 |
| | KNN_50 | 55.25 | 90.68 | 52.15 | 92.37 | 15.75 | 97.28 | 91.25 | 71.62 | 53.60 | 87.99 | 28.77 | 96.10 | 4.90 | 98.59 |
| | KNN_100 | 56.20 | 90.31 | 54.75 | 92.17 | 9.14 | 98.00 | 91.13 | 75.11 | 52.80 | 88.90 | 30.14 | 95.77 | 6.50 | 98.35 |
| | KNN_200 | 55.64 | 90.00 | 54.24 | 92.14 | 8.39 | 98.14 | 90.90 | 77.82 | 52.29 | 89.52 | 32.88 | 95.49 | 7.60 | 98.12 |
| **Pretrain on ImageNet** | | | | | | | | | | | | | | | |
| | KNN_10 | 99.84 | 43.55 | 99.76 | 20.64 | 47.92 | 93.20 | 98.91 | 37.55 | 86.61 | 48.74 | 58.90 | 93.88 | 1.60 | 99.32 |
| | KNN_20 | 99.84 | 43.78 | 99.76 | 19.61 | 44.76 | 93.61 | 98.91 | 37.01 | 85.82 | 48.50 | 65.75 | 93.42 | 2.10 | 99.20 |
| | KNN_50 | 99.84 | 44.47 | 99.80 | 18.36 | 41.31 | 94.14 | 99.03 | 36.45 | 85.00 | 48.36 | 72.60 | 92.69 | 2.60 | 99.00 |
| | KNN_100 | 99.88 | 45.26 | 99.80 | 17.92 | 39.97 | 94.39 | 99.03 | 36.71 | 84.67 | 48.57 | 79.45 | 91.97 | 3.70 | 98.81 |
| | KNN_200 | 99.88 | 45.90 | 99.72 | 17.52 | 36.64 | 94.64 | 98.76 | 37.33 | 83.75 | 48.85 | 80.82 | 91.29 | 4.40 | 98.56 |
| **ViT_Base — Pretrain on ImageNet→ Finetune on RVL-CDIP (ID)** | | | | | | | | | | | | | | | |
| | MSP | 61.25 | 85.84 | 66.57 | 85.04 | 40.44 | 93.10 | 85.84 | 81.83 | 63.52 | 86.45 | 73.97 | 80.66 | 60.30 | 90.41 |
| | MaxLogit | 53.02 | 90.37 | 55.77 | 88.86 | 19.91 | 96.25 | 92.38 | 79.09 | 55.27 | 88.79 | 76.71 | 85.16 | 50.60 | 93.12 |
| | Energy | 51.79 | 90.49 | 55.07 | 89.03 | 17.53 | 96.53 | 92.69 | 79.20 | 54.27 | 88.81 | 79.45 | 85.01 | 50.10 | 93.20 |
| | Maha_Norm | 97.26 | 43.71 | 93.89 | 56.82 | 96.76 | 37.35 | 97.47 | 63.36 | 96.34 | 50.31 | 100.00 | 55.27 | 87.20 | 65.24 |
| | Maha_UnNorm | 97.26 | 43.71 | 93.89 | 56.82 | 96.76 | 37.35 | 97.47 | 63.36 | 96.34 | 50.31 | 100.00 | 55.27 | 87.20 | 65.24 |
| 89.97 | KNN_10 | 54.13 | 91.18 | 52.86 | 91.18 | 58.49 | 87.46 | 92.88 | 65.98 | 64.59 | 83.95 | 42.47 | 95.07 | 11.00 | 97.94 |
| | KNN_20 | 54.21 | 91.18 | 53.17 | 90.99 | 50.61 | 89.35 | 93.04 | 67.52 | 62.76 | 84.76 | 43.84 | 94.98 | 13.10 | 97.62 |
| | KNN_50 | 54.53 | 91.05 | 53.33 | 90.79 | 41.95 | 92.82 | 93.00 | 72.06 | 60.70 | 86.68 | 42.47 | 94.74 | 17.30 | 97.12 |
| | KNN_100 | 54.65 | 90.81 | 54.12 | 90.56 | 30.79 | 95.78 | 93.04 | 75.39 | 58.15 | 88.14 | 45.21 | 94.24 | 22.00 | 96.58 |
| | KNN_200 | 56.00 | 90.51 | 55.42 | 90.40 | 15.31 | 97.14 | 92.61 | 77.76 | 54.84 | 88.95 | 47.95 | 93.51 | 27.40 | 95.93 |
| | KNN_400 | 57.35 | 90.11 | 55.77 | 90.36 | 11.56 | 97.45 | 92.49 | 78.73 | 54.29 | 89.16 | 56.16 | 92.87 | 34.00 | 95.31 |
| **Pretrain on ImageNet** | | | | | | | | | | | | | | | |
| | KNN_10 | 99.80 | 46.46 | 99.68 | 26.50 | 58.65 | 90.61 | 98.72 | 46.40 | 89.21 | 52.49 | 87.67 | 91.39 | 19.90 | 97.25 |
| | KNN_20 | 99.80 | 46.02 | 99.65 | 25.69 | 57.30 | 91.01 | 98.72 | 46.46 | 88.87 | 52.30 | 90.41 | 90.87 | 21.70 | 97.01 |
| | KNN_50 | 99.80 | 45.48 | 99.61 | 24.76 | 55.16 | 91.52 | 98.76 | 46.69 | 88.33 | 52.11 | 94.52 | 89.99 | 24.30 | 96.62 |
| | KNN_100 | 99.80 | 45.33 | 99.65 | 24.43 | 54.81 | 91.90 | 98.72 | 47.10 | 88.24 | 52.19 | 95.89 | 89.31 | 28.80 | 96.27 |
| | KNN_200 | 99.68 | 45.19 | 99.65 | 24.18 | 53.66 | 92.27 | 98.64 | 47.72 | 87.91 | 52.34 | 97.26 | 88.57 | 32.00 | 95.88 |

Table 11: OOD detection performance for document classification (randomly select four categories as OOD).

| Model | ID Acc | Method | Letter FPR95 | Letter AUROC | Handwritten FPR95 | Handwritten AUROC | Advertisement FPR95 | Advertisement AUROC | Memo FPR95 | Memo AUROC | Average FPR95 | Average AUROC | Sci. Poster FPR95 | Sci. Poster AUROC | Receipt FPR95 | Receipt AUROC |
|---|---|---|---|---|---|---|---|---|---|---|---|---|---|---|---|---|
| | | | | | | | **OOD Dataset (In-Domain)** | | | | | | | **OOD Dataset (Out-Domain)** | | |
| $RoBERTa_{Base}$ | | *Pretrain on pure-text data→ Finetune on RVL-CDIP (ID)* | | | | | | | | | | | | | | |
| | 88.86 | MSP | 70.22 | 79.21 | 50.14 | 87.24 | 84.64 | 67.80 | 91.42 | 57.99 | 74.10 | 73.06 | 95.35 | 59.75 | 94.30 | 55.12 |
| | | MaxLogit | 66.04 | 87.51 | 39.65 | 92.53 | 86.47 | 77.03 | 91.67 | 71.84 | 70.96 | 82.23 | 100.00 | 77.89 | 96.80 | 71.96 |
| | | Energy | 66.20 | 87.57 | 38.19 | 92.59 | 87.35 | 77.03 | 91.67 | 71.89 | 70.85 | 82.27 | 100.00 | 77.92 | 96.80 | 71.96 |
| | | $Maha_{Norm}$ | 87.40 | 42.73 | 89.43 | 35.19 | 96.57 | 39.03 | 99.03 | 36.06 | 93.11 | 38.25 | 98.84 | 34.76 | 97.10 | 57.71 |
| | | $Maha_{UnNorm}$ | 89.07 | 40.42 | 89.35 | 33.53 | 96.69 | 38.59 | 99.03 | 35.87 | 93.54 | 37.10 | 98.84 | 33.06 | 97.10 | 57.49 |
| | | $KNN_{10}$ | 62.62 | 80.19 | 60.98 | 70.90 | 75.62 | 80.24 | 85.84 | 69.20 | 71.26 | 75.13 | 94.19 | 81.99 | 90.40 | 82.48 |
| | | $KNN_{20}$ | 63.18 | 80.10 | 60.07 | 71.17 | 75.90 | 80.03 | 85.72 | 68.88 | 71.22 | 75.04 | 94.19 | 81.75 | 91.20 | 81.89 |
| | | $KNN_{50}$ | 63.78 | 80.00 | 57.30 | 71.70 | 76.34 | 79.67 | 85.88 | 68.38 | 70.82 | 74.94 | 94.19 | 81.45 | 91.80 | 81.09 |
| | | $KNN_{100}$ | 64.77 | 79.98 | 54.33 | 71.94 | 77.37 | 79.32 | 86.08 | 67.80 | 70.64 | 74.76 | 94.19 | 81.20 | 91.90 | 80.47 |
| | | $KNN_{200}$ | 64.85 | 79.85 | 55.64 | 71.84 | 77.57 | 78.79 | 86.33 | 66.78 | 71.10 | 74.32 | 94.19 | 80.68 | 91.60 | 79.59 |
| | | *Pretrain on pure-text data* | | | | | | | | | | | | | | |
| | – | $KNN_{10}$ | 85.53 | 59.90 | 98.61 | 21.79 | 96.21 | 56.72 | 97.69 | 58.39 | 94.51 | 49.20 | 12.79 | 98.01 | 84.50 | 65.73 |
| | | $KNN_{20}$ | 85.45 | 59.27 | 98.73 | 21.19 | 96.21 | 55.63 | 97.90 | 57.05 | 94.57 | 48.28 | 12.79 | 97.91 | 86.10 | 63.57 |
| | | $KNN_{50}$ | 86.80 | 57.94 | 98.77 | 20.45 | 96.89 | 54.12 | 98.30 | 55.35 | 95.19 | 46.96 | 13.95 | 97.60 | 89.30 | 59.64 |
| | | $KNN_{100}$ | 88.47 | 56.71 | 98.81 | 19.97 | 96.81 | 52.89 | 98.18 | 53.93 | 95.57 | 45.88 | 13.95 | 97.38 | 91.10 | 55.17 |
| | | $KNN_{200}$ | 90.14 | 55.70 | 98.73 | 19.60 | 96.65 | 51.44 | 98.22 | 52.52 | 95.94 | 44.82 | 19.77 | 97.00 | 93.50 | 50.60 |
| $Longformer_{4096}$ | | *Pretrain on pure-text data→ Finetune on RVL-CDIP (ID)* | | | | | | | | | | | | | | |
| | 92.08 | MSP | 65.96 | 69.58 | 50.38 | 77.93 | 81.52 | 60.89 | 90.21 | 54.23 | 72.02 | 65.66 | 82.56 | 60.14 | 95.00 | 50.90 |
| | | MaxLogit | 62.19 | 87.35 | 44.64 | 89.79 | 79.97 | 78.84 | 88.39 | 68.08 | 68.80 | 81.02 | 80.23 | 84.19 | 94.30 | 77.36 |
| | | Energy | 61.27 | 87.35 | 43.61 | 89.81 | 79.13 | 78.85 | 88.15 | 68.08 | 68.04 | 81.02 | 80.23 | 84.19 | 94.30 | 77.37 |
| | | $Maha_{Norm}$ | 93.64 | 50.82 | 92.52 | 43.64 | 95.09 | 45.15 | 96.36 | 40.35 | 94.40 | 44.99 | 95.35 | 40.73 | 97.80 | 58.87 |
| | | $Maha_{UnNorm}$ | 95.98 | 44.60 | 94.58 | 32.97 | 98.44 | 37.27 | 98.83 | 29.41 | 96.96 | 36.06 | 95.35 | 37.84 | 98.70 | 59.91 |
| | | $KNN_{10}$ | 58.65 | 79.54 | 50.77 | 71.81 | 66.56 | 83.48 | 80.87 | 75.19 | 64.21 | 77.51 | 58.14 | 92.78 | 90.00 | 77.76 |
| | | $KNN_{20}$ | 57.81 | 79.43 | 51.40 | 71.72 | 67.00 | 83.35 | 81.15 | 74.86 | 64.34 | 77.34 | 58.14 | 92.57 | 89.70 | 77.12 |
| | | $KNN_{50}$ | 58.77 | 79.30 | 51.60 | 71.67 | 66.72 | 83.15 | 81.31 | 74.36 | 64.60 | 77.12 | 61.63 | 92.24 | 89.80 | 76.17 |
| | | $KNN_{100}$ | 61.39 | 79.16 | 52.75 | 71.61 | 67.84 | 82.93 | 81.76 | 73.91 | 65.94 | 76.90 | 62.79 | 91.99 | 89.80 | 75.29 |
| | | $KNN_{200}$ | 62.78 | 78.99 | 53.94 | 71.54 | 69.39 | 82.65 | 82.77 | 73.30 | 67.22 | 76.62 | 63.95 | 91.78 | 89.70 | 74.24 |
| | | *Pretrain on pure-text data* | | | | | | | | | | | | | | |
| | – | $KNN_{10}$ | 99.40 | 47.83 | 100.00 | 27.75 | 98.28 | 47.03 | 93.20 | 60.40 | 97.72 | 45.75 | 46.51 | 93.85 | 100.00 | 63.64 |
| | | $KNN_{20}$ | 99.44 | 47.33 | 100.00 | 27.48 | 98.32 | 46.49 | 93.24 | 60.22 | 97.75 | 45.38 | 48.84 | 93.70 | 100.00 | 62.79 |
| | | $KNN_{50}$ | 99.44 | 46.33 | 100.00 | 27.23 | 98.40 | 45.85 | 93.41 | 60.05 | 97.81 | 44.86 | 51.16 | 93.51 | 100.00 | 61.55 |
| | | $KNN_{100}$ | 99.44 | 45.67 | 100.00 | 27.31 | 98.44 | 45.23 | 93.53 | 59.90 | 97.85 | 44.53 | 52.33 | 93.40 | 100.00 | 60.31 |
| | | $KNN_{200}$ | 99.44 | 45.06 | 100.00 | 27.32 | 98.52 | 44.39 | 93.61 | 59.66 | 97.89 | 44.11 | 52.33 | 93.31 | 100.00 | 58.52 |
| ResNet-50 | | *Pretrain on ImageNet→ Finetune on RVL-CDIP (ID)* | | | | | | | | | | | | | | |
| | 87.80 | MSP | 70.58 | 85.35 | 55.29 | 89.88 | 64.29 | 86.54 | 71.15 | 85.58 | 65.33 | 86.84 | 54.79 | 91.70 | 77.20 | 84.67 |
| | | MaxLogit | 64.25 | 87.46 | 53.59 | 90.72 | 49.70 | 90.60 | 64.45 | 88.71 | 58.00 | 89.37 | 36.99 | 95.13 | 78.90 | 86.86 |
| | | Energy | 62.66 | 87.65 | 58.33 | 90.33 | 46.00 | 91.26 | 63.56 | 89.05 | 57.64 | 89.57 | 32.88 | 95.69 | 83.00 | 87.05 |
| | | $Maha_{Norm}$ | 95.25 | 64.97 | 38.94 | 92.28 | 66.60 | 88.40 | 94.74 | 68.11 | 73.88 | 78.44 | 1.37 | 99.43 | 0.70 | 99.51 |
| | | $Maha_{UnNorm}$ | 96.71 | 54.59 | 91.94 | 51.50 | 96.66 | 52.22 | 98.92 | 46.56 | 96.06 | 51.22 | 91.78 | 59.51 | 79.70 | 71.18 |
| | | $KNN_{10}$ | 90.99 | 79.37 | 56.36 | 90.64 | 72.41 | 86.20 | 89.17 | 81.74 | 77.23 | 84.49 | 2.74 | 99.32 | 39.70 | 93.70 |
| | | $KNN_{20}$ | 92.17 | 78.00 | 47.47 | 92.61 | 68.27 | 88.42 | 90.85 | 80.23 | 74.69 | 84.82 | 2.74 | 99.25 | 43.80 | 93.08 |
| | | $KNN_{50}$ | 94.32 | 75.96 | 28.44 | 94.49 | 65.65 | 89.27 | 92.78 | 77.91 | 70.30 | 84.41 | 1.37 | 98.97 | 49.70 | 92.09 |
| | | $KNN_{100}$ | 95.58 | 74.02 | 27.21 | 95.07 | 60.44 | 89.78 | 94.22 | 75.63 | 69.36 | 83.62 | 2.74 | 98.67 | 53.80 | 91.10 |
| | | $KNN_{200}$ | 96.23 | 71.92 | 28.08 | 94.97 | 52.05 | 90.26 | 95.79 | 73.16 | 68.04 | 82.58 | 9.59 | 98.25 | 56.40 | 90.30 |
| | | *Pretrain on ImageNet* | | | | | | | | | | | | | | |
| | – | $KNN_{10}$ | 98.46 | 42.21 | 77.29 | 81.41 | 27.87 | 91.16 | 99.08 | 43.47 | 75.68 | 64.56 | 80.82 | 89.98 | 12.30 | 98.17 |
| | | $KNN_{20}$ | 98.66 | 41.00 | 76.78 | 81.70 | 29.22 | 92.27 | 99.08 | 42.29 | 75.94 | 64.32 | 83.56 | 89.30 | 14.10 | 97.97 |
| | | $KNN_{50}$ | 98.58 | 39.53 | 76.58 | 81.81 | 31.01 | 92.05 | 99.12 | 40.80 | 76.32 | 63.55 | 83.56 | 88.51 | 16.30 | 97.61 |
| | | $KNN_{100}$ | 98.62 | 38.62 | 77.13 | 81.49 | 32.64 | 91.84 | 99.12 | 39.86 | 76.88 | 62.95 | 83.56 | 87.80 | 19.50 | 97.23 |
| | | $KNN_{200}$ | 98.78 | 37.57 | 79.34 | 80.86 | 35.19 | 91.51 | 99.28 | 38.79 | 78.15 | 62.18 | 86.30 | 86.76 | 25.70 | 96.71 |
| $Swin_{Base}$ | | *Pretrain on ImageNet→ Finetune on RVL-CDIP (ID)* | | | | | | | | | | | | | | |
| | 92.42 | MSP | 63.96 | 87.03 | 65.21 | 88.15 | 73.56 | 79.72 | 61.40 | 88.46 | 66.03 | 85.84 | 84.93 | 74.34 | 49.60 | 92.49 |
| | | MaxLogit | 56.49 | 90.22 | 75.36 | 87.00 | 72.64 | 84.26 | 44.22 | 93.01 | 62.18 | 88.62 | 72.60 | 84.16 | 29.10 | 95.70 |
| | | Energy | 57.43 | 90.11 | 77.01 | 86.60 | 73.44 | 84.17 | 43.78 | 93.06 | 62.92 | 88.48 | 73.97 | 84.25 | 28.00 | 95.69 |
| | | $Maha_{Norm}$ | 99.96 | 27.23 | 99.09 | 49.37 | 99.56 | 43.96 | 99.96 | 24.62 | 99.64 | 36.30 | 100.00 | 60.31 | 100.00 | 28.99 |
| | | $Maha_{UnNorm}$ | 99.96 | 27.44 | 99.05 | 49.56 | 99.72 | 43.90 | 99.96 | 24.86 | 99.67 | 36.44 | 100.00 | 60.54 | 100.00 | 29.29 |
| | | $KNN_{10}$ | 60.27 | 90.12 | 66.90 | 90.76 | 49.66 | 89.15 | 47.67 | 92.67 | 56.12 | 90.68 | 42.47 | 94.28 | 7.20 | 98.56 |
| | | $KNN_{20}$ | 61.32 | 90.01 | 61.37 | 91.31 | 48.83 | 90.33 | 49.00 | 92.52 | 55.13 | 91.04 | 30.14 | 95.56 | 8.80 | 98.33 |
| | | $KNN_{50}$ | 62.22 | 89.78 | 56.44 | 91.56 | 50.34 | 89.55 | 48.52 | 92.30 | 54.38 | 90.80 | 26.03 | 95.72 | 11.80 | 97.97 |
| | | $KNN_{100}$ | 62.62 | 89.60 | 54.98 | 91.85 | 50.70 | 88.93 | 47.63 | 92.18 | 53.98 | 90.64 | 30.14 | 95.54 | 13.90 | 97.66 |
| | | $KNN_{200}$ | 61.77 | 89.66 | 56.67 | 91.63 | 51.89 | 88.34 | 47.63 | 92.24 | 54.49 | 90.47 | 35.62 | 95.15 | 15.90 | 97.32 |
| | | *Pretrain on ImageNet* | | | | | | | | | | | | | | |
| | – | $KNN_{10}$ | 99.15 | 45.57 | 86.02 | 79.44 | 32.45 | 90.98 | 99.52 | 46.20 | 79.28 | 65.55 | 24.66 | 96.24 | 0.40 | 99.78 |
| | | $KNN_{20}$ | 99.19 | 44.11 | 86.89 | 80.35 | 33.48 | 92.19 | 99.60 | 44.79 | 79.79 | 65.36 | 27.40 | 95.62 | 0.50 | 99.73 |
| | | $KNN_{50}$ | 99.23 | 42.39 | 87.99 | 81.66 | 36.78 | 91.59 | 99.60 | 43.07 | 80.90 | 64.68 | 43.84 | 94.57 | 0.80 | 99.63 |
| | | $KNN_{100}$ | 99.19 | 41.46 | 89.02 | 82.63 | 40.60 | 91.15 | 99.60 | 42.14 | 82.10 | 64.32 | 52.05 | 93.49 | 1.20 | 99.53 |
| | | $KNN_{200}$ | 99.23 | 40.61 | 90.88 | 83.30 | 45.41 | 90.40 | 99.60 | 41.26 | 83.78 | 63.89 | 65.75 | 92.17 | 2.10 | 99.36 |
| $ViT_{Base}$ | | *Pretrain on ImageNet→ Finetune on RVL-CDIP (ID)* | | | | | | | | | | | | | | |
| | 91.03 | MSP | 69.68 | 86.81 | 69.67 | 87.88 | 72.25 | 80.78 | 69.38 | 86.61 | 70.24 | 85.52 | 67.12 | 85.97 | 58.50 | 91.47 |
| | | MaxLogit | 63.35 | 89.20 | 68.40 | 88.58 | 69.58 | 84.38 | 61.08 | 89.94 | 65.60 | 88.02 | 57.53 | 89.41 | 48.40 | 93.04 |
| | | Energy | 62.22 | 89.21 | 70.34 | 88.43 | 70.26 | 84.37 | 60.75 | 90.03 | 65.89 | 88.01 | 58.90 | 89.47 | 49.70 | 93.03 |
| | | $Maha_{Norm}$ | 92.05 | 67.35 | 92.34 | 65.45 | 92.09 | 62.99 | 92.30 | 66.35 | 92.20 | 65.54 | 97.26 | 55.92 | 85.50 | 72.82 |
| | | $Maha_{UnNorm}$ | 92.05 | 67.35 | 92.34 | 65.45 | 92.09 | 62.99 | 92.30 | 66.35 | 92.20 | 65.54 | 97.26 | 55.92 | 85.50 | 72.82 |
| | | $KNN_{10}$ | 68.10 | 88.99 | 54.90 | 92.30 | 53.44 | 88.05 | 58.19 | 91.34 | 58.66 | 90.17 | 38.36 | 95.02 | 22.90 | 96.71 |
| | | $KNN_{20}$ | 67.61 | 88.95 | 49.01 | 92.85 | 51.53 | 89.25 | 58.59 | 91.16 | 56.68 | 90.55 | 41.10 | 94.47 | 25.40 | 96.35 |
| | | $KNN_{50}$ | 67.29 | 88.91 | 42.54 | 93.15 | 53.96 | 88.43 | 58.75 | 90.88 | 55.64 | 90.34 | 42.47 | 93.60 | 29.90 | 95.78 |
| | | $KNN_{100}$ | 66.19 | 88.90 | 43.80 | 93.19 | 55.71 | 87.73 | 59.11 | 90.64 | 56.20 | 90.12 | 45.21 | 92.86 | 34.90 | 95.27 |
| | | $KNN_{200}$ | 66.07 | 88.96 | 47.27 | 92.73 | 58.57 | 86.99 | 60.43 | 90.50 | 58.08 | 89.80 | 46.58 | 92.12 | 38.50 | 94.72 |
| | | *Pretrain on ImageNet* | | | | | | | | | | | | | | |
| | – | $KNN_{10}$ | 98.90 | 41.98 | 90.96 | 77.15 | 34.87 | 90.69 | 99.40 | 41.21 | 81.03 | 62.76 | 54.79 | 94.27 | 10.80 | 98.47 |
| | | $KNN_{20}$ | 98.94 | 40.54 | 91.67 | 77.20 | 36.82 | 91.71 | 99.44 | 39.85 | 81.72 | 62.32 | 64.38 | 93.57 | 12.70 | 98.25 |
| | | $KNN_{50}$ | 99.07 | 38.75 | 92.61 | 76.99 | 40.00 | 91.17 | 99.52 | 38.14 | 82.80 | 61.26 | 75.34 | 92.47 | 15.90 | 97.87 |
| | | $KNN_{100}$ | 99.11 | 37.43 | 93.25 | 76.56 | 43.38 | 90.68 | 99.56 | 36.93 | 83.82 | 60.40 | 82.19 | 91.52 | 18.90 | 97.49 |
| | | $KNN_{200}$ | 99.11 | 35.99 | 94.27 | 75.93 | 46.92 | 90.11 | 99.56 | 35.57 | 84.96 | 59.40 | 89.04 | 90.46 | 22.90 | 97.01 |

Table 12: OOD detection performance for document classification. All models are pretrained on IIT-CDIP. For LayoutLM models, we adopt the checkpoints from the Huggingface model hub. For UDoc, we pretrain the model on our side. All models are finetuned on RVL-CDIP ID data.

| | ID Acc | Method | OOD Dataset (In-Domain) | | | | | | | | | | Sci. Report | | OOD Dataset (Out-Domain) | | | |
|---|---|---|---|---|---|---|---|---|---|---|---|---|---|---|---|---|---|---|
| | | | Sci. Report | | Presentation | | Form | | Letter | | Average | | Sci. Poster | | Receipt | |
| | | | FPR95 | AUROC | FPR95 | AUROC | FPR95 | AUROC | FPR95 | AUROC | FPR95 | AUROC | FPR95 | AUROC | FPR95 | AUROC |
| $\text{LayoutLMv1}_{Base}$ | 97.28 | MSP | 47.48 | 74.91 | 59.74 | 68.72 | 66.40 | 65.36 | 58.89 | 69.12 | 58.13 | 69.53 | 43.02 | 77.15 | 72.40 | 62.40 |
| | | MaxLogit | 27.06 | 92.38 | 37.97 | 91.52 | 45.65 | 88.36 | 35.92 | 91.22 | 36.65 | 90.87 | 24.42 | 94.96 | 57.30 | 86.70 |
| | | Energy | 27.06 | 92.40 | 37.97 | 91.54 | 45.65 | 88.36 | 35.92 | 91.23 | 36.65 | 90.88 | 24.42 | 94.97 | 57.30 | 86.70 |
| | | $\text{Maha}_{Norm}$ | 17.73 | 96.67 | 33.83 | 94.27 | 36.95 | 92.47 | 24.55 | 95.34 | 28.26 | 94.69 | 13.95 | 97.49 | 43.60 | 94.74 |
| | | $\text{Maha}_{UnNorm}$ | 22.86 | 90.33 | 37.61 | 83.70 | 41.42 | 82.77 | 36.16 | 83.43 | 34.51 | 85.06 | 20.93 | 90.95 | 25.80 | 93.06 |
| | | $\text{KNN}_{10}$ | 20.82 | 96.09 | 35.32 | 93.82 | 40.06 | 91.34 | 28.65 | 94.80 | 31.21 | 94.01 | 17.44 | 97.00 | 49.80 | 93.92 |
| | | $\text{KNN}_{20}$ | 21.74 | 95.93 | 36.20 | 93.77 | 41.42 | 91.12 | 30.44 | 94.61 | 32.45 | 93.86 | 17.44 | 96.82 | 51.70 | 93.73 |
| | | $\text{KNN}_{50}$ | 24.34 | 95.56 | 38.25 | 93.41 | 43.93 | 90.69 | 33.64 | 94.19 | 35.04 | 93.46 | 23.26 | 96.44 | 53.80 | 93.70 |
| | | $\text{KNN}_{100}$ | 25.54 | 95.30 | 39.13 | 93.20 | 45.17 | 90.35 | 34.78 | 93.99 | 36.16 | 93.21 | 25.58 | 96.24 | 54.70 | 93.45 |
| | | $\text{KNN}_{200}$ | 26.54 | 95.04 | 39.53 | 92.95 | 46.21 | 89.97 | 35.75 | 93.80 | 37.01 | 92.94 | 25.58 | 96.04 | 56.80 | 93.10 |
| | | $\text{KNN}_{400}$ | 27.62 | 94.76 | 40.22 | 92.65 | 47.29 | 89.54 | 36.69 | 93.61 | 37.96 | 92.64 | 30.23 | 95.84 | 57.60 | 92.62 |
| $\text{LayoutLMv3}$ | 97.81 | MSP | 56.16 | 70.81 | 63.44 | 67.17 | 67.16 | 65.30 | 58.60 | 69.58 | 61.34 | 68.22 | 52.33 | 72.70 | 43.60 | 77.10 |
| | | MaxLogit | 30.70 | 89.17 | 40.42 | 88.18 | 42.98 | 84.09 | 33.12 | 88.22 | 36.80 | 87.42 | 19.77 | 94.50 | 11.70 | 97.02 |
| | | Energy | 30.70 | 89.18 | 40.42 | 88.18 | 42.98 | 84.10 | 33.12 | 88.23 | 36.80 | 87.42 | 19.77 | 94.51 | 11.70 | 97.03 |
| | | $\text{Maha}_{Norm}$ | 99.16 | 16.28 | 98.51 | 35.13 | 99.12 | 20.45 | 99.19 | 14.42 | 99.00 | 21.57 | 100.00 | 7.79 | 98.50 | 42.00 |
| | | $\text{Maha}_{UnNorm}$ | 99.76 | 15.30 | 99.16 | 34.20 | 99.52 | 19.77 | 99.92 | 12.86 | 99.59 | 20.53 | 100.00 | 7.57 | 98.30 | 42.14 |
| | | $\text{KNN}_{10}$ | 21.74 | 95.03 | 35.68 | 93.38 | 32.88 | 91.86 | 18.51 | 96.26 | 27.20 | 94.13 | 11.63 | 97.58 | 8.90 | 97.97 |
| | | $\text{KNN}_{20}$ | 22.74 | 94.90 | 36.56 | 93.20 | 33.96 | 91.66 | 19.64 | 96.15 | 28.22 | 93.98 | 12.79 | 97.44 | 10.00 | 97.89 |
| | | $\text{KNN}_{50}$ | 24.62 | 94.62 | 38.37 | 92.71 | 35.83 | 91.38 | 21.63 | 95.93 | 30.11 | 93.66 | 13.95 | 97.20 | 10.70 | 97.72 |
| | | $\text{KNN}_{100}$ | 25.22 | 94.38 | 39.29 | 92.32 | 36.55 | 91.09 | 22.48 | 95.79 | 30.88 | 93.40 | 16.28 | 97.04 | 11.80 | 97.59 |
| | | $\text{KNN}_{200}$ | 26.02 | 94.13 | 39.82 | 91.91 | 37.19 | 90.78 | 23.30 | 95.68 | 31.58 | 93.12 | 18.60 | 96.91 | 12.80 | 97.46 |
| | | $\text{KNN}_{400}$ | 26.42 | 93.86 | 40.46 | 91.45 | 37.95 | 90.65 | 23.70 | 95.58 | 32.13 | 92.82 | 18.60 | 96.81 | 14.10 | 97.33 |
| $\text{UDoc}_{ResNet50}$ | 97.36 | MSP | 66.13 | 65.73 | 69.43 | 64.09 | 71.03 | 63.28 | 71.06 | 63.25 | 69.41 | 64.09 | 40.70 | 78.47 | 39.80 | 78.99 |
| | | MaxLogit | 45.96 | 82.12 | 47.21 | 86.39 | 49.64 | 83.16 | 49.59 | 83.13 | 48.10 | 83.70 | 2.33 | 98.57 | 4.00 | 98.34 |
| | | Energy | 45.96 | 82.12 | 47.21 | 86.40 | 49.64 | 83.16 | 49.59 | 83.13 | 48.10 | 83.70 | 2.33 | 98.60 | 4.00 | 98.36 |
| | | $\text{Maha}_{Norm}$ | 93.31 | 53.49 | 94.70 | 50.13 | 93.93 | 53.21 | 95.37 | 50.02 | 94.33 | 51.71 | 82.56 | 71.06 | 94.90 | 47.24 |
| | | $\text{Maha}_{UnNorm}$ | 94.16 | 50.58 | 94.62 | 50.19 | 94.37 | 49.56 | 95.29 | 49.30 | 94.61 | 49.91 | 87.21 | 69.12 | 94.90 | 49.92 |
| | | $\text{KNN}_{10}$ | 30.02 | 94.47 | 41.22 | 88.66 | 41.90 | 90.99 | 36.65 | 93.48 | 37.45 | 91.90 | 1.16 | 99.13 | 5.50 | 98.42 |
| | | $\text{KNN}_{20}$ | 31.10 | 94.36 | 41.98 | 88.44 | 42.10 | 90.90 | 38.03 | 93.35 | 38.30 | 91.76 | 1.16 | 99.04 | 6.90 | 98.32 |
| | | $\text{KNN}_{50}$ | 33.95 | 94.07 | 43.35 | 87.89 | 44.01 | 90.72 | 40.71 | 93.06 | 40.51 | 91.43 | 1.16 | 98.84 | 7.40 | 98.26 |
| | | $\text{KNN}_{100}$ | 34.83 | 93.84 | 43.75 | 87.51 | 45.01 | 90.61 | 41.96 | 92.90 | 41.39 | 91.22 | 1.16 | 98.72 | 8.30 | 98.16 |
| | | $\text{KNN}_{200}$ | 35.63 | 93.63 | 44.11 | 87.08 | 45.29 | 90.57 | 42.49 | 92.79 | 41.88 | 91.02 | 1.16 | 98.65 | 8.60 | 98.08 |
| | | $\text{KNN}_{400}$ | 36.39 | 93.29 | 44.80 | 86.43 | 45.65 | 90.65 | 42.94 | 92.60 | 42.44 | 90.74 | 1.16 | 98.61 | 9.10 | 98.04 |

