# OpenReview forum: "A Critical Analysis of Out-of-Distribution Detection for Document Understanding"
_ICLR.cc/2023/Conference — Submitted to ICLR 2023_

### Official Review · Reviewer_oNdB · 2022-10-24

**Confidence:** 5
**Correctness:** 3
**Technical Novelty And Significance:** 2
**Empirical Novelty And Significance:** 3
**Recommendation:** 5

**Clarity, Quality, Novelty And Reproducibility:**

* Clarity:  the manuscript is well written and clear.
* Quality: the flow of experiments is clear and logical.
* Novelty: the novelty is marginal.
* Reproducibility: enough details are included to enable reproducing the results contingent on the data being publicly released.

**Strength And Weaknesses:**

Strengths:
* This paper provides a first benchmark dataset for out-of-domain detection for document understanding.
* It provides comprehensive experiment studies on the different categories of multimodal transformer based models for OOD detection.

Weaknesses:
* The distinction between the in-domain and out-domain types of OOD data is loose. There should be a more rigorous way of defining this two categories based on some concrete similarity metrics.
* The experiments design is not convincing. For example, the impact of fine-tuning on improving the OOD detection is not surprising, especially as you have defined the in-domain OOD data based on the classification accuracy in the first place.
* The conclusions drawn from the experiments are not rigorous enough (e.g. Figure 5). For example, in zero-shot OOD detection, the architecture of the model seems to play a more important role than the OOD data being in-domain vs out-domain.

**Summary Of The Paper:**

This paper provides analysis of out-of-domain detection for different tasks in document understanding domain, including document classification and information extraction. Authors create a benchmark dataset for out-of-domain detection and study the performance a 4 main categories of models for document based on text, image, text+layout, and text+layout+image.

**Summary Of The Review:**

Overall, this paper needs improvements to meet the bar of ICLR. Specifically, the in-domain vs out-domain data needs to be more rigorously defined and experiments designs be modified to make it easier to understand the impact of model architecture vs the data domain, thus making the conclusions more reliable.

---

### Official Review · Reviewer_4z2u · 2022-10-25

**Confidence:** 4
**Correctness:** 3
**Technical Novelty And Significance:** 3
**Empirical Novelty And Significance:** 3
**Recommendation:** 6

**Clarity, Quality, Novelty And Reproducibility:**

* For clarity, the authors need to enhance their presentation by significantly polishing organization and writing.
* The results are interesting with good technical quality, and the proposed spatial-aware adapter also provides the novelty.
* As mentioned in weakness, the paper does not provide the information about reproducibility.


**Strength And Weaknesses:**

Strengths
* In-depth analysis shows the motivation and the potential of leveraging spatial information for document understanding.
* Extensive experiments not only demonstrate comprehensive comparisons, but also indicate that spatial-aware models outperform conventional models.

Weaknesses
* Writing needs to be significantly polished. The paper is not very easy to follow. Plus, there are many typos like “special-aware”.
* The authors do not mention the plan of open-sourcing their implementation of spatial-aware models and analysis scripts for the reproducibility.


**Summary Of The Paper:**

In this paper, the authors analyze the characteristics of spatial information in document understanding and propose a framework to leverage spatial information for document OOD detection. They start from an analysis on the effects across different OOD types and emphasize the importance of spatial information for document understanding. Following the observation, the authors propose to employ an add-on module and adapt transformer-based neural language models to document domains. Analysis and experiments on RVL-CDIP and IIT-CDIP demonstrate their observation and the improvement of Spatial-aware models over the conventional ones.

**Summary Of The Review:**

In sum, I would recommend “8: accept, good paper” based on the extensive analysis and experimental results. Although there are some flaws in presentation and clarity, I believe the authors should be able to address the issues in the final version.

Edit: After the reviewer discussion, we decide to change to the score to "6: marginally above the acceptance threshold".

---

### Official Review · Reviewer_Dbcz · 2022-11-05

**Confidence:** 4
**Correctness:** 4
**Technical Novelty And Significance:** 2
**Empirical Novelty And Significance:** 2
**Recommendation:** 3

**Clarity, Quality, Novelty And Reproducibility:**

Clarity - The paper presentation and writing style is clear and a reader in this domain can understand the material.

Quality - Please see the Weakness section for several issues in experimental

Novelty - not sufficiently novel for ICLR. The paper is more of an analysis piece and it has issues which need to be addressed.

Reproducibility - no code shared

**Strength And Weaknesses:**

Strengths
1. The authors position the paper as a analysis of OOD for document understanding domain. Given the practical implications and prevalence of documents, this is important work.
2. The authors evaluate recent transformer based document understanding models for ID and OOD behavior and try to made deductions based on experiments.

Weaknesses
1. The authors inherently start with an assumption that OOD detection is multi-modal and that single modality cannot solve OOD. They do not show experimental justification on this assumption. Why is a single modality not sufficient?
2. Miss citing relevant document understanding papers see below. In fact DocFormer has the sota performance on RVL-CDIP dataset (main considered dataset in this paper) and is not cited, nor discussed.
3. Page 3 - the notation does not seems correct. If threshold gamma decides ID vs OOD, it is possible a model f, wrongly predicts a label for ID test-data. the authors do not dis-ambiguate this scenario in notation.
4. Page 4 - the authors miss gradient based OOD detection. Logit and distance based are not the only broad research directions for OOD.
5. Fig. 2 (a) is not entirely correct. e.g. in Spatial+Vision+Language have spatial embedding layers. The authors do not make it clear how this is different from their spatial-aware adapter and in which way it benefits over the embedding approach.
6. The deductions made in Sec 3.1 seem faulty and has some un-answered assumptions and . e.g. for receipts the authors claim “improvement of fine-tuning is less significant for out-domain OOD data.” It’s possible that the pre-training data had several receipts which led to ViT exhibiting this behavior. Also why was this experiment only done using vision-only model? its possible text/multi-modal exhibit different behaviors
7. Page 7, Zero-shot performance. The experiment seems faulty. The IIT-CDIP data the authors use, have the data which the authors deem OOD although in un-labelled form (IIT-CDIP has letters, forms, etc.) For strict OOD zero-shot performance, shouldn’t the authors remove such data? Please comment.

[1] DocFormer - end to end document understanding using transformers ICCV 2021
[2] more recent document understanding papers also not cited.

**Summary Of The Paper:**

Summary: The authors aim to provide a systematic and in-depth analysis on OOD detection for document understanding models. They deduce that spatial information is critical for document OOD detection. The authors also propose a simple yet effective special-aware adapter, which serves as an add-on module to adapt transformer-based language models to document domain. They perform experiments to show that their method consistently improves ID accuracy and OOD detection performance compared to baselines.

**Summary Of The Review:**

Based on the weaknesses in experimental setup, I vote reject for ICLR 2023.

---

### Decision · Program_Chairs · 2023-01-20

**Decision:**

Reject

**Justification For Why Not Higher Score:**

I tend to agree with the negative review that this paper is ultimately unsurprising. Sure, multiple modalities can only improve performance and that is what is found. Moreover, there are no especially novel algorithmic or architectural ideas here and nothing that would seem to apply to other domains. Basically, it's kind of engineering---anyone building a document OOD detector should be able to land upon this solution. Putting the code up might be a valuable open-source contribution but it's not clear the scientific contribution is important enough. If this paper is rejected, the authors might be encouraged to submit to a more specialized venue. It could also be accepted if the chairs feel that there will be a broad enough audience interested in the result.

**Justification For Why Not Lower Score:**

N/A

**Metareview: Summary, Strengths And Weaknesses:**

This paper addresses the problem of detecting out-of-distribution documents in a document dataset, which is useful for the downstream task of document understanding. In particular, the novel contribution is a relatively straightforward extension to a multi-modal architecture whereas previous approaches were all single modal. Experiments show that using multiple modes indeed does improve performance at this task.

The strengths of this paper are that it highlights the fact that for the task of OOD document detection, multiple modalities are useful. Some of the reviewers found this to be a useful message to practitioners in the field, while one reviewer found this to be unsurprising and of interest only to specialists in the field. The weaknesses of the paper include: being unsurprising and being focused on a task that is borderline general interest.

**Summary Of Ac-Reviewer Meeting:**

The two positive reviewers agreed and explained their enthusiasm for the paper because they work or have directly worked on this problem. They also felt that the authors addressed the concerns of the negative reviewer. The negative reviewer was not able to make the group meeting due to illness but I synced with them afterwards. They acknowledged that the author's rebuttal addressed some of their concerns but still did not address their concerns of generality/surprisingness.